# Right Side Up? Disentangling Orientation Understanding in MLLMs with Fine-grained Multi-axis Perception Tasks

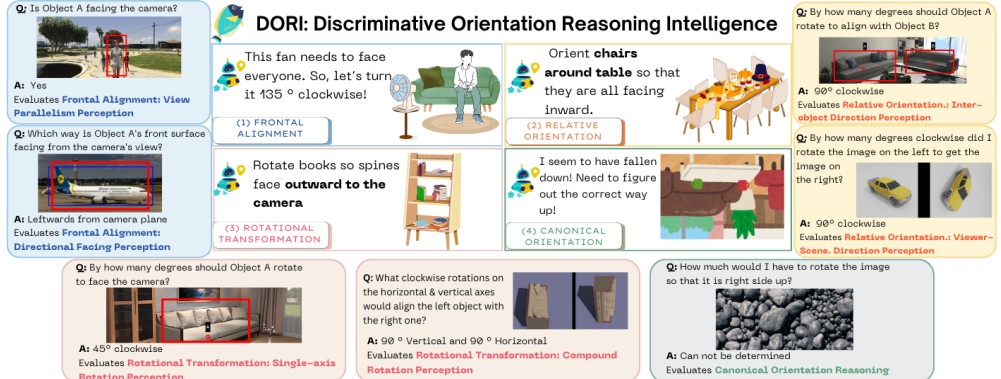

Figure 1: DORI captures four core dimensions of orientation reasoning intelligence: (1) object's **directional alignment**, (2) its **orientation relative to** viewers, scenes, and other objects, (3) required **rotational transformation** for different objectives, and (4) its natural/**canonical orientation** in the world. Each dimension evaluates specific perceptual abilities through visual tasks in varying settings. DORI provides a holistic understanding of object orientation reasoning.

## ABSTRACT

Object orientation understanding represents a fundamental challenge in visual perception that underpins critical real-world applications like robotic manipulation and augmented reality. However, current vision-language benchmarks fail to isolate and evaluate this core capability, often conflating it with positional relationships (such as above/below or proximity between objects) and general scene understanding. To address this, we introduce **D**iscriminative **O**rientation **R**easoning **I**ntelligence (**DORI**), a comprehensive hierarchical benchmark that establishes object orientation perception as a primary evaluation target. DORI rigorously assesses four essential dimensions of object(s) orientation comprehension: frontal alignment, rotational transformations, relative directional relationships, and canonical orientation understanding. DORI provides valuable insights on how existing multi-modal systems process and understand object orientations through carefully curated tasks from 14 sources that spans 67 object categories across synthetic and real-world scenarios. Our evaluation of 22 state-of-the-art vision-language models using DORI reveals critical limitations: even the best models achieve only $54.2\%$ accuracy on coarse tasks and $45.0\%$ on granular orientation judgments, with performance deteriorating substantially for tasks requiring reference frame shifts or compound rotations. These findings demonstrate the urgent need for dedicated orientation representation mechanisms in future architectures, as models show a systematic inability to perform precise angular estimations, track orientation changes across multiple viewpoints, and understand compound rotations—suggesting fundamental limitations in their internal 3D spatial representations. As the first diagnostic framework specifically designed for advancing orientation awareness in multimodal systems, DORI offers immediate implications for improving robotic control, 3D scene reconstruction, and human-AI interaction in physical environments

## 1 INTRODUCTION

Object orientation understanding demands complex, multi-stage processing of intrinsic object features, viewer perspective, angular relationships, and reference frame transformations (Harris, 2024; Kallmayer et al., 2023; Viganò et al., 2023; Chavez et al., 2023). Humans master this fundamental aspect of visual cognition (Cohn, 1997; Alomari et al., 2022) through various inherent sensory-motor experiences, proprioceptive integration and neural formation (Tuthill & Azim, 2018). In the human cognitive system, these mechanisms develop progressively from basic frontal orientation recognition to complex rotational transformations (Tversky & Suwa, 2009; Mallot, 2023; Vasilyeva & Lourenco, 2012). This equips us with a crucial aptitude for real-world visual tasks that require precise spatial interaction with the environment, such as tool manipulation and navigation. Many applications still require human-like sophisticated orientation understanding capabilities. For example, autonomous vehicles must determine which way objects are facing for effective navigation (Ganesan et al., 2024; Janai et al., 2020). Similarly, augmented reality requires alignment of virtual objects with physical ones (Mu et al., 2023; Pei et al., 2024; Wang et al., 2023a), and robotic grasping has to understand an object's orientation to determine approach angles (Cong et al., 2021; Chen et al., 2023).

Multimodal large language models (MLLMs) (Bi et al., 2024; Wang et al., 2024; Touvron et al., 2023) are attractive for many of these applications as they perform well across many tasks (Guan et al., 2024; Lin et al., 2024; Pei et al., 2024; Mu et al., 2023; Wei et al., 2024; Gao et al., 2024). Prior work has found that MLLMs perform poorly on object orientation (Tong et al., 2024a; Li et al., 2024a; Cheng et al., 2024; Chen et al., 2024; Liu et al., 2025; Lei et al., 2025). However, these benchmarks consider a narrow understanding of orientation. For example, some focus on simple directional judgments (Weston et al., 2016; Shi et al., 2022), use only synthetic data (Wang et al., 2025b; Li et al., 2023), employ ambiguous question wording (Mirzaee et al., 2021; Mirzaee & Kordjamshidi, 2022), test only egocentric viewpoints (Jung et al., 2025), or have very few samples (Fu et al., 2024a). These limitations lead to an incomplete assessment of a model's orientation reasoning abilities, potentially resulting in a failure to identify critical weaknesses in real-world scenarios and inability to distinguish between true geometric understanding versus memorized patterns or statistical shortcuts.

To address these shortcomings, we introduce **D**iscriminative **O**rientation **R**easoning **I**ntelligence (**DORI**), a human cognitive study-informed benchmark to evaluate object orientation understanding in multimodal language models. As summarized in Fig. 1, we decompose evaluation of this critical ability into four fundamental dimensions with progressive complexity: (1) frontal alignment perception, (2) rotational transformations, (3) ego and allocentric relative orientation understanding, and (4) natural or canonical orientation of objects. Furthermore, for a holistic view of a given model's performance, we employ a two-tiered assessment framework – **coarse-grained** questions to evaluate basic categorical understanding (*e.g.*, "is the car facing toward or away from the camera?") and **fine-grained** questions to probe precise metric relationships (*e.g.*, "at what angle is the car oriented relative to the camera?"). DORI leverages **13652** images from **14** sources to generate **33,656** multiple-choice questions, combining real-world images (37%) with simulated renders (63%) to ensure we have a large dataset with varying levels of visual complexity.

We generate clear, unambiguous prompt-answer pairs through a rigorous three-step process: (1) isolating objects with bounding boxes to tackle cluttered scenes, (2) employing standardized orientation terminology (*e.g.*, "frontal alignment") with explicit spatial frames of reference (*e.g.*, egocentric, allocentric, and object-centric), examples, and task descriptions, and (3) ensuring difficulty progression from simple categorical judgments to precise angular measurements across all orientation dimensions. This systematic approach isolates orientation from scene perception skills, minimizes confounding factors such as object recognition difficulty, scene clutter, linguistic ambiguity, and contextual distractions that plague existing benchmarks (Cheng et al., 2025; Chen et al., 2024; Li et al., 2024b). Our extensive experiments with 22 state-of-the-art MLLMs on DORI reveal several key findings:

- Models perform 30% worse on complex, dynamic rotational tasks that require mental tracking of object rotations between images than simple, static orientation tasks (*e.g.*, identifying object poses).
- Models particularly struggle on tasks requiring perspective shifts (*e.g.*, adapting viewpoints different from the camera, such as determining if two objects are facing each other when viewed from their own frame of reference), showing a 25% drop in accuracy compared to egocentric frame tasks.
- Token-based integration (like Mantis-Idefics2-8B) consistently outperforms linear projection in orientation tasks, indicating that architectural design significantly impacts orientation reasoning.

Table 1: **Characteristics of different datasets for orientation-reasoning**. Asterisks (*) indicate that we only counted the number of classes, sources, or samples refers specifically to orientation-related tasks. DORI is larger and/or more diverse than similar datasets. *N-Imgs = # Natural Images, S-Imgs = # Simulated Images, A = Allocentric, E = Egocentric, C = Coarse, G = Granular*

| Dataset | Objects | Granularity | View | Sources | N-Imgs | S-Imgs | VQA Pairs |
|---|---|---|---|---|---|---|---|
| Spatial-MM (Shiri et al., 2024) | 342 | C | AE | 1 | 537 | – | 695 |
| ScanQA (Azuma et al., 2022) | 20 | C | A | 1 | 800 | – | ∼7445* |
| BLINK (Fu et al., 2024b) | 71* | C | AE | 2 | 552* | – | 552* |
| KiVA (Yiu et al., 2025) | 62* | G | A | 2* | – | 400* | 600* |
| EgoOrientBench (Jung et al., 2025) | 197 | C | E | 5 | 5992 | 2373 | 33460 |
| EmbSpatial-Bench (Du et al., 2024) | 294 | C | E | 3 | 1498* | 683* | 2435* |
| CLEVR-3D (Yan et al., 2023) | 160 | C | A | 2 | 1044 | – | 2320* |
| 3DSRBench (Ma et al., 2025) | 7* | C | A | 1* | 2073* | – | 6188* |
| PO3D-VQA (Wang et al., 2023b) | 10 | C | A | 1 | – | ∼30k | ∼300k* |
| SR-Bench (Stogiannidis et al., 2025) | 197 | C | E | 2* | 280* | 520* | 800* |
| Spatial457 (Wang et al., 2025c) | 5* | C | A | 1 | – | 992* | 4555* |
| **DORI (Ours)** | **67** | **CG** | **AE** | **14** | **5051** | **8601** | **33656** |

- Model scale alone does not guarantee better orientation understanding; smaller, dialogue-tuned variants (*e.g.*, DeepSeek-1.3B-Chat) often outperform larger base models (*e.g.*, DeepSeek-7B-Base).
- LoRA finetuning with our dataset can also boost performance by up to 27% on other benchmarks like BLINK (Fu et al., 2024b), SAT (Ray et al., 2025), and 3DSRBench (Stogiannidis et al., 2025).

## 2 Related Work

Tab. 1 compares our proposed benchmark, DORI, to existing datasets for evaluating MLLMs with orientation questions. These datasets only provide a limited evaluation of orientation with small scales or only basic questions (*e.g.*, "is the object facing left or right?"), with most of their questions focusing on spatial reasoning tasks instead. In contrast, DORI introduces a comprehensive framework that isolates orientation perception through carefully designed tasks spanning four core aspects. In particular, DORI contains a larger number of samples from at least three times as many sources, and has a large variety of objects, question granularity, and views.

For example, PO3D-VQA (Wang et al., 2023b) evaluates a MLLM's ability to understand eight cardinal directions with tolerance. EgoOrientBench's (Jung et al., 2025) discrete orientation classes discard crucial continuous rotational information, resulting in identical scoring for models with significantly different error margins. DORI addresses this by introducing hierarchical evaluation through coarse to granular QA pairs. Additionally, some benchmarks rely entirely on simulated data (Wang et al., 2023b; Yiu et al., 2025), raising questions about generalization to the real-world. Most datasets with natural images have fewer than 2100 samples, whereas our work has 2.5K more. This puts DORI on par with EgoOrientBench, but with more diverse questions, sources, and views.

Our work has some relation to pose estimation (Deng et al., 2024; Du et al., 2021; Zheng et al., 2023; Barroso-Laguna et al., 2024; Biggs et al., 2020), where models are often tasked with identifying a set of keypoints to provide a detailed accounting of an object's orientation. This format requires using special techniques to evaluate MLLMs (Corsetti et al., 2024; Feng et al., 2024; Pulli et al., 2024), and are expensive to annotate. Further, these keypoints are not easily human-interpretable format. For example, the upper-left question of Fig. 1, where a model is asked if the object is facing the camera, a keypoint estimation model would require a complex function to map the pose into the direction it is facing as there are many poses that human annotators would consider is facing forward. Instead, DORI evaluates a MLLM on its ability to understand orientation question from natural language.

## 3 *DORI*: Discriminative Orientation Reasoning Intelligence

Evaluating a model's understanding of object orientation mandates a closer inspection of several fundamental research questions including how well a MLLM matches human performance, if MLLMs

and humans follow similar developmental patterns (Blades & Spencer, 1994; Vasilyeva & Lourenco, 2012; Spinelli et al., 1999), and what architectural or training components contribute to their reasoning gaps. These critical questions guide the task structure, data sample selection, and evaluation probe development in DORI to effectively dissect the orientation reasoning capabilities of MLLMs. In this work, we aim to ascertain MLLMs' understanding of object orientation across static scenarios and dynamic manipulations in natural and simulated settings. This distinction is crucial as it mirrors the progression from simple to complex spatial reasoning observed in human cognitive development. We discuss the four aspects of object orientation comprehension that guide our benchmark creation below.

## 3.1 CORE CAPABILITIES

Drawing on the established frameworks in cognitive neuroscience (see App. A), DORI decomposes object orientation comprehension into four fundamental dimensions that reflect distinct neural and cognitive processes humans employ when reasoning about an object's orientation.

**1. Frontal Alignment** evaluates the fundamental ability to perceive how an object's front-facing surface is oriented relative to the viewer—a prerequisite for any orientation-based reasoning. Humans rapidly identify which way an object is facing (*e.g.*, deciding if a vehicle is approaching or departing) by recognizing structural and functional features such as faces, headlights, or entrances. However, we require additional reasoning steps for orientation interpretation (Kourtzi & Nakayama, 2002), such as assessing objects' angular relationship with the viewing plane defined as the imaginary plane onto which a scene is projected onto. For MLLMs, we assess this frontal alignment capability through two complementary tasks: **view parallelism analysis**, which quantifies the degree to which an object's frontal surface deviates from being parallel to the image plane, providing angular measurements (*e.g.*, is a chair directly facing the camera or at a 45-degree angle). **Directional facing perception** asks the cardinal direction an object's front is oriented relative to the camera position (*e.g.*, whether a desk is facing toward, away, leftward, or rightward from the viewer's perspective). This dual assessment is supported by research that reports viewpoint-invariant recognition of frontal features operates through different neural mechanisms than precise angular estimation (Harris, 2024). Prior work also demonstrates MLLMs' inability to perceive object frontality (*e.g.*, confusing left/right perspectives) even when provided with bounding boxes (Tong et al., 2024b; Shiri et al., 2024), often being on par with random predictions (Yin et al., 2025). For objects whose frontality is difficult to define (*e.g.*, symmetric objects like a ball), models are expected to respond with the "Cannot be determined" option, which is provided as a choice in every question.

**2. Rotational Transformation** examines the ability of an MLLM to comprehend orientation changes through rotation, reflecting requirements such as embodied object manipulation (*e.g.*, fitting a key into a lock), and viewpoint-dependent navigation (*e.g.*, reorienting a rotated map for wayfinding). Mental rotation capabilities allow humans to predict how objects align when rotated (Muto, 2021), with neuroimaging evidence showing premotor and parietal activation during both physical & imagined rotations (Doganci et al., 2023; Muto, 2021). Both processes trigger similar neural activations (Xue et al., 2017), highlighting the inherent complexity of spatial transformation tasks that MLLMs must simulate. Inspired by this, we design the rotational tasks in our benchmark to progress from simple to complex levels mirroring human cognitive processing demands (Li et al., 2019; Ter Horst et al., 2010; Xue et al., 2017). Thus, the first subtask examines **single-axis rotation** & evaluates basic angular transformations rotated along one spatial dimension (*e.g.*, determining the shortest rotation to face a chair toward the camera). This establishes baseline capabilities before progressing to more cognitively demanding compound rotation (Ter Horst et al., 2010), involving **multiple-axes rotation** (*e.g.*, aligning objects through a sequence of horizontal/vertical rotations). These subtasks represent common scenarios in assembling products through item manipulation and real-world scene planning by embodied agents, where object orientation understanding directly impacts final task performance.

**3. Relative Orientation** examines the understanding of how objects are oriented in relation to each other and with respect to the viewer. Humans navigate a complex visual world by seamlessly tracking orientation changes across scenes and viewpoints, which is crucial for scene comprehension and geometric coordination. The human brain contains a specific interconnected region facilitating "mental orientation" -the ability to effectively spatially orient an object to different viewpoints and perspectives (Peer et al., 2015; Loy & Demberg, 2023; Gramann et al., 2010; Zaehle et al., 2007). In contrast, studies have shown that MLLMs struggle significantly with questions posed from non-egocentric perspectives (Shiri et al., 2024; Wang et al., 2023a; Yeh et al., 2025) and with maintaining

## DORI Prompt Structure

**Expected Response Format**
"answer": "A",
"reasoning": "Object A appears to be a personwhose face and front of the body are clearlyvisible in the image. The person's eyes arelooking toward the camera, indicating thatObject A is facing toward the camera."

**1. TASK:** "Determine which way Object A's front is facing relative to the camera."

**2. CONTEXT:** "An object is considered 'front facing' when its inherent structural features are visible from the camera."

**3. INSTRUCTIONS:** "1. Identify Object A and its key structural features 2. Determine the orientation of its front surface relative to the camera..."

**4. MULTIPLE-CHOICE OPTIONS:** "A. Facing toward the camera B. Facing away from the camera C. Facing L...."

**5. EXAMPLES:** "A person whose body is directed towards the camera would be 'facing toward the camera'"

Figure 2: Structured Prompt Design in DORI. The five key components are: task description, contextual information, step-by-step instructions, multiple-choice options, and examples. The structured format ensures consistent evaluation of orientation perception abilities.

consistent dimensional relationships between multiple objects across time and viewpoints (Yang et al., 2025c) We systematically probe this aspect of object's relative orientation via the following sub-tasks: (1) **inter-object directional relationships**, to assess the relative facing directions of objects (*e.g.*, determine if two cars are facing the same or opposite directions), and (2) **image-pair rotational relationships**, to measure the ability to track orientation changes between two images (*e.g.*, identify the degree of rotation between two views of the same object).

**4. Canonical Orientation Perception** evaluates the ability to recognize when objects deviate from their expected orientations, and to determine what transformations would restore them to their canonical state (*e.g.*, identifying that a flipped image of a building is upside-down and needs 180-degree rotation to appear normal). Humans possess an innate ability to predict and understand the physical properties of objects and their interaction through specialized neural processing (Ballaz et al., 2005; Mezuman & Weiss, 2012; Harris, 2024) and use functional cues for inferring canonical orientation, such as gravity alignment (Fu et al., 2008), functional feature positioning, and ecological validity (Bramley et al., 2018; Hamrick et al., 2011). Prior work shows that MLLMs struggle to reason with intuitive physics (Buschoff et al., 2025; Jassim et al., 2024). In our benchmark, we break this complex task into two sub-tasks: first, assess the ability to identify deviations from canonical orientation across object categories, then evaluate a model's ability to determine the specific geometric operations (rotation, flipping, or combinations) required to restore the object to its canonical state.

### 3.2 BENCHMARK CREATION PROCESS

As illustrated in Fig. 1, DORI contains seven carefully designed orientation-reasoning tasks. Each multiple-choice question has two assessment levels: coarse-grained questions for basic categorical judgments and fine-grained questions for precise angular measurements. This hierarchical approach enables systematic evaluation from fundamental perception to advanced orientation reasoning. Our data was collected via two primary means: converting existing 3D information to orientation questions (sampled from JTA (Fabbri et al., 2018), 3D-Future (Fu et al., 2021), Get3D (Gao et al., 2022), ShapeNet (Guibas, 2017), OmniObject3D (Wu et al., 2023), NOCS REAL (Wang et al., 2019), Objectron (Ahmadyan et al., 2021), a collection we refer to as SSFRB (unsplash, 2020; Gontier et al., 2023; Liu et al., 2024c; Willett et al., 2013; neelgajare, 2022), the OminNOCS (Krishnan et al., 2024) subsets of KITTI (Geiger et al., 2012) and Cityscapes (Cordts et al., 2016)) or manually annotating samples (for COCO (Lin et al., 2014)). Each question type contains a subset of sources (discussed further in Sec. 3.3). For example, Inter-object direction perception requires at least two objects in a scene, so ShapeNet (Guibas, 2017), which contains single objects was not used, and instead created questions with 3D-Future (Fu et al., 2021) and NOCS REAL (Wang et al., 2019). Additionally, to account for objects that are symmetric/have ambiguity in frontality, we have included the option

"Cannot be determined" for all questions regardless of if this response is true or false for it. We provide a detailed description of the process used to curate samples from each dataset in App. B.

We designed DORI's evaluation prompts using a systematic, human-centered approach to isolate orientation perception from confounding factors (*e.g.* object recognition difficulty, scene clutter, linguistic ambiguity, and contextual distractions) (Castle, 2024), see App. B for detailed discussion. As highlighted in Fig. 2, the prompts follow a carefully structured format with five key components: (1) a concise task description specifying the orientation dimension being tested, (2) contextual information explaining relevant orientation concepts (*e.g.*, "An object is considered 'front facing' when its inherent structural features are visible from the camera"), (3) step-by-step analysis instructions (*e.g.* "1. Identify Object A and its key structural features 2. Determine the orientation...")(4) multiple-choice options, and (5) concrete examples illustrating expected reasoning (*e.g.* "A person whose body is directed towards the camera would be 'facing toward the camera'"). This structured approach was inspired by effective instruction-tuning datasets like LLaVA's, which demonstrate that explicit task framing and example-driven guidance significantly improve model comprehension (Hewing & Leinhos, 2024).

We iteratively refine our prompts through multiple cycles of human feedback from non-expert annotators to address ambiguities, clarify terminology, and improve the task specificity (Lin et al., 2025). For example, early versions of rotational transformation prompts yielded inconsistent interpretations of rotation axes, which required us to incorporate more precise language and visual references (*e.g.*, "like a ballerina spinning clockwise" to clearly illustrate vertical axis rotation versus abstract directional descriptions). This process ensured that the MLLMs performance differences genuinely reflect their object orientation understanding capabilities rather than prompt interpretation challenges (see App. B for further details). Additionally, we standardized response formats (requiring explicit answer responses and reasoning explanations) to facilitate consistent evaluation while providing insight into models' reasoning. See App. I for examples of our 14 question prompts.

To ensure a comprehensive coverage of object orientation reasoning, we developed a two-tiered question framework across four orientation dimensions.

- **Coarse-grained questions** evaluate basic categorical understanding (*e.g.*, "Has the object rotated between the two images?" for rotational transformation, or "Are objects A and B facing the same direction?" for relative orientation) and provide a foundation for assessing orientation perception.
- **Fine-grained questions** probe precise quantitative estimations (*e.g.*, "How many degrees clockwise did the object rotate?" for rotational transformation, or "What specific transformations would restore this image to its canonical orientation?" for canonical orientation perception), requiring detailed metric understanding of angular relationships.

### 3.3 DORI Statistics

Collectively, DORI encompasses **13, 652** images spanning both natural (37%) and simulated (63%) environments, has **33, 656** carefully constructed multiple-choice questions. The benchmark covers 67 object categories (**31** household and **36** outdoor item categories) across **14** diverse computer vision datasets including KITTI (Geiger et al., 2012), Cityscapes (Cordts et al., 2016), COCO (Lin et al., 2014), JTA (Fabbri et al., 2018), 3D-FUTURE (Fu et al., 2021), Objectron (Ahmadyan et al., 2021), ShapeNet (Guibas, 2017), and OmniObject3D (Wu et al., 2023), amoung others (full list in Sec. 3.2). Fig. 3 illustrates the object category and dataset distribution. This multi-source approach allows us to balance complex, real-world environments (37% of images) with controlled synthetic environments (63%) where orientation parameters are precisely known. Knowing precise orientation parameters in synthetic data is crucial as it provides ground truth angular measurements with known accuracy, eliminating visual ambiguity or occlusion that might confound assessment. Meanwhile, real-world images introduce natural complexity and diversity while maintaining clear ground truth through expert annotation. Additional statistics on fine-grained categories across tasks are in App. D, broad categories in App. E, and data composition in App. F. See App. I for examples.

### 4 Experiments

We evaluate 22 state-of-the-art multimodal models spanning diverse architectures, parameter scales, and pretraining methodologies across both open-source and proprietary systems. The models are as follows: LLaVA-v1.6-13B (Liu et al., 2024a), LLaVA-v1.6-34B (Liu et al., 2024a), LLava-Next-

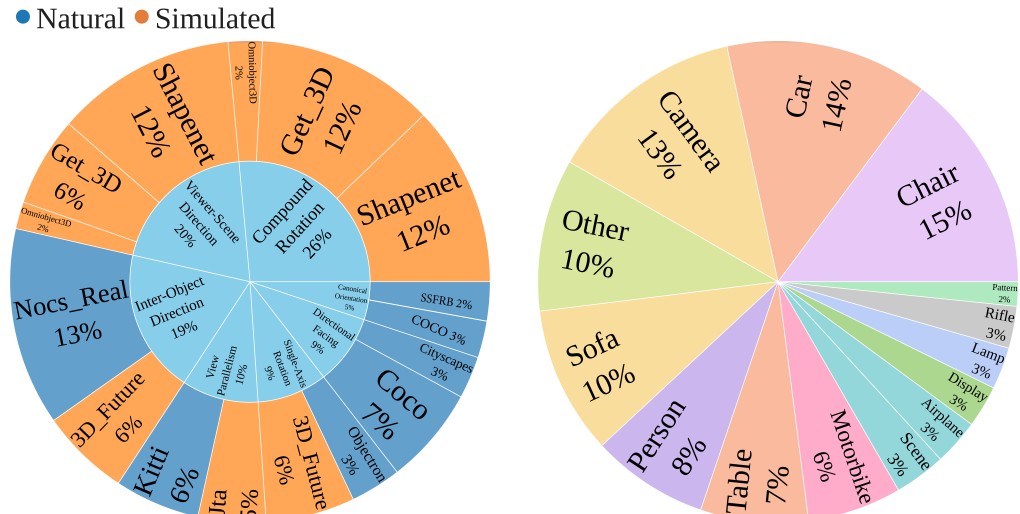

Figure 3: (a) The DORI benchmark embodies **seven distinct orientation tasks** (inner circle) with a balanced and systematic distribution of samples across both **natural and simulated images**. (b) DORI captures diverse objects commonly encountered in day-to-day life, supporting a comprehensive analysis of models' orientation understanding capabilities.

8B (Liu et al., 2024b), Yi-VL-6B (AI et al., 2024), Yi-VL-34B (AI et al., 2024), Mantis-CLIP (Jiang et al., 2024), Mantis-Idfs-8B (Jiang et al., 2024), DS-1.3B-Base (Bi et al., 2024), DS-1.3B-Chat (Bi et al., 2024), DS-7B-Base (Bi et al., 2024), DS-7B-Chat (Bi et al., 2024), Qwen2.5-3B-Inst. (Team, 2024), Qwen3-8B-Inst. (Yang et al., 2025a), Qwen32-32B-Inst. (Yang et al., 2025a), Magma-8B (Yang et al., 2025b), RP-v1-13B (Yuan et al., 2025), IntVL-14B-Inst (Wang et al., 2025a), and IntVL-30B-A3B-Inst. (Wang et al., 2025a). We evaluate proprietary models via their official APIs and most open-source models on a single NVIDIA RTX A6000 GPU, with 34B parameter models being evaluated on a single NVIDIA A100-80G. We use answer accuracy across both coarse and granular question types to measure performance.

## 4.1 RESULTS

Tab. 2 reports open-source MLLM's accuracy on DORI. Notably, many models obtain around random performance. As expected, more models struggle with challenging granular rather than coarse questions. This indicates that current models cannot perform precise angular reasoning, instead relying on coarse categorical judgments, even for models trained for robotics like Magma-8B and RP-V1-13B. Larger models of each model family generally perform better on average, particularly on coarse questions. Closed-source models in Tab. 3 obtain 9-15% better performance than the best open-source models. However, all closed-source models continue to struggle, especially on rotational transformation and relative orientation questions, highlighting substantial room for improvement even in state-of-the-art commercial systems. See App. L for a detailed error analysis.

We hypothesize the poor performance stems from how MLLMs are pretrained - most models being evaluated use CLIP-style contrastive objectives that optimize for high-level image-text semantic alignment rather than core geometric understanding (Zhong et al., 2022). Tong et al. (2024b) similarly noted that pretraining creates a "dimensional collapse" in the embedding space, where continuous orientation variations become compressed into discrete semantic clusters (*e.g.*, treating "left" and "right" as opposing categorical concepts rather than points along a continuous angular spectrum). This explains why models can often distinguish between categorical extremes but fail on tasks that require precise angular discrimination. While this may be slightly mitigated by using generative objectives (Li et al., 2025), MLLMs lack the necessary equivalent neural inductive biases utilized by humans (Ramakrishnan et al., 2025; Ling et al., 2009; Delhaye et al., 2018; Goble et al., 2012). Instead, MLLMs approximate neural mechanisms through suboptimal attention patterns leading to hallucinations (Huang et al., 2024; Yamada et al., 2024; Deng et al., 2020; Yang et al., 2024).

Table 2: **Performance of several open-source MLLMs on DORI**. Most models perform poorly, particularly on granular questions. These experiments reveal a systemic gap in object orientation understanding across all four dimensions studied in DORI. See App. K for results with model variance. C: coarse, G: granular.

| | Frontal Alignment | | | | Rotational Transformation | | | | Relative Orient. | | | | Canonical Orient. | | | |
| | View Parallel. | | Dir. Facing | | Single-axis Rot. | | Compound Rot. | | Inter-Obj. Dir. | | Viewer-scene Dir. | | | | Avg. | |
| | C | G | C | G | C | G | C | G | C | G | C | G | C | G | C | G |
|---|---|---|---|---|---|---|---|---|---|---|---|---|---|---|---|---|
| Random | 35.7 | 25.5 | 20.2 | 19.8 | 20.7 | 17.4 | 20.4 | 6.6 | 15.0 | 10.1 | 33.1 | 20.3 | 34.8 | 16.0 | 25.7 | 16.5 |
| LLaVA-v1.6-13B | 55.9 | 26.8 | 22.6 | 19.8 | 23.4 | 16.5 | 32.8 | **10.6** | 7.2 | 12.4 | 46.6 | 20.1 | 16.7 | 11.3 | 22.1 | 16.7 |
| LLaVA-v1.6-34B | 52.8 | 35.3 | **32.6** | **26.4** | 22.1 | 26.2 | 13.0 | 4.3 | 16.2 | 14.8 | 34.8 | 25.4 | 9.9 | 11.6 | 25.9 | 20.5 |
| LLava-Next-8B | 33.2 | 25.2 | 21.3 | 21.0 | 20.5 | 17.9 | **40.6** | 6.3 | 10.3 | 12.0 | 61.5 | 25.2 | 10.5 | 9.2 | 28.3 | 14.4 |
| Yi-VL-6B | 38.9 | 31.0 | 25.0 | 25.2 | **28.5** | 14.7 | 29.6 | 3.0 | 15.2 | 12.3 | 41.4 | 10.8 | 30.3 | 14.4 | 29.8 | 15.9 |
| Yi-VL-34B | 53.1 | **35.1** | 23.3 | 24.0 | 28.1 | 21.2 | 32.5 | 4.4 | 13.4 | 14.5 | 61.1 | 19.7 | 11.3 | 12.5 | 31.8 | 18.7 |
| Mantis-CLIP | **60.1** | 24.3 | 26.1 | 16.3 | 15.4 | 20.9 | 9.1 | 5.8 | 13.2 | 10.1 | 37.7 | 17.9 | 23.7 | **34.5** | 23.8 | 15.0 |
| Mantis-Idfs-8B | 57.8 | 33.0 | 22.5 | 12.7 | 25.7 | **23.4** | 25.9 | 6.6 | 17.6 | 9.0 | 55.4 | 24.5 | **48.8** | 41.0 | 34.5 | 17.5 |
| DS-1.3B-Base | 58.1 | 24.8 | 15.0 | 19.4 | 21.1 | 15.2 | 17.0 | 5.3 | 17.0 | 11.9 | 61.3 | 26.0 | 2.5 | 2.3 | 29.3 | 14.7 |
| DS-1.3B-Chat | 58.1 | 24.8 | 22.6 | 22.0 | 21.2 | 15.5 | 33.9 | 5.8 | 15.3 | 10.5 | 47.6 | 18.0 | 29.2 | 17.3 | 33.0 | 14.1 |
| DS-7B-Base | 26.3 | 31.1 | 14.6 | 16.4 | 18.3 | 14.3 | 35.5 | 2.8 | 3.6 | 4.1 | 35.7 | 6.8 | 31.3 | 10.7 | 24.7 | 9.6 |
| DS-7B-Chat | 59.4 | 31.3 | 31.3 | 24.5 | 28.2 | 17.8 | 35.8 | 6.0 | **20.2** | **14.9** | 18.5 | **32.2** | 15.2 | 32.2 | 29.4 | 18.6 |
| Qwen2.5-3B-Inst. | 56.3 | 29.3 | 23.4 | 3.1 | 18.2 | 17.7 | 25.4 | 0.21 | 16.3 | 13.8 | **62.8** | 16.9 | 7.5 | 11.4 | 29.9 | 13.2 |
| Qwen3-8B-Inst. | 73.7 | 45.6 | 50.8 | 54.8 | 32.9 | 45.9 | 49.0 | 7.5 | 18.2 | 21.9 | 79.1 | 37.2 | 6.4 | 21.1 | **44.3** | 33.4 |
| Qwen3-32B-Inst | 75.8 | 54.4 | 61.1 | 59.2 | 31.6 | 44.1 | 39.6 | 8.6 | 9.2 | 20.5 | 78.9 | 46.1 | 7.0 | 25.5 | 43.3 | **36.9** |
| Magma-8B | 51.5 | 21.4 | 25.4 | 20.9 | 20.6 | 18.7 | 32.0 | 5.4 | 16.3 | 11.5 | 32.6 | 21.0 | 15.6 | 16.1 | 27.7 | 16.4 |
| RP-v1-13B | 45.7 | 20.5 | 24.7 | 22.3 | 24.9 | 14.7 | 37.6 | 5.6 | 12.9 | 12.7 | 61.2 | 20.9 | 11.2 | 10.6 | 31.1 | 15.3 |
| IntVL-14B-Inst. | 63.4 | 37.2 | 37.9 | 48.6 | 30.3 | 21.4 | 26.4 | 7.2 | 4.3 | 15.0 | 94.3 | 44.1 | 7.0 | 16.0 | 37.7 | 27.1 |
| IntVL-30B-A3B-Inst. | 72.1 | 33.9 | 46.6 | 50.0 | 17.4 | 17.7 | 33.3 | 6.8 | 19.3 | 10.0 | 90.1 | 35.0 | 8.3 | 22.2 | 41.0 | 25.1 |

Table 3: **Performance of proprietary vs. open-source MLLMs** on 100 randomly sampled questions across all four dimensions and granularities. Commercial models also struggle with complex orientation tasks, particularly with angular precision and reference shifts. C: coarse, G: granular.

| | Frontal Alignment | | | | Rotational Transformation | | | | Relative Orient. | | | | Canonical Orient. | | | |
| | View Parallel. | | Dir. Facing | | Single-axis Rot. | | Compound Rot. | | Inter-Obj. Dir. | | Viewer-scene Dir. | | | | Avg. | |
| | C | G | C | G | C | G | C | G | C | G | C | G | C | G | C | G |
|---|---|---|---|---|---|---|---|---|---|---|---|---|---|---|---|---|
| LLaVA-v1.6-34B | 45.5 | 30.0 | 22.5 | 23.5 | 15.0 | 15.5 | 19.3 | 7.6 | 15.0 | 10.5 | 37.3 | 34.3 | 17.0 | 19.0 | 24.5 | 20.0 |
| LLava-Next-8B | 34.0 | 13.0 | 18.0 | 19.0 | 12.5 | 17.5 | 40.3 | 7.0 | 12.0 | 13.5 | 64.0 | 25.0 | 17.0 | 16.0 | 28.2 | 15.8 |
| DS-7B-Chat | 52.5 | 28.0 | 24.5 | 24.5 | 21.0 | 15.0 | 35.3 | 5.0 | 24.0 | 14.0 | 23.6 | 32.0 | 16.0 | 2.0 | 28.1 | 19.7 |
| Qwen3-8B-Inst. | 61.5 | 41.0 | 38.5 | 36.0 | 23.5 | 37.0 | 69.6 | 9.3 | 17.5 | 13 | 88.3 | 47 | 23.0 | 26.0 | 45.9 | 29.9 |
| Qwen3-32B-Inst | 60.0 | 39.5 | 37.5 | 33.5 | 20.5 | 37.5 | 68.3 | 10.3 | 16.5 | 13.5 | 86.6 | 44.6 | 20.0 | 26.0 | 44.2 | 29.2 |
| IntVL-14B-Inst. | 61.5 | 32.0 | 32.5 | 42.5 | 29.0 | 16.0 | 31.3 | 8.3 | 6.5 | 14.0 | 93.3 | 45.6 | 14.0 | 25.0 | 38.3 | 26.2 |
| IntVL-30B-A3B-Inst. | 67.5 | 34.0 | 40.5 | 39.0 | 18.0 | 11.0 | 35.0 | 8.0 | 26.0 | 13.0 | 89.0 | 33.6 | 1.0 | 33.0 | 40.8 | 24.5 |
| Gemini 1.5 Pro | **68.5** | **53.0** | **43.5** | **39.0** | 24.0 | 37.5 | **65.3** | 12.3 | **25.0** | 14.5 | 91.0 | **47.3** | **22.0** | 28.0 | **54.2** | 33.0 |
| Gemini 2.0 Flash | 67.0 | 35.0 | 37.5 | 33.5 | 28.5 | 27.0 | 52.0 | 14.3 | 23.0 | **17.0** | **92.0** | 43.6 | 5.0 | 29.0 | 49.9 | 28.5 |
| GPT-4o | 44.5 | 45.0 | 30.5 | 35.5 | 29.5 | 34.0 | 39.3 | **15.3** | 23.5 | 13.5 | 88.3 | 53.3 | 19.0 | 45.0 | 19.0 | **45.0** |
| GPT-4-1 | 48.5 | 41.0 | 39.0 | **40.5** | **32.5** | **42.0** | 31.6 | 14.0 | 22.5 | 8.5 | 87.6 | 44.6 | 20.0 | **46.0** | 40.2 | 33.8 |

**Architecture impact.** Among open-source systems in Tab. 2, a clear architectural pattern emerges: token-based approaches like Mantis-Idefics2-8B outperform linear projection methods, suggesting that token-level integration preserves richer dimensional information. Notably, DeepSeek-1.3B-Chat (33.0% average accuracy across all coarse tasks) significantly outperforms DeepSeek-7B-Base (24.7% average coarse accuracy) despite having fewer parameters. We also find dialogue-tuned variants consistently outperform base counterparts (*e.g.*, DeepSeek-7B-Chat vs. DeepSeek-7B-Base), suggesting that instruction tuning, which emphasizes structured, logical reasoning—enhances models' ability to

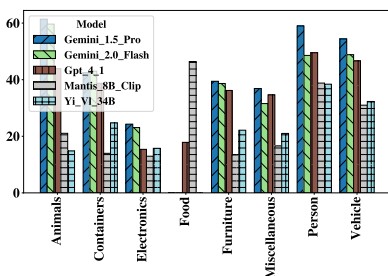
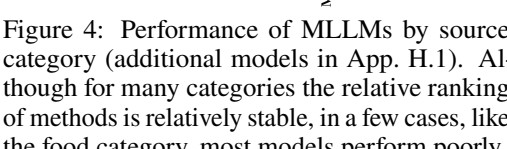
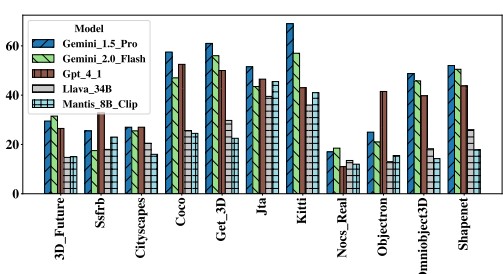

Figure 4: Performance of MLLMs by source category (additional models in App. H.1). Although for many categories the relative ranking of methods is relatively stable, in a few cases, like the food category, most models perform poorly.

Figure 5: Performance of MLLMs by source dataset (additional models in App. H.2). Models perform better on simulated datasets and coarse questions, revealing a gap in generalizing to natural images and more fine-grained questions.

Table 4: Comparison of a base Qwen2.5-VL-3B to a version tuned on DORI with LoRA. We report zero-shot transfer to three datasets: 3DSRBench (Ma et al., 2025), Blink (Fu et al., 2024a), and SAT (Ray et al., 2025). Finetuning on our dataset reports up to a 27% gain on other benchmarks.

| | Blink | | | 3DSRBench | | | | SAT |
|---|---|---|---|---|---|---|---|---|
| | Multi-View Reasoning | Visual-Corresp. | Relative-Depth | Orient. | Multi-Obj. View To. Obj. | Multi-Obj. Parallel | Multi-Obj. Same Dir. | |
| Base Model | 42.9 | 25.3 | 64.1 | 32.5 | 11.4 | 11.4 | 46.8 | 51.3 |
| +Finetuning | **45.1** | **26.5** | **65.3** | **38.6** | **14.0** | **38.1** | **50.3** | **63.3** |

follow multi-step orientation reasoning instructions and produce coherent judgments (Ranasinghe et al., 2024). The substantial gap between Qwen3 models and previous iterations (*e.g*., a 14% gain on coarse questions between Qwen2.5-3B and Qwen3-8B) highlights that recent architectural improvements, including enhanced vision-language alignment and more sophisticated multimodal fusion, significantly benefit orientation understanding. These results suggests that improving orientation reasoning requires a careful consideration of *how* to combine information from multiple modalities, instead of just scaling models. App. H.3 reports an experiment where we replace the fusion strategy of a pretrained Qwen model, which also supports our observations. However, as there are additional confounding factors such as the pretraining data and strategy used by each MLLM. Thus, a fully controlled experiment would be required to validate these observations.

**Performance across data sources and object categories.** Fig. 4 reveals that models generally perform better on orientation tasks involving people and animals, presumably because these categories have a clear front/back distinctions through faces compared to more ambiguous objects like furniture or containers. This pattern may also suggest that current models rely heavily on semantic features (*e.g*., recognizing faces) rather than a fundamental geometric understanding when determining object orientation. Fig. 5 provides a performance breakdown by source, where we find that models generally struggle the most on natural images with complex scenes. However, the models do generally well on some datasets, like COCO, but this also raises a potential risk that these large models with privately held models may have used some of these samples during pretraining. To address this concern, in App. N we used the approach from Teterwak et al. (2025) to separate into (likely) in-pretraining and out-of-pretraining sets, and find that models generally perform similarly on both sets. Thus, the effect of any potential contamination on our dataset is negligible.

**Generalization performance.** Tab. 4 reports the effect of finetuning Qwen2.5-VL-3B on 27K real + synthetic samples from our dataset using LoRA (Hu et al., 2022) on SAT (Ray et al., 2025), 3DSRBench (Ma et al., 2025), and BLINK (Fu et al., 2024a). We find that training on DORI boosts performance by up to 27%, demonstrating the benefit of our dataset. In App. M we also report that this results in a 37-46% boost over the base model on 7K held-out DORI samples.

**Human evaluation.** We recruited 14 experts with experience in complex annotation procedures to assess orientation perception abilities. Each participant evaluated 50 examples for both coarse and

Table 5: Comparison of **Human performance vs. MLLMs** on 50 randomly sampled questions of each type. We find a significant performance gap exists between our expert annotators and MLLMs

| | Frontal Alignment | | | | Rotational Transformation | | | | Relative Orient. | | | | Canonical Orient. | | Avg. | |
| | View Parallel. | | Dir. Facing | | Single-axis Rot. | | Compo-und Rot. | | Inter-Obj. Dir. | | Viewer-scene Dir. | | | | | |
| | C | G | C | G | C | G | C | G | C | G | C | G | C | G | C | G |
|---|---|---|---|---|---|---|---|---|---|---|---|---|---|---|---|---|
| GPT-4o | 22.0 | 36.0 | 38.0 | 6.0 | 36.0 | 34.0 | 76.0 | 66.6 | 38.0 | 48.0 | 88.0 | 70.6 | 20.0 | 20.0 | 45.4 | 40.1 |
| Gemini 1.5 Pro | 64.0 | 72.0 | 76.0 | 80.0 | 56.0 | 54.0 | 86.0 | 66.0 | 64.0 | 62.0 | 98.0 | 80.0 | 28.0 | 64.0 | 67.4 | 68.2 |
| Human | 82.0 | 92.0 | 92.0 | 82.0 | 80.0 | 78.0 | 86.0 | 86.0 | 86.0 | 74.0 | 98.0 | 90.0 | 92.0 | 92.0 | 88.0 | 84.8 |

granular tasks using identical images (resulting in 14 question types $\times$ 50 = 700 total samples). Each question contained three answer options (the correct answer, "Cannot be determined," and a randomly sampled incorrect answer). See App. O for example instructions. Tab. 5 shows that humans achieved 85-88% accuracy, demonstrating that they largely agreed with our labels, with the best closed-source model around 20% lower. This highlights both the quality of our annotations and the amount of potential growth of MLLMs.

## 5 Conclusion

DORI reveals critical limitations in current MLLMs' orientation understanding capabilities, with even state-of-the-art models achieving only 54.2% accuracy on coarse tasks and 45.0% on granular orientation judgments and perform significantly below human performance. Our comprehensive evaluation demonstrates that MLLMs systematically falter when tasks require precise angular estimations, multi-axis rotational transformations, or perspective shifts beyond egocentric frames. These deficiencies likely stem from fundamental architectural constraints that compress continuous geometric information into discrete semantic categories rather than developing true geometric representations. As the first diagnostic framework specifically targeting orientation awareness, DORI provides clear metrics for measuring progress and establishes specific pathways for advancing multimodal systems toward the robust orientation reasoning capabilities essential for real-world applications in robotics, autonomous navigation, augmented reality, and embodied AI. The results strongly suggest that future architectures must develop specialized mechanisms for continuous geometric representation to bridge this critical gap in machine perception.

## Ethics Statement

While our work on evaluating orientation reasoning in multimodal large language models aims to advance AI capabilities in spatial understanding, we acknowledge several ethical considerations. First, our human evaluation component involved recruiting annotators to assess orientation perception abilities, and we ensured informed consent and fair compensation for their participation. The benchmark's applications in robotics, autonomous navigation, and augmented reality could significantly benefit society by improving human-AI interaction and enabling more capable assistive technologies. However, we recognize that enhanced spatial reasoning capabilities could potentially be misused in surveillance applications that infringe on privacy rights or in autonomous systems that make consequential decisions without adequate human oversight. Additionally, our evaluation reveals systematic biases in current models' performance across different object categories and visual contexts, which could perpetuate unfair outcomes if deployed without consideration of these limitations. The benchmark combines real-world and synthetic data from multiple computer vision datasets, and we have ensured proper licensing and ethical use of all source materials. We emphasize the importance of responsible development and deployment of spatially-aware AI systems, particularly in safety-critical applications, and encourage ongoing dialogue about the ethical implications of advancing machine spatial reasoning capabilities.

## REPRODUCIBILITY

To ensure the reproducibility of our work, we have taken comprehensive steps to document all aspects of our experimental methodology. The complete implementation of DORI, including the systematic question generation process, structured prompt design, and evaluation metrics, is described in detail in the main text and appendix. We provide full specifications of our annotation methodology, including the human evaluation protocols used with expert annotators, the soft accuracy calculation methods, and the systematic error analysis framework. All hyperparameters, model configurations, and evaluation procedures are explicitly documented, with specific details about the prompting strategies used for each of the 22 evaluated models. The benchmark construction process, including the conversion of 3D pose information to orientation questions and the manual annotation procedures for datasets lacking orientation metadata, is thoroughly described with examples provided in the appendix. We include comprehensive statistics about dataset composition, object category distributions, and the mapping between fine-grained object labels and broader semantic classes. While some evaluated models are proprietary, we provide detailed API specifications and prompting protocols to enable consistent evaluation. Our error analysis methodology, including the systematic failure pattern identification and component-wise error decomposition for compound rotations, is fully specified with clear criteria and thresholds. All preprocessing steps, evaluation scripts, and analysis code will be made available to facilitate accurate replication of our results and enable future research building upon this benchmark.

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

# Appendix

## Table of Contents

Table 6: Developmental progression and neural substrates underlying DORI's four orientation reasoning dimensions

| Dimension | Cognitive Stage | Neural Substrate | Age (yrs) | Aspect Tested |
|---|---|---|---|---|
| Frontal Alignment | Basic Perception | Visual cortex (V1/V2) (Otten et al., 2025) | ~1 | Directional Perception |
| Rotational Transformation | Mental Rotation | Premotor/Parietal Cortex (Frick et al., 2014) | ~4-5 | Spatial Transformation |
| Relative Orientation | Allocentric Reasoning | Hippocampus/Parietal (Pullano & Foti, 2022) | ~6-7 | Multi-object Relationships |
| Canonical Orientation | Semantic Reasoning | Inferotemporal/Prefrontal (Dumontheil, 2014) | ~7-9 | Semantic Properties |

## A  Cognitive Neuroscience Foundations of DORI Dimensions

Based on established frameworks in cognitive neuroscience, DORI decomposes object orientation comprehension into four fundamental dimensions that reflect distinct neural mechanisms and cognitive processes.

**Frontal Alignment:** Cognitive studies confirm that V1/V2 directional perception precedes complex spatial reasoning (Otten et al., 2025). Infants' visual cortex development enables directional perception necessary for grasping and object manipulation (e.g., identifying "Which way does the sensor point?"), which maps to our Frontal Alignment dimension.

**Rotational Transformation:** Research on mental rotation shows that at ages 4-5, premotor and parietal cortex support mentally imagining objects transforming across axes (Frick et al., 2014). Our Rotational Transformation dimension evaluates this capability in models (e.g., "Rotate 90° horizontal + 45° vertical").

**Relative Orientation:** By ages 6-7, hippocampal-parietal networks enable allocentric orientation reasoning—understanding spatial relationships relative to external frames, not just egocentric viewpoints (Pullano & Foti, 2022). DORI tests this reasoning about objects' orientation relative to each other (e.g., "Are chairs facing each other around a table?").

**Canonical Orientation:** This dimension involves semantic reasoning about objects' "correct" or functional orientations (e.g., "Which way is the bowl right-side-up?"), requiring abstract reasoning (Dumontheil, 2014). This capability is supported by prefrontal and inferotemporal cortex development in children ages 7-9 years.

This categorization is both complete and necessary. The four dimensions span the full developmental arc (1 year to 7-9 years) across all major neural systems. Each dimension is irreducible—removing any creates evaluation gaps. The progression follows a strict hierarchy: frontal identification (perception) → transformation (manipulation) → relational (coordination) → semantic (understanding), ensuring comprehensive assessment of orientation reasoning capabilities. We summarize this discussion in Tab. 6.

## B  Task-wise Dataset Creation

**View parallelism perception** task evaluates a model's ability to determine whether an object's front-facing surface is oriented toward, away from, or perpendicular to the camera plane. We constructed this dataset using images from the JTA (Fabbri et al., 2018) and KITTI(Geiger et al., 2012) datasets (specifically the subset used in OmniNOCS (Krishnan et al., 2024)). For JTA, which contains 3D human pose annotations, we calculated orientation by analyzing shoulder positions relative to the camera and head angle to precisely determine facing direction. For KITTI, we leveraged the available rotation matrices to categorize vehicles and pedestrians based on their orientation relative to the camera. This task is critical for fundamental scene understanding, where determining which objects are facing an agent is essential for interaction and navigation decisions.

**Frontal alignment perception** task extends orientation assessment to cardinal directions, requiring models to identify if objects are facing toward, away, leftward, or rightward relative to the camera. We developed this dataset using images from COCO (Lin et al., 2014) and Cityscapes (Cordts et al., 2016) (from the OmniNOCS (Krishnan et al., 2024) subset). For COCO images, which lack orientation annotations, we conducted expert manual labeling of object orientations. For Cityscapes, we utilized rotational matrices to precisely determine directional orientation, limiting images to contain at most three objects to ensure assessment clarity. This directional understanding is vital for spatial navigation and object manipulation tasks where agents must understand not just if objects face them, but their specific directional orientation.

**Single-axis rotation** task assesses understanding of rotational transformations around a vertical axis by asking models to determine the optimal rotation direction and precise angular adjustment needed for objects to face the camera. We constructed this dataset using 3D-Future (Fu et al., 2021), which provides high-resolution 3D furniture models with known 6-DoF parameters. We focused primarily on chair variants with distinctive front/back features, calculating the exact rotational adjustment needed for the object to face the camera directly. This capability forms the foundation for computational manipulation planning and scene reconfiguration understanding. Furthermore, we utilized the Objectron (Ahmadyan et al., 2021) subset of the OmniNOCS (Krishnan et al., 2024) dataset. This also included 6 DoF information which we used to determine the angle of the object with respect to the camera, for this dataset we utilized bikes, chairs and bottles.

**Compound rotation** task evaluates comprehension of complex rotations involving sequential transformations around multiple axes, where the rotation order impacts the final orientation. We developed this dataset using 3D-rendered objects from Get3D (Gao et al., 2022), ShapeNet (Guibas, 2017), and OmniObject3D, (Wu et al., 2023) implementing a controlled rendering pipeline in Blender. For each object, we rendered an initial third-person view, then applied precise rotations along horizontal and vertical axes in varying sequences, rendering the transformed state. This task assesses the sophisticated mental rotation capabilities required for complex object manipulation and orientation reasoning across multiple dimensions.

**Inter-object direction perception** task evaluates understanding of relative orientation between multiple objects from their own perspectives rather than the camera view. Using the 3D Future (Fu et al., 2021) and NOCS REAL (Wang et al., 2019) datasets, we leveraged 6 DoF parameters to calculate precise angular relationships between object pairs. The task requires models to determine if objects face the same direction, opposite directions, or have perpendicular orientations, progressing to granular assessment of the exact rotation needed for objects to align. This capability is essential for understanding agent-object and object-object relationships in complex scenes, particularly for collaborative robotic tasks or scene arrangement planning.

**Viewer-scene direction perception** task evaluates perception of rotational changes between two images of the same object. Using Get3D (Gao et al., 2022), ShapeNet (Guibas, 2017), and OmniObject3D (Wu et al., 2023) datasets, we rendered objects with a ground plane reference, then created corresponding images with the object rotated by specific angles around a vertical axis. Models must determine whether rotation occurred and, at the granular level, specify the exact degree of rotation. This assessment examines the ability to track orientation changes across views—a crucial capability for video understanding, temporal reasoning, and object tracking applications.

**Canonical orientation reasoning** task evaluates models' understanding of normal object orientations and their ability to identify when objects appear in non-canonical positions. Using a subset of COCO images with clear orientation expectations (e.g., people standing upright, vehicles with wheels on the ground), we created variations with systematic flips and rotations. Models must first identify whether images appear in their canonical orientation, then determine the specific transformations (rotation, flipping, or both) needed to restore proper orientation. This capability assesses world knowledge about typical object positioning, which is critical for anomaly detection, image correction, and understanding intentional vs. unintentional orientation deviations.

Table 7: Models perform much worse for most of the tasks when the prompts are unstructured and have no clarification examples. The degradation is particularly prominent for Compound rotation and Canonical orientation related tasks

| | Frontal Alignment | | | | Rotational Transformation | | | | Relative Orient. | | | | Canonical Orient. | | Avg. | |
| | View Parallel. | | Dir. Facing | | Single-axis Rot. | | Compound Rot. | | Inter-Obj. Dir. | | Viewer-scene Dir. | | | | | |
| | C | G | C | G | C | G | C | G | C | G | C | G | C | G | C | G |
|---|---|---|---|---|---|---|---|---|---|---|---|---|---|---|---|---|
| LLaVA-v1.6-13B | 57.9 | 33.0 | 25.4 | 0.01 | 20.2 | 16.5 | 27.7 | 0.01 | 16.7 | 12.2 | 60.4 | 19.8 | 0.01 | 0.0 | 29.7 | 11.6 |
| Yi-VL-6B | 52.4 | 37.8 | 23.9 | 20.3 | 36.8 | 15.9 | 19.7 | 21.6 | 11.0 | 16.5 | 78.6 | 18.6 | 29.1 | 12.9 | 35.9 | 20.5 |
| Mantis-CLIP | 40.5 | 9.1 | 6.9 | 6.8 | 6.2 | 2.6 | 0.9 | 1.8 | 0.2 | 0.0 | 57.9 | 1.9 | 36.3 | 46.7 | 22.4 | 6.7 |
| Mantis-If2-8B | 51.2 | 29.9 | 29.7 | 13.6 | 15.3 | 10.5 | 62.5 | 5.0 | 5.2 | 18.4 | 85.9 | 32.1 | 45.9 | 46.1 | 53.4 | 19.9 |
| LLava_next 8B | 56.9 | 1.8 | 25.5 | 29.9 | 22.2 | 11.7 | 34.1 | 6.6 | 24.3 | 10.0 | 66.0 | 26.4 | 11.3 | 11.0 | 34.3 | 13.9 |
| DS-1.3B-Chat | 40.7 | 30.2 | 22.6 | 21.8 | 12.2 | 19.8 | 35.6 | 5.6 | 16.2 | 14.2 | 31.1 | 21.6 | 35.2 | 9.8 | 27.6 | 17.5 |
| DS-7B-Base | 45.2 | 35.0 | 21.6 | 21.2 | 25.4 | 18.0 | 2.8 | 5.7 | 15.0 | 11.7 | 33.6 | 25.7 | 3.7 | 16.5 | 21.0 | 19.1 |
| DS-7B-Chat | 48.1 | 32.4 | 27.9 | 25.7 | 47.0 | 19.5 | 1.0 | 5.1 | 49.6 | 15.8 | 36.6 | 22.7 | 32.9 | 10.7 | 34.7 | 18.8 |

## C  CONFOUNDING FACTORS & PROMPT ITERATION

In our dataset we have carefully designed our questions for both the prompt and image to reduce the amount of different confounding factors that are present in past datasets. We include aspects like: defining axes, defining what the canonical view is, and we direct the readers to Appendix I& J.2 which showcases these elements present in our questions. We can see in examples like Figure 16 we utilize a bounding box which helps the model/rater determine which object to focus on; this helps reduce object recognition difficulty. For scene clutter we can look at Figure 26 which focuses on particular objects without any objects being present in the image. Figure 18 showcases an example of reducing linguistic ambiguity, by describing what the MLLM should look at and identify its front facing surface, additionally objects that have ambiguous front facing surface like a table we have labelled those as "Cannot be determined". For contextual distractions we can also look to Figure 21 which contains backgrounds that do not influence the exact orientation of the object.

To examine the effect of prompt engineering on orientation understanding, we conducted ablation studies on 8 models by removing key prompt components: structural formatting, answer examples, detailed task conceptualizations, and step-by-step reasoning instructions (Fig. 3.a). The ablation results (Tab. 7) reveal substantial performance degradation compared to structured prompts (Tab. 2), with the magnitude and pattern of decline varying significantly across task types and model architectures.

Most models exhibit severe degradation with unstructured prompts. LLaVA-v1.6-13B experiences near-complete collapse on Direction Facing granular (99.9% relative decline), Compound Rotation granular (99.9% relative decline), and Canonical Orientation (100% relative decline). LLava-Next-8B shows similarly catastrophic degradation on View Parallelism granular (93% relative decline). These patterns indicate that multi-step spatial reasoning becomes effectively unsolvable without explicit scaffolding, with structured prompts providing 50-100× performance improvements on fine-grained orientation tasks for these architectures. However, prompt sensitivity varies dramatically across model families. Mantis-Idfics-8B demonstrates counterintuitive improvements under unstructured prompts, with overall coarse performance improving 55% relative and Compound Rotation coarse surging 141% relative. This suggests that structured prompts may impose reasoning templates incompatible with Mantis's interleaved vision-language architecture, potentially constraining its native multi-image spatial reasoning mechanisms. DS-7B-Chat exhibits similar non-monotonic behavior: Single-axis Rotation coarse improves 67% relative and Inter-Object Direction coarse surges 145% relative under ablation, yet Compound Rotation coarse collapses 97% relative. These divergent patterns indicate that structured prompts successfully disambiguate certain spatial transformations while introducing interference on others, with the effect highly dependent on architectural priors and task complexity. Canonical Orientation remains challenging regardless of prompt structure across all models, with minimal performance differences between conditions. This persistence suggests that prompt engineering alone cannot compensate for fundamental gaps in geometric priors and

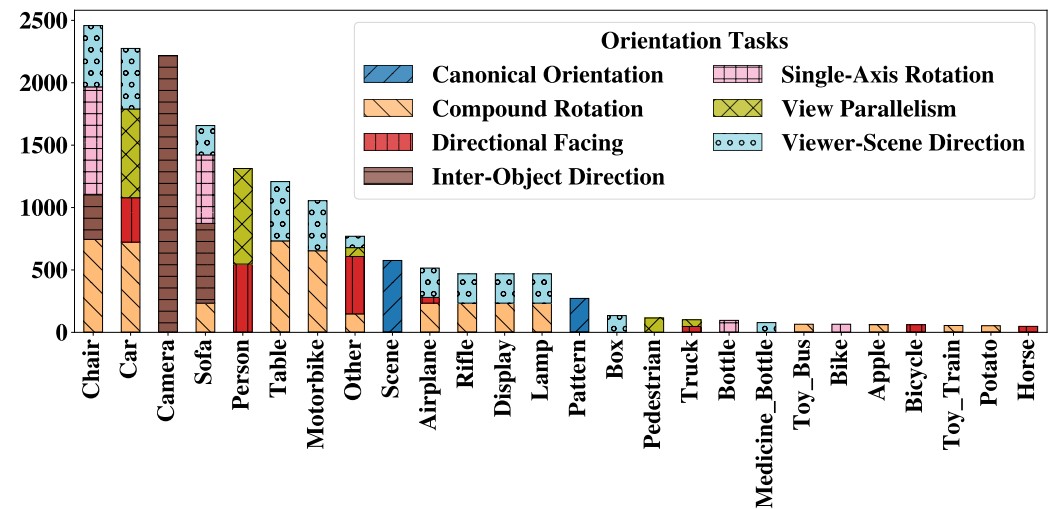

Figure 6: Sample category distribution of top 25 object categories showing their associated orientation tasks

world knowledge—these capabilities likely require architectural innovations or specialized pretraining objectives rather than improved task specification.

# D  CATEGORY DISTRIBUTION ACROSS ORIENTATION TASKS

Fig. 6 presents the distribution of the top 25 most frequent object categories in DORI, annotated with their corresponding orientation task types. This visualization highlights not only which categories are most prevalent, but also how they are utilized across the various tasks in DORI.

While commonly occurring objects such as *chair*, *car*, and *sofa* span several task types, others are more narrowly concentrated. This overlap and separation across tasks reflect the intentional diversity of DORI, designed to evaluate model capabilities across both general-purpose and category-specific orientation challenges. The distribution also underscores the need for models to generalize effectively across unevenly represented categories and task combinations.

# E  MAPPING OBJECT CATEGORIES TO BROAD CLASSES

To enable category-level analysis, we group the fine-grained object labels in DORI into a smaller set of semantically meaningful broad classes. This abstraction facilitates more interpretable evaluation across heterogeneous datasets and reduces sparsity in underrepresented categories. The mapping spans common categories such as `Vehicle`, `Person`, `Animals`, `Food`, `Containers`, `Furniture`, `Electronics`, and `Miscellaneous`, and is illustrated in Tab. 8 and Fig. 7.

Table 8: Mapping from object categories to broad semantic classes.

| Object Category | Broad Class |
|---|---|
| car, truck, tram, van, bus, train, airplane, boat, | Vehicle |
| motorcycle, bicycle, coach, bike, motorbike, trailer | Vehicle |
| pedestrian, cyclist, person | Person |
| zebra, dog, elephant, giraffe, cat, horse, sheep, | Animals |
| bird, cow, bear, starfish | Animals |
| pizza, broccoli, donut, orange, apple, tomato, potato | Food |
| cake, egg, wine glass, bowl, bottle, onion | Food |
| bottle, cup, wine glass, bowl, medicine_bottle, box | Containers |
| chair, sofa, table, lamp, bench, sofa-chair, soft-sofa | Furniture |
| laptop, tv, display, camera, laptop-camera | Electronics |
| clock, vase, toilet, umbrella, teddy bear, rifle | Miscellaneous |
| stop sign, toy_train, toy_bus, house, ball | Miscellaneous |
| scene, pattern | Miscellaneous |

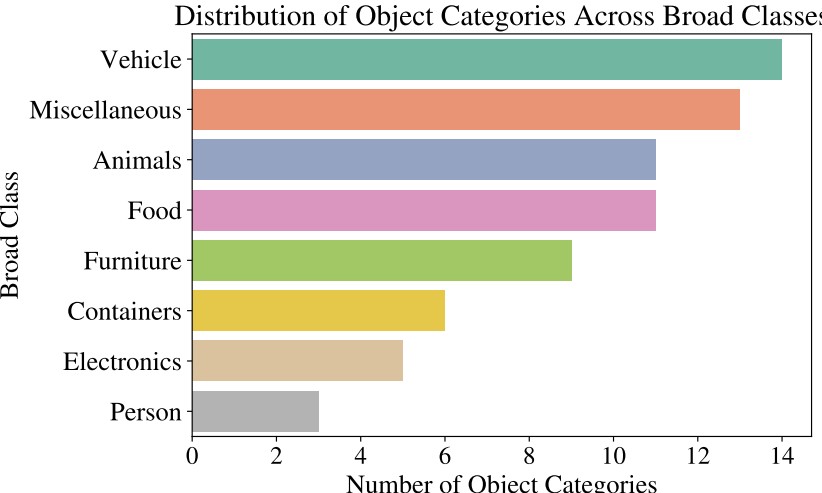

Figure 7: Distribution of fine-grained object categories mapped to broader semantic classes. Most categories fall under *Vehicle*, *Miscellaneous*, and *Animals*, reflecting common object types in the evaluated datasets.

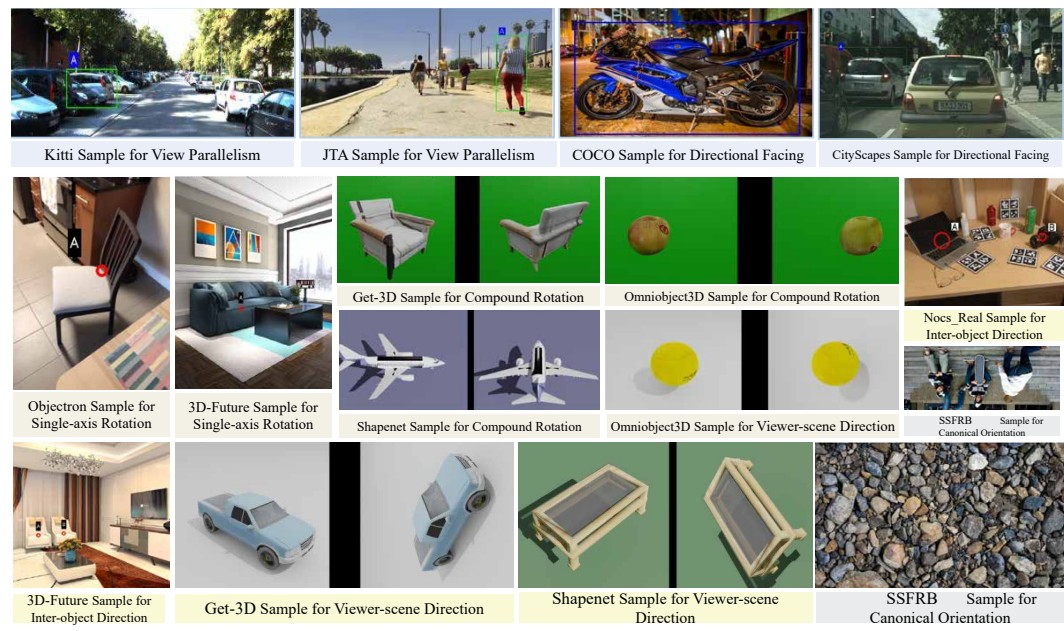

Figure 8: Representative image samples drawn from each constituent dataset. Natural datasets include Kitti, Cityscapes, Coco, SSFRB, Nocs-real, and Objectron. In the figure, we also show simulated dataset samples including: JTA, 3D-Future, Get-3D, Omniobject3D, and Shapenet.

## F  Dataset Composition and Visual Diversity

To support a comprehensive evaluation of Multimodal Large Language Models (MLLMs) on orientation reasoning and object-centric understanding, we construct our dataset by curating and aligning images from diverse computer vision sources. Specifically, DORI incorporates scenes from diverse computer vision datasets including KITTI (Geiger et al., 2012), Cityscapes (Cordts et al., 2016), COCO (Lin et al., 2014), JTA (Fabbri et al., 2018), 3D-FUTURE (Fu et al., 2021), Objectron (Ahmadyan et al., 2021), ShapeNet (Guibas, 2017), OmniObject3D (Wu et al., 2023), and SSFRB (unsplash, 2020; Gontier et al., 2023; Liu et al., 2024c; Willett et al., 2013; neelgajare, 2022). These datasets span various natural and simulated domains, capturing varied object instances, backgrounds, occlusion levels, lighting conditions, and viewpoints.

Fig. 8 showcases representative image samples from each source, illustrating the diversity of environments, object types, and scene structures. This breadth of distribution ensures that the evaluation probes both the generalization ability of MLLMs and their robustness to contextual, visual, and category shifts across domains.

Fig. 9 further complements this by presenting the distribution of the top 25 most frequent object categories across the datasets. While a few object classes, such as *car*, *person*, *camera*, and *chair* dominate in frequency, the distribution spans a broad range of object types and source domains. This heterogeneity plays a critical role in evaluating the performance of models not only on popular categories but also on long-tail classes, thereby encouraging more balanced and comprehensive assessment.

## G  Task Complexity Hierarchy

Our multi-dimensional analysis reveals a clear hierarchy of difficulty in orientation reasoning tasks that closely mirrors human cognitive development. Infants first master basic frontal orientation recognition before developing the neural machinery for complex mental rotation operations (Tversky & Suwa, 2009; Mallot, 2023; Vasilyeva & Lourenco, 2012). Similarly, models perform most competently on frontal alignment tasks (particularly view parallelism), where the top model (Gemini 1.5 Pro in Table

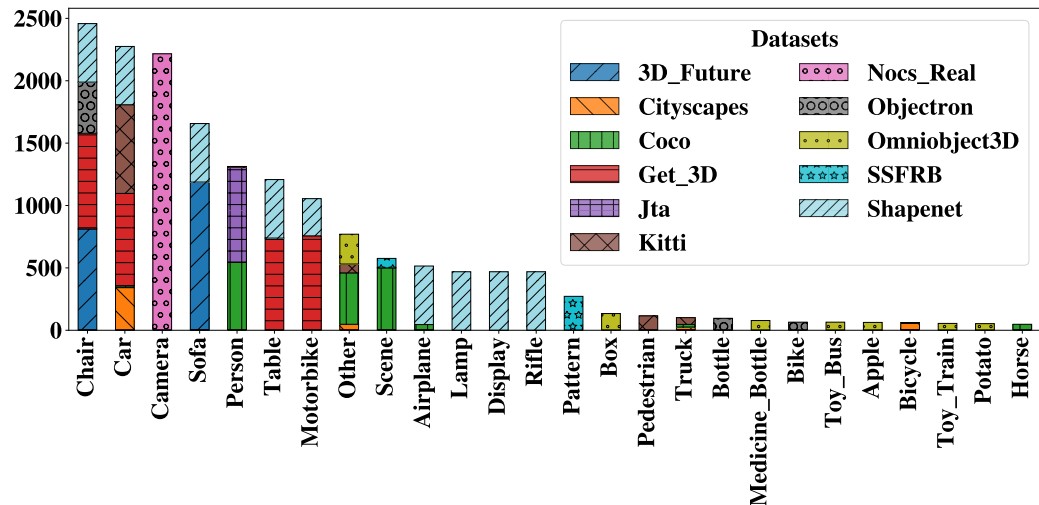

Figure 9: Sample category distribution of top 25 object categories showing their sources

3 in the main paper) achieves 68.5% coarse accuracy. Performance systematically deteriorates as tasks require more complex transformations, with compound rotation proving exceptionally challenging (best granular performance of only 15.3% achieved by GPT-4o in Table 3 in the main paper). The models' difficulty with compound rotations strongly suggests they lack the neural inductive biases that allow humans to mentally simulate and track objects through complex transformations. The most striking performance gap appears in tasks requiring perspective shifts—specifically, *i.e.* viewer-scene direction perception and inter-object direction perception in Tables 2 and 3 in the main paper. While viewer-scene direction perception shows relatively strong performance (Gemini models reaching 91-92% coarse accuracy), inter-object direction tasks reveal a fundamental weakness across all models (best performance of only 25% coarse). This disparity demonstrates that current MLLMs struggle to mentally adopt perspectives different from the allocentric viewpoint—a cognitive ability that humans develop through perspective-taking experiences. This limitation is particularly concerning for embodied AI applications like robotics and navigation, which require reasoning about object relationships from multiple viewpoints. Canonical orientation understanding (determining if objects appear in their "natural" orientation) shows highly variable performance across models (Tab. 2 in the main paper). Some systems perform exceptionally poorly (DeepSeek-1.3B-base at 2.5% coarse) while others demonstrate relatively robust capabilities (GPT-4o at 45% and GPT-4-1 at 46% granular, as shown in Table 3 in the main paper). This variability suggests that recognizing natural object orientations depends on world knowledge that is inconsistently encoded across different architectural approaches and training regimes.

## H    ADDITIONAL ANALYSES OF MODEL PERFORMANCE

To deepen our understanding of current Multimodal Large Language Models (MLLMs) on DORI, we present a series of additional analyses that dissect performance across key axes of task structure and model behavior.

Fig. 10 examines the relationship between answer set size and model accuracy. We observe a peak in performance at 3-option questions (42.5%), with accuracy declining markedly as the number of candidate answers increases. At 5 and 6 options, performance drops to 26% and 19%, respectively, with a further decline to 13% at 9 options, and a minimum of 6% at 16 options. This degradation illustrates the difficulty MLLMs face when navigating more complex decision spaces, suggesting that increasing output space size strains their orientation reasoning capabilities.

In Fig. 11, we break down model performance across simulated vs. natural imagery and annotation granularity levels. Gemini and GPT variants dominate across the board. Yet, all models exhibit

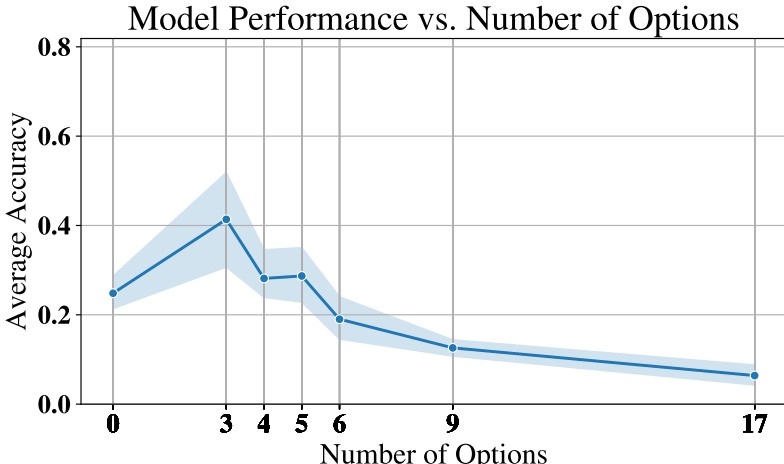

Figure 10: Model accuracy as a function of answer set size. Accuracy peaks at 42.5% for 3-option questions, then declines steadily, dropping to about 25% at 5 options, 19% at 6, about 13% at 9 options, and reaching a low of about 6% at 16 options, indicating increasing difficulty with larger candidate sets.

significantly higher accuracy on simulated datasets and coarse-type questions, revealing a persistent challenge in transferring orientation understanding to more fine-grained and realistic visual contexts.

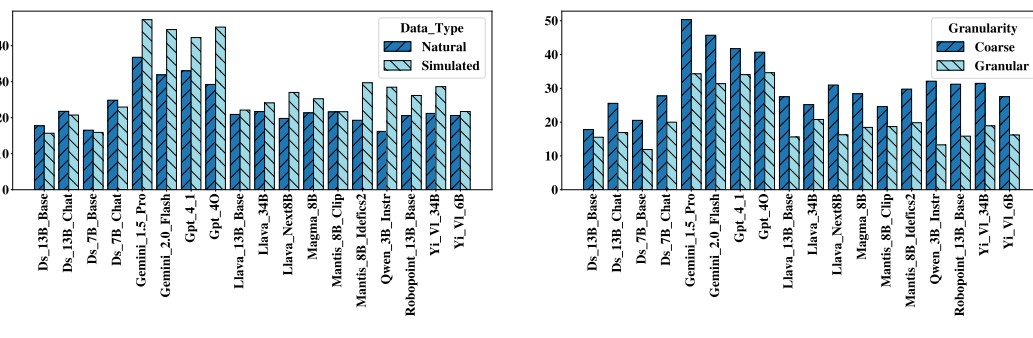

(a) Performance grouped by data type.   (b) Performance grouped by question granularity.

Figure 11: Performance of 18 leading Multimodal Large Language Models (MLLMs) across data types **(a)** and annotation granularity levels **(b)**. **Gemini** and **GPT** models lead overall. Models perform noticeably better on simulated datasets and coarse-type questions, revealing a gap in generalizing to natural images and more fine-grained questions.

### H.1 PERFORMANCE BY COMMON CATEGORIES

We can see in Fig 12 we look at the performance for each of the models based on common categories. We see a similar trend for each model across each of the different categories with Food being the one that appears to standout. In this instance it seems that only the GPT variants perform better with GPT-4o performing the best. In terms of datasets that contain food in them, this would be attributed to the COCO images for the Directional Facing question and OmniObject-3D images for Compound Rotation and View-scene direction questions. Where some of these questions tend to have "Cannot be determined" since it involves items like Pizza or Cake, GPT-4o appears to be able to determine these sort of questions. This could be attributed to the training data that was used for the GPT family of models since GPT-4-1 also is shown to do well for the Food category. For the person category, however, most models tend to perform best as is expected as most models would have been exposed

to persons in their training data, for instance if we look at the most popular category in COCO (Lin et al., 2014) it is persons.

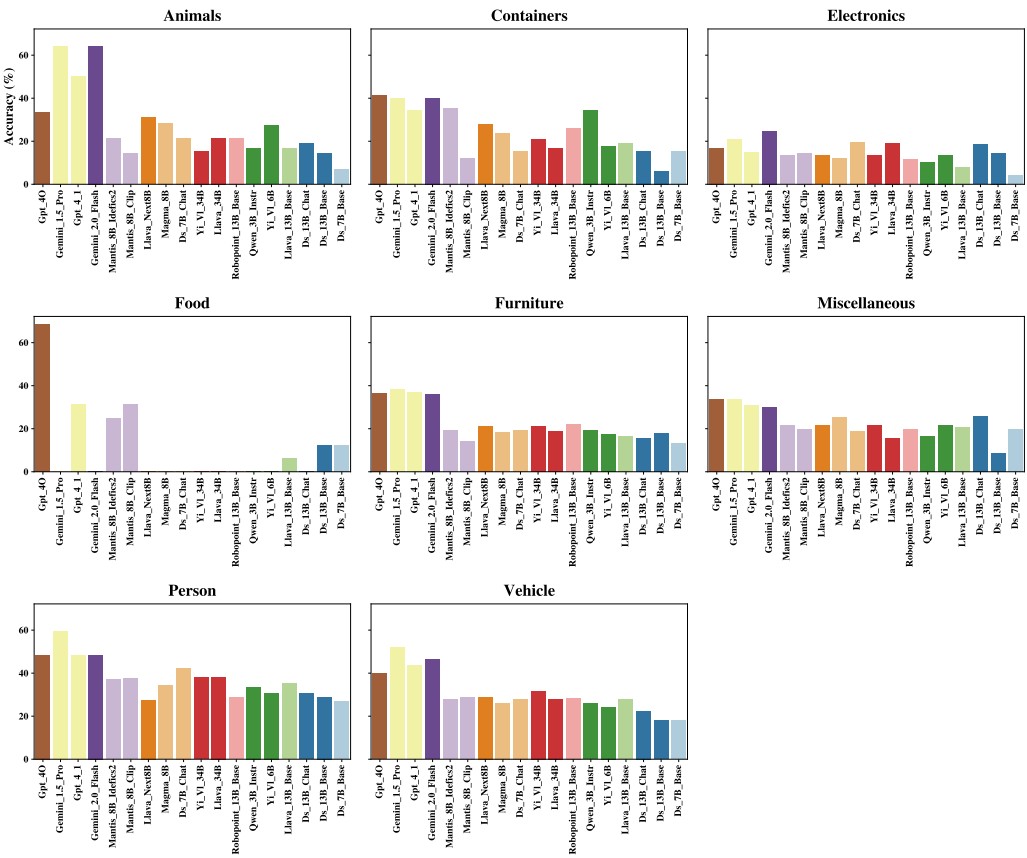

Figure 12: Model accuracy for the 18 models across the broad categories described in Section E with most categories tending to performing similarly except for Food which tends to have lower performances except for GPT-4o. Which highlights a gap for orientation related questions related to food.

## H.2 Performance by Data-Source

When looking at Fig 13 we see the main data-sources present in our dataset. We see that for more simple synthetic data models tended to perform better like in JTA, ShapeNet and Get3D; however for natural datasets like Objectron and NOCS_Real, both the open and closed source models tended to have a drop in performance. This could be first attributed to the dense nature of the number of objects found in the NOCS_Real dataset which includes a number of objects like cars and persons, the same can be said for Cityscapes which is also densely populated. For Objectron which can look at an object at different angles this can also prove difficult for most open-source models, with the closed source models tending to performing better which could be attributed to the pretraining data as these closed-source models are exposed to a large set of data with object potentially at different views making it more robust to object rotational views.

## H.3 Performance by Linear Projection vs Token-based Fusion

We also tried to determine the effect of training a Qwen2.5-3B-Inst. model finetuned on the DORI dataset to determine the performance of using a linear projection method versus the token-based fusion method the model is trained with. We additionally wanted to determine the effect of including a bottleneck in the linear projection method to determine the effect it would have on results. We

Table 9: Models performance of the Qwen2.5-3B-Inst. model finetuned using LoRA using Linear Projection, Linear Projection with a bottleneck and Token-based Fusion on the DORI dataset, all methods are trained for 2000 steps

| | Frontal Alignment | | | | Rotational Transformation | | | | Relative Orient. | | | | Canonical Orient. | | | |
|---|---|---|---|---|---|---|---|---|---|---|---|---|---|---|---|---|
| | View Parallel. | | Dir. Facing | | Single-axis Rot. | | Compo-und Rot. | | Inter-Obj. Dir. | | Viewer-scene Dir. | | | | Avg. | |
| | C | G | C | G | C | G | C | G | C | G | C | G | C | G | C | G |
| Linear Proj | 55.9 | 39.4 | 19.4 | 0.0 | 36.6 | 28.0 | 55.9 | 5.6 | 25.0 | 17.8 | 77.0 | 26.2 | 0.0 | 0.0 | 38.5 | 16.7 |
| Linear Proj+Btlneck | 55.6 | 32.4 | 26.1 | 0.0 | 19.1 | 23.8 | 30.1 | 5.0 | 19.0 | 10.8 | 41.6 | 18.8 | 12.9 | 2.9 | 29.2 | 13.4 |
| Token based fusion | 87.2 | 72.4 | 75.0 | 2.5 | 64.9 | 65.6 | 63.1 | 14.9 | 67.7 | 56.5 | 94.1 | 69.5 | 42.9 | 35.2 | 70.7 | 45.2 |

utilized a single linear layer fusion module while the bottleneck variant included a linear module with a small hidden dimension (256) with the activation function being GELU. The token-based fusion method is the default method used by the QWEN model. All methods are trained for 2000 steps, we can see from Table 9 that the token-based fusion method performs the best, however we will highlight that the way that the QWEN family of methods are trained utilized the token-based fusion method, hence this is not an exhaustive experiment since we did not train the QWEN model from scratch using the linear projection method with all the data used to train the original QWEN model.

# I  VQA Examples in DORI

To further contextualize model behavior, we present a curated selection of Visual Question Answering (VQA) examples drawn from the DORI benchmark, covering each of the seven distinct orientation reasoning tasks. These qualitative illustrations shed light on the nuanced challenges faced by state-of-the-art Multimodal Large Language Models (MLLMs), beyond aggregate metrics.

Each example is carefully chosen to represent either a canonical failure or, in rarer cases, a surprising success. The samples span both coarse-level and granular-level questions, reflecting the dual axes of abstraction and visual complexity within DORI. Despite confident language in many model predictions, we observe frequent dissonance between answer correctness and the underlying rationale, especially for tasks requiring precise spatial alignment or inter-object reasoning.

For instance, in the **View Parallelism** and **Directional Facing** tasks, models often misjudge subtle orientation cues, such as limb articulation or gaze direction as we show in Figures 15, 16, 17, and 18, leading to confidently incorrect predictions. Likewise, for **Single-Axis** and **Compound Rotation** scenarios, we show in Figures 19, 20, 21, and 22, that even top-performing models struggle with mentally simulating object motion, resulting in angular miscalculations or overly generic justifications.

Notably, **Inter-object Direction** and **Viewer-Scene Direction** tasks as we show in Figures 23, 24, 25, and 26, expose limitations in relational orientation reasoning, with models frequently underestimating angular disparities or misrepresenting the directional frame of reference. The final task, **Canonical Orientation** as seen in Fig. 27, underscores a broader epistemic gap: models often assert certainty in inherently ambiguous scenarios, revealing an overconfidence not grounded in the visual evidence.

Together, these examples highlight persistent limitations in visual-spatial grounding, even among the most capable contemporary MLLMs. They underscore the need for further architectural innovations and training strategies to imbue models with a deeper, more structured understanding of object orientation dynamics.

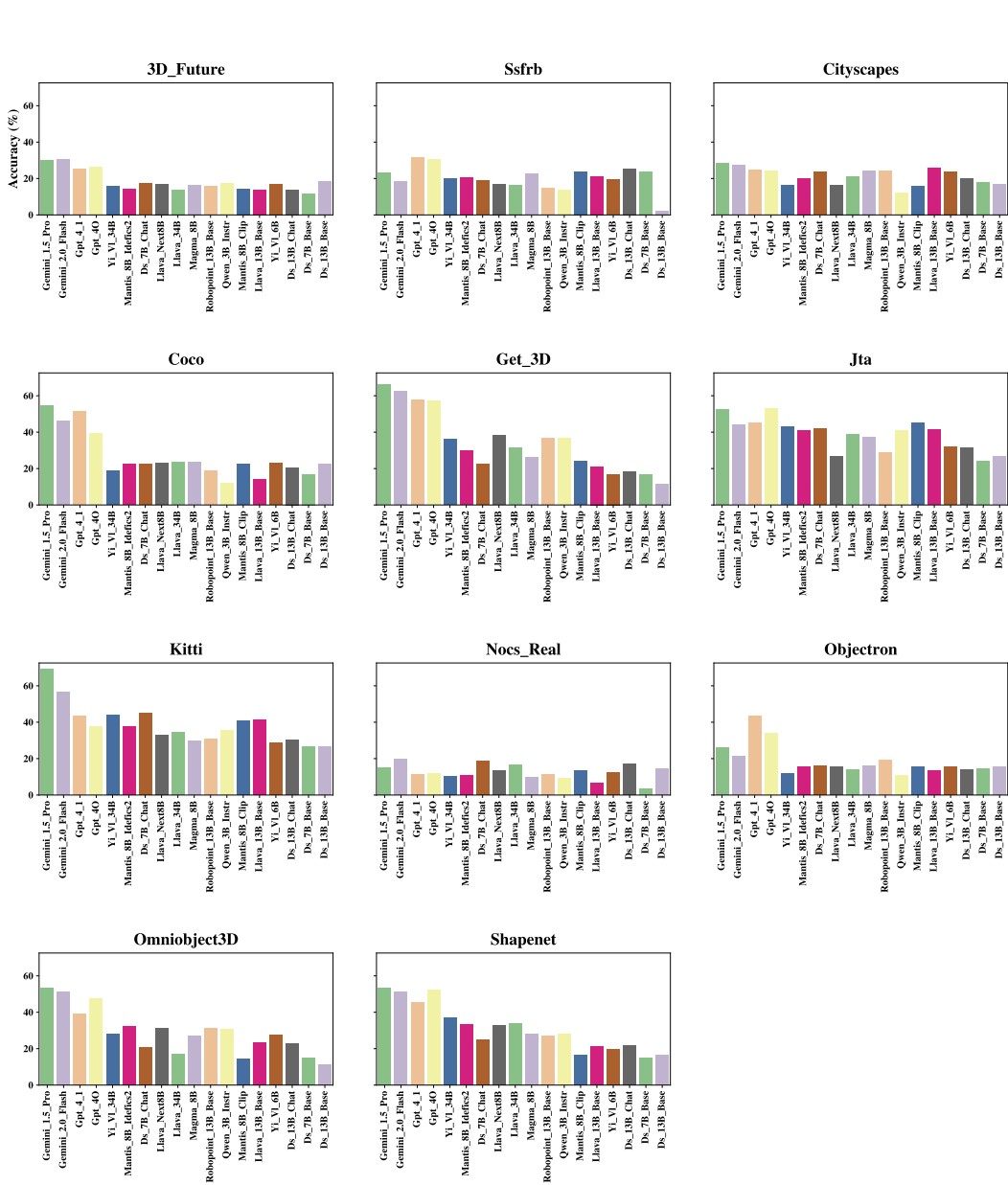

Figure 13: Model accuracy for the 18 models across the data-sources present in DORI. We see that overall the closed source models tend to perform better than the open source models. With more real and smaller datasets like NOCS_Real and Objectron proving to be challenging for most of the models.

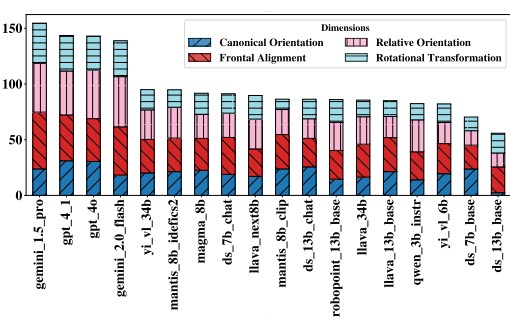
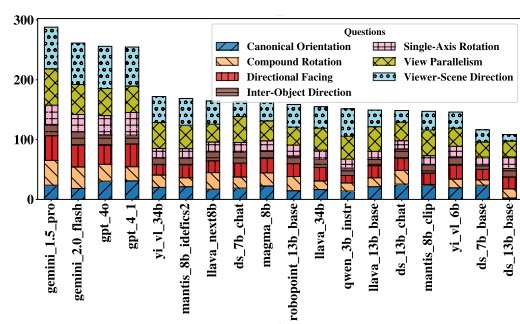

(a) Performance per model across the four core dimensions of orientation.

(b) Performance per model across the different questions present in DORI.

Figure 14: Performance of 18 leading Multimodal Large Language Models (MLLMs) different levels of our questions hierachy with **(left)** being the being the four core dimensions of orientation and **(right)** being the different questions present in DORI

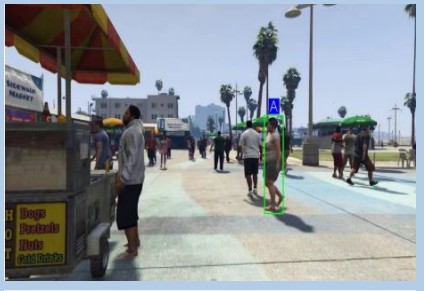

**Ground Truth:**

C: Facing away from the camera

**Gemini_1.5_pro Answer:**

"answer": "B",

"reasoning": "Object A is a person. The key structural feature for a person is their face. We can see her face clearly and we can tell her body is oriented towards the camera, therefore Object A is facing directly towards the camera."

**Prompt:**

TASK: Determine which way Object A's front is facing relative to the camera.

CONTEXT: Object A is marked with a bounding box in the image. An object is considered 'front facing' when its inherent structural features (e.g., a person's face, car headlights, TV screen) are visible from the camera.

Please analyze this image using these steps:
1. Identify Object A and its key structural features 2. Determine the orientation of its front surface relative to the camera 3. Compare with the provided options 4. Select the most appropriate answer 5. Explain your reasoning with specific visual evidence

OPTIONS:
A. Not visible enough to determine B. Directly facing the camera
C. Facing away from the camera

FORMAT YOUR RESPONSE AS:
"answer": "Letter/number of chosen option",
 "reasoning": "Step-by-step explanation citing specific visual evidence"

EXAMPLES:
- A person whose body is directed towards the camera would be "facing toward the camera"

Figure 15: An example VQA from the **View Parallelism orientation task** illustrating a failure case on a coarse-level question in DORI, with a prediction from Gemini 1.5 Pro. Many models failed this question, highlighting a common challenge among MLLMs in understanding fundamental scene geometry—specifically, in reasoning whether an object's front-facing surface is oriented toward, away from, or perpendicular to the camera plane.

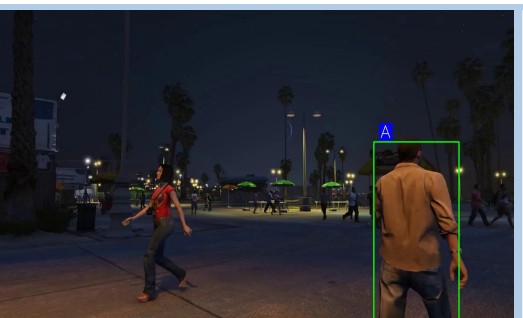

**Prompt:**

**TASK:** Determine how much Object A's front surface deviates from being parallel to the camera plane.

**CONTEXT:** Object A is marked with a bounding box in the image. You'll be assessing the angle between Object A's front surface and the camera plane (the camera taking this photo). Please analyze this image using these steps: 1. Identify Object A and establish its front-facing surface 2. Estimate the angular deviation between this surface and the camera plane 3. Compare with the provided options 4. Select the most appropriate answer 5. Explain your reasoning with specific visual evidence

**OPTIONS:** A. 135 degrees to 180 degrees B. 0 degrees to 15 degrees C. Cannot be determined D. 65 degrees to 95 degrees

**FORMAT YOUR RESPONSE AS:** "answer": "Letter/number of chosen option", "reasoning": "Step-by-step explanation citing specific visual evidence" EXAMPLES: - A person looking directly at the camera: 0-15 degrees - A person is perpendicular to the camera: 65-95 degrees - A car with its rear-end to the camera: 135-180 degrees

**Ground Truth:**

A : 135 degrees to 180 degrees

**Gemini-2.0-Flash Answer:**

"Answer": "B",

"Reasoning": "Object A's back is to the camera. Therefore, the deviation from the camera plane is between 0 degrees and 15 degrees."

Figure 16: An example VQA from the **View Parallelism orientation task** highlighting a failure case on a granular-level question in DORI, with a response from Gemini 2.0 Flash. Although the model confidently selects an answer, its reasoning reflects a fundamental misunderstanding of the object's orientation relative to the camera. While the ground truth indicates the object is turned between 135° and 180° away, the model incorrectly infers a near-parallel alignment, underscoring limitations in fine-grained spatial perception among current MLLMs.

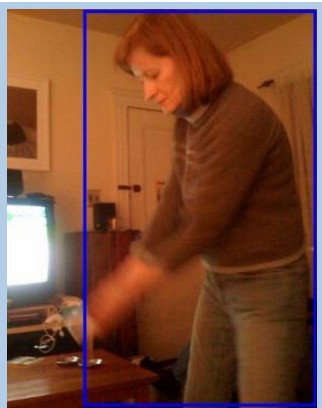

**Prompt:**
**TASK:** Determine which direction Object A's front-facing surface is oriented from the camera's viewpoint.

**CONTEXT:** Object A is marked with a bounding box in the image. You will identify its front-facing surface orientation based on inherent structural features.Please analyze this image using these steps:1. Identify Object A and establish its front-facing surface 2. Determine the direction this surface is oriented relative to the camera3. Compare with the provided options 4. Select the most appropriate answer 5. Explain your reasoning with specific visual evidence

**OPTIONS:**{'A': 'Toward the camera' 'B': "Can't be determined" 'C': 'Away from the camera' 'D': 'Rightward relative to the camera plane' 'E': 'Leftward relative to the camera plane'}

**FORMAT YOUR RESPONSE AS:** "answer": "Letter/number of chosen option" "reasoning": "Step-by-step explanation citing specific visual evidence"

**EXAMPLES:**- If you see a car with its headlights directly pointing toward the camera select "Toward the camera"- If you see a person with their back to the camera select "Away from the camera"

**Ground Truth:**

E: Leftward relative to the camera plane

**Gpt_4_1 Answer:**

Answer: "B"

"reasoning": "The person inside the bounding box (Object A) is facing away from the camera. This is evident because the back of their head and body are visible while their face is not. The orientation of the shoulders and the direction of the arms also indicate that the front-facing surface (the person's face and torso) is directed away from the camera."

Figure 17: A coarse-level VQA from the **Directional Facing task** in DORI, illustrating a failure case by GPT-4-1. While the ground truth indicates a leftward orientation, the model predicts 'away,' misreading body posture and head direction. Notably, 13 out of 15 models failed this question, underscoring a widespread struggle with coarse directional inference.

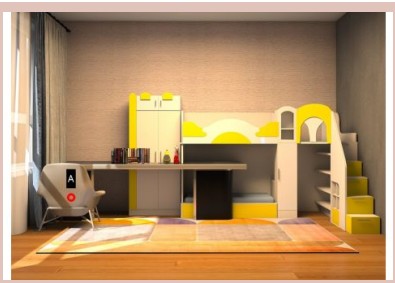

**Prompt:**

TASK: Determine which direction Object A's front-facing surface is oriented from the camera's viewpoint.

CONTEXT: Object A is marked with a bounding box in the image. You will identify its front-facing surface orientation based on inherent structural features. Please analyze this image using these steps: 1. Identify Object A and establish its front-facing surface 2. Determine the direction this surface is oriented relative to the camera 3. Compare with the provided options 4. Select the most appropriate answer 5. Explain your reasoning with specific visual evidence

OPTIONS: A. Can't be determined B. Rightward relative to the camera plane C. Away from the camera D. Toward the camera E. Leftward relative to the camera plane FORMAT YOUR RESPONSE AS: "answer": "Letter/number of chosen option", "reasoning": "Step-by-step explanation citing specific visual evidence"

EXAMPLES: - If you see a car with its headlights directly pointing toward the camera, select "Toward the camera" - If you see a person with their back to the camera, select "Away from the camera"

**Ground Truth:**

B : Rightward relative to the camera plane

**Gemini-2.0-Flash Answer:**

"Answer": "E",

"Reasoning": "The giraffe in the image (Object A) is oriented so that its front-facing surface (as indicated by its face, neck, and body) is primarily directed leftward relative to the camera's perspective. This is because the giraffe is largely facing towards the left side of the image."

Figure 18: A coarse-level VQA from the **Directional Facing task** in DORI, showing a failure case by Gemini-2.0-Flash. While the ground truth identifies the giraffe's orientation as rightward, the model incorrectly infers a leftward direction. This highlights the difficulty in interpreting animal pose and orientation cues, a challenge shared by the majority of models in this example.

**Prompt:**

TASK: Determine the closest clockwise rotation needed for Object A to face the camera.

CONTEXT: Object A is marked with a bounding box in the image. Objects can rotate clockwise or counterclockwise around a vertical axis (running from top to bottom).Please analyze this image using these steps: 1. Identify Object A and establish its front-facing surface 2. Determine its current orientation relative to the camera3. Calculate how many degrees of clockwise rotation would be needed to make it face the camera4. Compare with the provided options 5. Select the most appropriate answer 6. Explain your reasoning with specific visual evidence

OPTIONS:{'A': '0 degrees' 'B': '90 degrees' 'C': '45 degrees' 'D': '135 degrees' 'E': '180 degrees' 'F': 'Cannot be determined'} FORMAT YOUR RESPONSE AS:"answer": "Letter/number of chosen option" "reasoning": "Step-by-step explanation citing specific visual evidence"

EXAMPLES:- If the object is already facing the camera the answer is "0 degrees"- If the object's front is facing to your right a "90 degrees" clockwise rotation would make it face the camera- If the object's front is at a 45-degree angle to your right a "45 degrees" clockwise rotation would make it face the camera- If the object's front is at a 135-degree angle to your right a "135 degrees" clockwise rotation would make it face the camera- If the object's front is facing away from the camera a "180 degrees" clockwise rotation would make it face the camera- If the object has no clear front or its orientation cannot be determined select "Cannot be determined"

**Ground Truth:**

E: 180 degrees

**GPT-4O Answer:**

"Answer": "E"
"reasoning": "The object marked as 'A' is a chair and it does not have a clear front-facing surface that can be definitively identified. Without a clear indication of its front orientation it is impossible to determine how many degrees of clockwise rotation would be needed for it to face the camera. Therefore the correct answer is 'Cannot be determined."

Figure 19: A granular-level VQA from the **Single-Axis Rotation** task in DORI, featuring a case where GPT-4o selects the correct answer (180° rotation) but provides flawed reasoning. While the model concludes with the correct choice, its explanation suggests uncertainty due to perceived ambiguity in object orientation. This mismatch between prediction and rationale underscores gaps in visual reasoning, with only 2 out of 15 models answering correctly.

**Prompt:**

TASK: Determine the closest clockwise rotation needed for Object A to face the camera.

CONTEXT: Object A is marked with a bounding box in the image. Objects can rotate clockwise or counterclockwise around a vertical axis (running from top to bottom). Please analyze this image using these steps: 1. Identify Object A and establish its front-facing surface 2. Determine its current orientation relative to the camera 3. Calculate how many degrees of clockwise rotation would be needed to make it face the camera 4. Compare with the provided options 5. Select the most appropriate answer 6. Explain your reasoning with specific visual evidence

OPTIONS: A. 0 degrees B. 90 degrees C. 45 degrees D. Cannot be determined E. 180 degrees F. 135 degrees FORMAT YOUR RESPONSE AS: "answer": "Letter/number of chosen option", "reasoning": "Step-by-step explanation citing specific visual evidence"

EXAMPLES: - If the object is already facing the camera, the answer is "0 degrees" - If the object's front is facing to your right, a "90 degrees" clockwise rotation would make it face the camera - If the object's front is at a 45-degree angle to your right, a "45 degrees" clockwise rotation would make it face the camera - If the object's front is at a 135-degree angle to your right, a "135 degrees" clockwise rotation would make it face the camera - If the object's front is facing away from the camera, a "180 degrees" clockwise rotation would make it face the camera - If the object has no clear front or its orientation cannot be determined, select "Cannot be determined"

**Ground Truth:**

E: 180 degrees

**Gemini-2.0-Flash Answer:**

"Answer": "B",

"Reasoning": "Object A is currently oriented approximately 90 degrees clockwise from facing the camera. A clockwise rotation of 90 degrees would be needed to make it directly face the camera."

Figure 20: A granular-level VQA from the **Single-Axis Rotation** in DORI, showing a failure case with a prediction from Gemini 2.0 Flash. While the correct answer is a 180° rotation, the model incorrectly selects 90° (option B), suggesting a misjudgment of the object's current orientation relative to the camera. This highlights challenges models face in reasoning about precise rotational alignment.

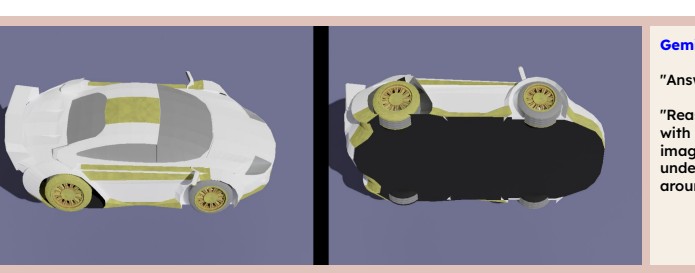

**Gemini-2.0-Flash Answer:**

"Answer": "C",

"Reasoning": "The car in the first image is upright with the wheels on the bottom. In the second image, the car is upside down with the undercarriage showing. This indicates a rotation around a horizontal axis."

**Prompt:**

TASK: Determine what type of rotation the object has undergone between the two images.

CONTEXT: Two images of the same object are shown side by side. The object can rotate around two axes: Horizontal (like a cartwheel coming out of the screen) and Vertical (like a ballerina spinning clockwise). Please analyze these images using these steps: 1. Identify the object in both images and establish its key features 2. Compare the object's orientation in both images 3. Determine which axes the object has rotated around 4. Compare with the provided options 5. Select the most appropriate answer 6. Explain your reasoning with specific visual evidence

OPTIONS: A. Cannot be determined B. Only Vertical C. Only Horizontal D. Both Horizontal and Vertical E. No Change FORMAT YOUR RESPONSE AS: "answer": "Letter/number of chosen option", "reasoning": "Step-by-step explanation citing specific visual evidence"

EXAMPLES: - If an object appears to have moved to face left or right between images, select "Only Vertical" - If an object appears to have moved up or down (like nodding), select "Only Horizontal" - If an object has rotated both horizontally and vertically, select "Both Horizontal and Vertical" - If the object appears identical in both images with no rotation, select "No Change" - If you cannot clearly determine the type of rotation, select "Cannot be determined"

Figure 21: A coarse-level VQA from the **Compound Rotation task** in DORI, with a correct prediction from Gemini 2.0 Flash. The model identifies a horizontal axis rotation, but its reasoning overstates the change, describing the car as 'upside down' when it is only partially inverted.

**Ground Truth:**
E: 90 Degrees Horizontal, 90 Degrees Vertical

**Gemini-2.0-Flash Answer:**

"Answer": "E",

"Reasoning": "The motorcycle in the first image is upright. In the second image, the motorcycle is rotated 90 degrees horizontally, leaning to the right. Additionally, it has been rotated 180 degrees vertically, as evidenced by the seat and rear tire now facing upward."

**Prompt:**

TASK: Determine the exact rotation angles the object has undergone between the two images.

CONTEXT: Two images of the same object are shown side by side. The object can rotate around two axes: Horizontal (like a cartwheel coming out of the screen) and Vertical (like a ballerina spinning clockwise). Please analyze these images using these steps: 1. Identify the object in both images and establish its key features 2. Compare the object's orientation in both images 3. Determine the horizontal rotation angle first, then the vertical rotation angle 4. Compare with the provided options 5. Select the most appropriate answer 6. Explain your reasoning with specific visual evidence

OPTIONS: A. 270 degrees Horizontal then 180 degrees Vertical B. 180 degrees Horizontal then 0 degrees Vertical C. Cannot be determined D. 180 degrees Horizontal then 90 degrees Vertical E. 90 degrees Horizontal then 180 degrees Vertical F. 0 degrees Horizontal then 0 degrees Vertical G. 270 degrees Horizontal then 90 degrees Vertical H. 270 degrees Horizontal then 270 degrees Vertical I. 0 degrees Horizontal then 90 degrees Vertical J. 180 degrees Horizontal then 270 degrees Vertical K. 0 degrees Horizontal then 180 degrees Vertical L. 0 degrees Horizontal then 270 degrees Vertical M. 90 degrees Horizontal then 0 degrees Vertical N. 90 degrees Horizontal then 270 degrees Vertical O. 90 degrees Horizontal then 90 degrees Vertical P. 180 degrees Horizontal then 180 degrees Vertical Q. 270 degrees Horizontal then 0 degrees Vertical

FORMAT YOUR RESPONSE AS: "answer": "Letter/number of chosen option", "reasoning": "Step-by-step explanation citing specific visual evidence"

EXAMPLES: - If an upright cup in the first image appears upside-down in the second image, this might be "180 degrees Horizontal then 0 degrees Vertical" - If a car facing left in the first image appears to be tilted upward and facing the camera in the second image, this might be "90 degrees Horizontal then 90 degrees Vertical" - If you cannot clearly determine the type of rotation, select "Cannot be determined"

Figure 22: A granular-level VQA from the **Compound Rotation task** in DORI showing a failure case from Gemini 2.0 Flash. While the model selects the correct option, its reasoning misidentifies the vertical rotation, describing a 180° flip when the ground truth indicates a 90° transformation.

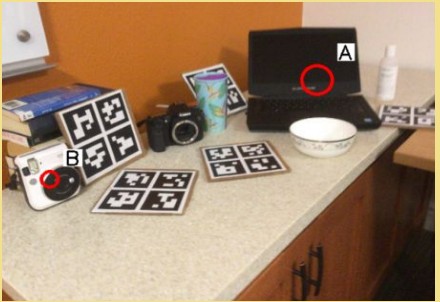

**Prompt:**
TASK: Determine if objects A and B are facing each other from their own perspectives.

CONTEXT: Objects A and B are marked with red dots in the image. You will analyze their relationship from the objects' perspectives not from the camera view. Please analyze this image using these steps: 1. Identify objects A and B and establish their front-facing surfaces 2. Mentally position yourself at each object's location 3. Determine the relative directional orientation between the objects 4. Compare with the provided options 5. Select the most appropriate answer 6. Explain your reasoning with specific visual evidence

OPTIONS:{'A': 'Cannot be determined' 'B': 'Partially facing the same direction' 'C': 'Partially facing the opposite directions' 'D': 'They are facing opposite directions' 'E': 'They are facing perpendicular to each other' 'F': 'They are facing the same direction'}

FORMAT YOUR RESPONSE AS:"answer": "Letter/number of chosen option" "reasoning": "Step-by-step explanation citing specific visual evidence"

EXAMPLES:- Two people looking at each other would be "facing opposite directions" (as their faces point toward each other)- Two cars with headlights pointing the same way would be "facing the same direction"- A car and a person where the car's side faces the person's front would be "facing perpendicular to each other"- Two objects with front surfaces angled slightly toward each other would be "partially facing opposite directions"- Two objects with front surfaces angled slightly away from each other would be "partially facing the same direction"

**Ground Truth:**
B: Partially facing the same direction

**Gpt-4-1 Answer:**

Answer: "C"

"reasoning": "...the most appropriate answer is 'Partially facing the opposite directions' as their front surfaces are somewhat oriented toward each other but not directly..."

Figure 23: A coarse-level VQA from the **Inter-object Direction task** in DORI showing a failure case with GPT-4-1. Although the correct answer is 'Partially facing the same direction,' the model incorrectly selects 'Partially facing opposite directions,' misjudging the relative orientations. Notably, 13 out of 15 models failed this example, highlighting a shared difficulty in reasoning about partially aligned object directions.

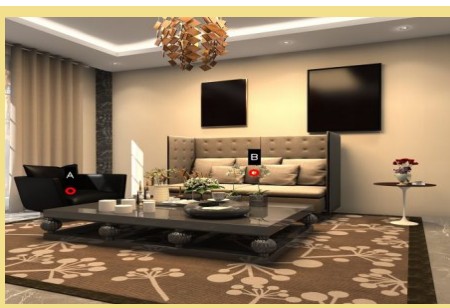

**Prompt**:
TASK: Determine how much Object B would need to rotate to face Object A.

CONTEXT: Objects A and B are marked with red dots in the image. You will analyze their relationship from the objects' perspectives, not from the camera view. Please analyze this image using these steps:
1. Identify objects A and B and establish their front-facing surfaces
2. Mentally position yourself at each object's location
3. Determine how much and in which direction Object B would need to rotate to face Object A
4. Compare with the provided options
5. Select the most appropriate answer
6. Explain your reasoning with specific visual evidence

OPTIONS:
A. Cannot be determined B. No rotation C. 180 degrees in either direction
D. 46 to 90 degrees clockwise E. 46 to 90 degrees counterclockwise
F. 0 to 45 degrees clockwise G. 91 to 135 degrees clockwise
H. 0 to 45 degrees counterclockwise I. 91 to 135 degrees counterclockwise

FORMAT YOUR RESPONSE AS:
"answer": "Letter/number of chosen option",
 "reasoning": "Step-by-step explanation citing specific visual evidence"

EXAMPLES:
- If Object B needs to turn slightly right to match Object A's direction, select "0 to 45 degrees clockwise"
- If Object B is already facing the same direction as Object A, select "0 degrees or no rotation"
- If Object B needs to turn completely around to match Object A, select "180 degrees in either direction"

**Ground Truth:**
D: 46 to 90 degrees clockwise

**Gemini_1.5_pro Answer:**

"Answer": "F",
"reasoning": "Object A (the black chair) is facing the coffee table in the center of the room. Object B (the sofa) is currently facing the wall with the two black canvases. To face the coffee table and therefore object A, object B needs to rotate clockwise a small amount, placing it between 0 and 45 degrees. "

Figure 24: A granular-level VQA from the **Inter-object Direction task** in DORI illustrating a failure case with Gemini 1.5 Pro. While the ground truth indicates a required clockwise rotation between 46° and 90° for alignment, the model underestimates this angle. Its reasoning misinterprets the spatial alignment between the chair and sofa. All 15 models failed this example, underscoring the challenge of inter-object directional understanding.

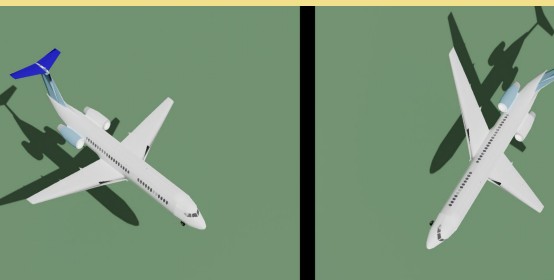

**Gemini-2.0-Flash Answer:**

"Answer": "C",

"Reasoning": "The airplane rotates 90 degrees clockwise. In the first picture, the plane's nose points towards the right, in the second picture, it points to the bottom."

**Prompt:**
TASK: Determine how many degrees clockwise the object has rotated between the two images.

CONTEXT: Two images of the same object are shown side by side. Assume the camera remains stationary, and the object can rotate around a vertical axis. Please analyze these images using these steps: 1. Identify the object in both images and establish its key features 2. Compare the object's orientation in both images 3. Determine the precise clockwise rotation angle around the vertical axis 4. Compare with the provided options 5. Select the most appropriate answer 6. Explain your reasoning with specific visual evidence

OPTIONS: A. Cannot be determined B. 180 degrees C. 90 degrees D. 270 degrees E. 0 degrees

FORMAT YOUR RESPONSE AS: "answer": "Letter/number of chosen option", "reasoning": "Step-by-step explanation citing specific visual evidence" EXAMPLES: - Objects that are symmetrical like a ball or a box would be considered "Cannot be determined

Figure 25: **Viewer-Scene Direction** – Example success case on a granular-level DORI question with Gemini-2.0-Flash. While the model correctly selects the 90-degree rotation, it inaccurately describes the directional shift as 'right' to 'bottom' instead of the more precise 'bottom-right' to 'bottom-left,' reflecting a partial misunderstanding of fine-grained orientation.

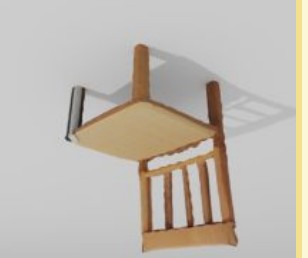

**Ground Truth:**

A: 180 degrees

**Gpt-4-1 Answer:**

Answer: "A"

"reasoning": "...Comparing the two images the chair has rotated 90 degrees clockwise around the vertical axis: the backrest moves from facing upper left to facing right ..."

**Prompt:**
TASK: Determine how many degrees clockwise the object has rotated between the two images.

CONTEXT: Two images of the same object are shown side by side. Assume the camera remains stationary and the object can rotate around a vertical axis.Please analyze these images using these steps: 1. Identify the object in both images and establish its key features 2. Compare the object's orientation in both images3. Determine the precise clockwise rotation angle around the vertical axis 4. Compare with the provided options 5. Select the most appropriate answer 6. Explain your reasoning with specific visual evidence

OPTIONS:{'A': '180 degrees' 'B': '0 degrees' 'C': '270 degrees' 'D': '90 degrees' 'E': 'Cannot be determined'}

FORMAT YOUR RESPONSE AS:"answer": "Letter/number of chosen option"  "reasoning": "Step-by-step explanation citing specific visual evidence" EXAMPLES:- Objects that are symmetrical like a ball or a box would be considered "Cannot be determined"

Figure 26: **Viewer-Scene Direction** – Example failure case on a granular-level DORI question with GPT-4-1. Although the model selects the correct answer of 180 degrees, its reasoning mistakenly describes a 90-degree rotation, highlighting a disconnect between answer selection and spatial understanding. Notably, 13 out of 15 models failed this case.

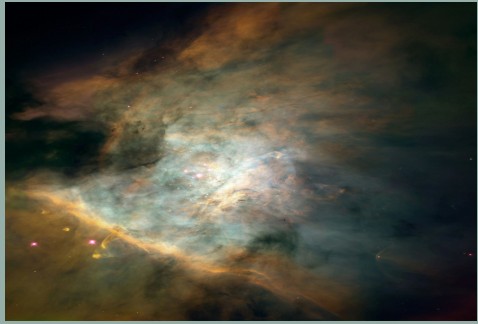

**Prompt:**
TASK: Determine if the image is in its canonical orientation.

CONTEXT: The image may be flipped or rotated or both. Canonical orientation refers to the standard way objects are typically viewed (e.g., people standing upright, cars with wheels on the ground, text reading left to right).An image that has been mirrored horizontally or vertically is considered flipped. An image that has been turned from its normal viewing angle is considered rotated. Please analyze this image using these steps: 1. Identify the main objects/elements in the image 2. Determine their expected canonical orientation 3. Compare with their current orientation in the image 4. Compare with the provided options 5. Select the most appropriate answer 6. Explain your reasoning with specific visual evidence

OPTIONS: A. Yes, the image is in the canonical orientation B. Cannot be determined C. No, the image is not in the canonical orientation

FORMAT YOUR RESPONSE AS: "answer": "Letter/number of chosen option", "reasoning": "Step-by-step explanation citing specific visual evidence"

**Ground Truth:**

B. Cannot be determined

**Gemini-2.0-Flash Answer:**

"Answer": "A",

"Reasoning": "The image depicts a nebula. There is no single canonical orientation for nebulae as they are typically viewed from various angles. Therefore, the image is most likely in its canonical orientation."

Figure 27: **Canonical Orientation** – Example failure case on a coarse-level DORI question with Gemini-2.0-Flash. While the ground truth is 'Cannot be determined,' the model incorrectly selects a definitive orientation. Its reasoning contradicts the inherent ambiguity it acknowledges, exposing uncertainty in handling objects with no fixed canonical pose.

# J  Enumerated List of Questions & Terminology Glossary

## J.1  Questions

**Coarse Questions**

- **Q1 - View Parallelism:** Determine which way Object A's front is facing relative to the camera.
- **Q2 - Directional Facing:** Determine which direction Object A's front-facing surface is oriented from the camera's viewpoint.
- **Q3 - Single-axis Rotation:** Determine the shortest direction of rotation for Object A to face the camera.
- **Q4 - Compound Rotation:** Determine what type of rotation the object has undergone between the two images.
- **Q5 - Inter-object Direction:** Determine if objects A and B are facing each other from their own perspectives.
- **Q6 - Viewer-Scene Direction:** Determine if the object has rotated between the two images.
- **Q7 - Canonical Orientation:** Determine if the image is in its canonical orientation.

**Fine-grained/Granular Questions**

- **Q1 - View Parallelism:** Determine how much Object A's front surface deviates from being parallel to the camera plane.
- **Q2 - Directional Facing:** Identify the precise orientation of Object A's front-facing surface from the camera's viewpoint
- **Q3 - Single-axis Rotation:** Determine the closest clockwise rotation needed for Object A to face the camera.
- **Q4 - Compound Rotation:** Determine the exact rotation angles the object has undergone between the two images.
- **Q5 - Inter-object Direction:** Determine how much Object B would need to rotate to face Object A.
- **Q6 - Viewer-Scene Direction:** Determine how many degrees clockwise the object has rotated between the two images.
- **Q7 - Canonical Orientation:** Determine how the image can be restored to its canonical orientation.

## J.2  Glossary

**Front-facing surface / Frontal surface:**  The front-facing surface (or frontal surface) of an object is the side that carries its primary functional or semantic features, such as a person's face and torso, a car's headlights and grille, or a laptop's screen, and this surface is what is considered to be facing the camera when those features are visible and oriented toward the viewer (e.g., a person standing and looking directly at the camera is presented with their frontal surface clearly visible).

**Canonical orientation** Canonical orientation is the typical, upright, real-world pose in which an object is most commonly encountered or depicted, often aligned with gravity and human conventions (for instance, a car resting on its wheels on a horizontal road, a building with the roof at the top and the foundation at the bottom, or a mug standing upright with the handle to the side), and questions about canonical orientation ask whether the object is in this usual pose or has been rotated or flipped away from it (e.g., a building image rotated 180° so the roof is at the bottom would be judged non-canonical).

**Camera plane / Camera axis** The camera plane (or image plane) is the 2D plane onto which the 3D scene is projected by the camera, while the camera axis (optical axis) is the line perpendicular to this plane passing through the camera center, so that orientation relative to the camera is described by how an object's surfaces are angled with respect to this plane and axis (for example, a frontal surface parallel to the image plane and centered on the optical axis corresponds to an object facing straight toward the camera).

**Camera Viewpoint vs. Object Viewpoint** The camera viewpoint is the position and orientation of the camera in the 3D scene, defining an egocentric frame where directions like "toward the camera" or "left of the camera" live, whereas the object viewpoint is the orientation and placement of the

object relative to its own intrinsic front/back/left/right axes or relative to other objects, so a question might either ask how the object appears from the camera's perspective ("Is the car facing toward the camera?") or, alternatively, how the object is oriented if one imagines standing at the object and looking out ("From the car's viewpoint, is it facing toward or away from the truck?").

**Rotation transformation** A rotation transformation changes an object's orientation by turning it around one or more axes while preserving its shape and size, such as rotating a mug 90° around the vertical axis to bring its handle from the right side to face the camera or rotating a picture frame 180° in the image plane so that it appears upside down relative to its original orientation.

**Sequential transformations** Sequential transformations refer to applying multiple geometric operations in a specific order, typically rotations and flips, where order matters because compositions can yield different final poses, for example first rotating a building image 90° clockwise in the image plane and then flipping it horizontally produces a different orientation than performing the horizontal flip first and then rotating, even though the same primitive operations are used.

**Vertical axis rotation** Vertical axis rotation is a 3D rotation around an axis that runs approximately up–down through the object and aligns with gravity (analogous to yaw), changing which horizontal direction the object's front faces, so turning a car so that it goes from facing east to facing north is a vertical axis rotation of about 90° and a person spinning in place to look from the camera to the right-hand side is another example of such a rotation.

**Horizontal axis rotation** Horizontal axis rotation is a 3D rotation around an axis that runs roughly left–right through the object (analogous to pitch), changing how much the object is tilted toward or away from the camera, so when a person nods their head "yes" or a box tips forward so its top leans toward the camera, these are horizontal axis rotations that alter how the object's surfaces are foreshortened in the image.

**Lateral Axis Rotation** Lateral axis rotation is a rotation around an axis that runs roughly front–back through the object (analogous to roll), causing the object to lean sideways relative to gravity and the image frame, so tilting your head so that your ear moves toward your shoulder or rolling a car so that it lies partially on its side are examples of lateral axis rotations that change which parts appear above or below in the image without changing which direction the object's front is pointing.

**Clockwise rotation** Clockwise rotation is defined with respect to a chosen viewing direction and axis: in this context, for rotations around the vertical axis, it means that when viewed from above along the vertical axis, the object's front turns in the same direction as the hands of a clock (e.g., a car that initially faces north and then rotates to face east has undergone a 90° clockwise rotation about the vertical axis), and in image-plane contexts, clockwise means the top of the object appears to rotate toward the right side of the image.

**Counter-clockwise rotation** Counter-clockwise rotation is the opposite directional sense from clockwise around a specified axis: for vertical axis rotations observed from above, the object's front turns in the opposite direction to clock hands (e.g., a car rotating from facing east to facing north has rotated 90° counter-clockwise about the vertical axis), and in the image plane, counter-clockwise means the top of the object appears to rotate toward the left side of the image.

**Facing each other (objects' perspective)** Two objects are said to be facing each other when each object's front-facing surface points approximately toward the other's position in the scene in an allocentric or object-centric frame, so that if you imagine rays extending from their fronts, those rays intersect between them (for example, two chairs arranged on opposite sides of a table with their seats and backs oriented toward the table center are facing each other)

**Facing same/opposite directions** Objects face the same direction when their front-facing axes are approximately parallel in 3D space, meaning they are oriented with similar yaw so they would move side-by-side if they advanced forward together, whereas they face opposite directions when their front-facing axes differ by about 180°, such as two cars parked in the same lane but one pointing north and the other pointing south along the road.

**Perpendicular facing** Two objects are perpendicular in facing when the angle between their front-facing directions is approximately 90° in yaw, such that one object's front points roughly to the side of the other, for example, when one car is parked facing north and another is parked facing east at an intersection, so their fronts form an L-shape orientation relative to each other.

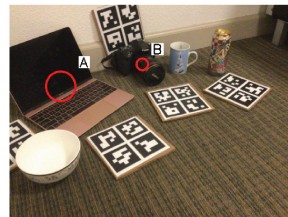
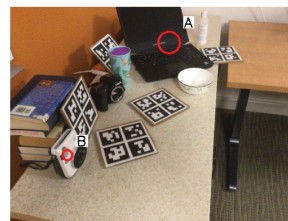

0 to 45 degrees clockwise

0 to 45 degrees counter-clockwise

46 to 90 degrees clockwise

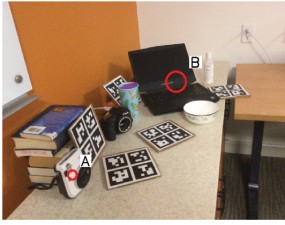
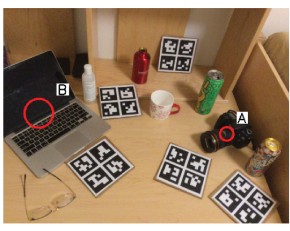
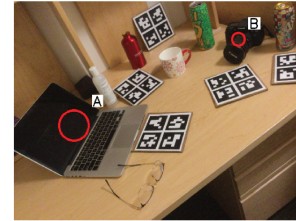

46 to 90 degrees counter-clockwise

91 to 135 degrees clockwise

91 to 135 degrees counter-clockwise

Figure 28: An example of clockwise and counter-clockwise rotations for the Inter-object direction perception task, we utilized samples from the NOCS REAL dataset

**Flip / Mirror** A flip or mirror transformation reverses an image or object along a specified axis without changing its depth order, so a horizontal flip mirrors the scene left–right (like viewing it in a mirror placed vertically), turning a car that originally faces left into one that appears to face right, while a vertical flip mirrors the scene top–bottom, making a building appear upside down with the roof at the bottom of the image.

**Ambiguous objects / Symmetrical objects** Ambiguous or symmetrical objects are those whose shape or appearance does not specify a unique front or canonical orientation because many rotations produce indistinguishable images, such as a perfect sphere, a uniform cylinder, or a highly symmetric abstract logo, so for such objects questions about "which way it is facing" or "whether it is upside down" may have no well-defined answer and are therefore treated as "Cannot be determined" cases in the dataset.

### J.3 CLOCKWISE AND COUNTER-CLOCKWISE ROTATION

We include examples of what clockwise and counter-clockwise rotations looks like as seen in Figure 28 for the Inter-object direction perception task, we utilize samples from the NOCS REAL dataset to highlight these different question samples.

## K PERFORMANCE ERROR BARS

The error bars in Figs. 29, 30, 31, 32, 33, 34, and 35 report the mean and standard deviation of various models on DORI questions. For each model and question type combination, the formula $error = std\_accuracy/sqrt(seed\_count)$ was applied. We used 3 different seeds: $[42, 1998, 107983]$.

Looking at the **View Parallelism** task (Fig. 29), we observe relatively narrow error bars for most models, indicating consistent performance across different seeds. However, both DeepSeek base models and the LLaVa-13 B-base models display wider error bars on coarse questions (approximately $\pm 3 - 4\%$), suggesting that their performance comes with greater variability. Despite LLaVa-13B-base demonstrating better performance than the other 2 base models, it's clear that all three base models' performance is more sensitive to initialization conditions. Qwen2.5-3B-instruct model demonstrates notable initialization variability for both coarse and granular questions.

The **Directional Facing** task (Fig. 30) reveals generally smaller error bars across all models, with most variations under $\pm 2\%$, indicating that performance on cardinal direction assessment remains relatively stable regardless of initialization. However, all models perform notably worse on this task compared to View Parallelism, with even the best model (DeepSeek-7B-Chat) achieving only 32.5% accuracy on coarse questions. LLaVA-Next-8B shows consistent but lower performance, while Qwen2.5-3B-Instruct has low performance with tight error bars.

The **Single-axis Rotation** task (Fig. 31) is more uniform across models, with most showing variations of $\pm 2 - 3\%$. The comparable error bar sizes across models suggest that this task presents similar levels of difficulty for all architectures, with no model demonstrating significantly more stable performance than others. This uniformity in variability indicates that improvements in this task may require fundamental architectural innovations rather than just parameter tuning.

The **Compound Rotation** task (Fig. 32) exhibits the most dramatic performance gap between coarse and granular questions, with granular accuracy dropping below 10% for all models. LLaVA-Next-8B shows the highest coarse performance but extremely low granular performance. The error bars for granular questions are relatively tight (except in LLaVa-13B-base), suggesting that models consistently struggle with this task rather than showing initialization-dependent variability. The narrow error bands on poor performance indicate a systematic limitation in the models' ability to track complex multi-axis rotations. Overall, this plot demonstrates noticeable variability in the error bands across both coarse and granular questions.

For **Inter-object Direction** (Fig. 33), we observe slightly asymmetric performance patterns between coarse and granular questions. While DeepSeek-7B-Chat achieves the highest coarse accuracy, all models show substantially lower performance on granular questions (generally below 15%). The error bars appear to be tight for coarse and granular questions, indicating consistent performance across different trials. The performance gap between coarse and granular questions suggests that while models can sometimes succeed at basic relational orientation tasks (determining if objects face the same/opposite directions), they systematically fail when asked to make precise angular judgments about inter-object relationships. Interestingly, for LLaVA-13B-base model, the granular performance slightly exceeds its coarse performance, running counter to the typical pattern observed in other tasks and models. This anomaly may indicate that LLaVA's training regime potentially encodes some specific features that assist with fine-grained angular estimations between objects, though its overall performance remains well below human capabilities on these tasks.

Similarly, the **Viewer-Scene Direction** task (Fig. 34) reveals intriguing performance inversions between coarse and granular questions for certain models. Qwen-3B-Instruct shows the highest coarse accuracy with poor granular performance, while DeepSeek-7B-Chat demonstrates the opposite pattern. These inversions, coupled with wide error bars, indicate that different model architectures encode rotation perception in fundamentally different ways. This task exposes fundamental inconsistencies in how current MLLMs process orientation changes, suggesting that rotation tracking may rely on different computational mechanisms than static orientation perception, with these mechanisms developing unevenly across model architectures and training regimes.

The **Canonical Orientation** task (Fig. 35) doesn't exhibit any high variability among the error bars across coarse and granular performance. The error bars remain relatively narrow for most models ($\pm 1 - 3\%$), indicating that performance limitations on this task are consistent across initialization seeds rather than highly variable. This consistency, coupled with generally poor performance, suggests that canonical orientation understanding, which requires both world knowledge about natural object positions and spatial transformation reasoning, represents a fundamental capability gap in current MLLMs. The results indicate that models lack robust internal representations of how objects "should" appear in the world, a crucial component for embodied navigation and manipulation tasks where recognizing and correcting non-canonical orientations is essential.

**Cross-Task Insights**  Across all tasks, our error bar analysis reveals several critical insights about MLLMs' orientation reasoning capabilities. First, model performance stability varies substantially across tasks, with simpler perception tasks (View Parallelism, Directional Facing) showing more consistent performance across initializations compared to complex reasoning tasks (Compound Rotation, Viewer-scene direction, Canonical Orientation). Moreover, the consistently tight error bars on poor-performing granular questions, particularly for rotational tasks, indicate systematic

limitations rather than chance variability, suggesting architectural rather than parametric constraints. We also observe that Chat-tuned models generally demonstrate more stable performance than their base counterparts, suggesting that instruction tuning not only improves accuracy but also reduces initialization sensitivity. Finally, the performance inversions observed between coarse and granular questions for some models highlight the disconnect between categorical and precise metric orientation understanding, a fundamental challenge that persists across all model families and architectures evaluated.

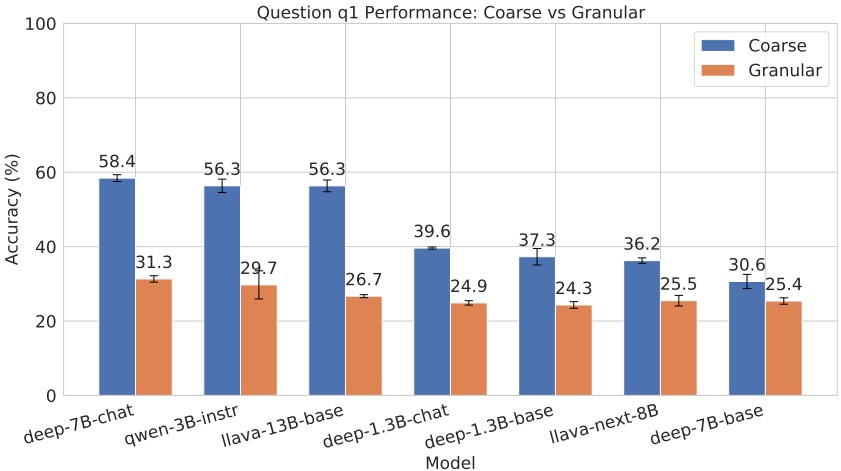

Figure 29: Mean and Standard Deviation of various models on View Parallelism task

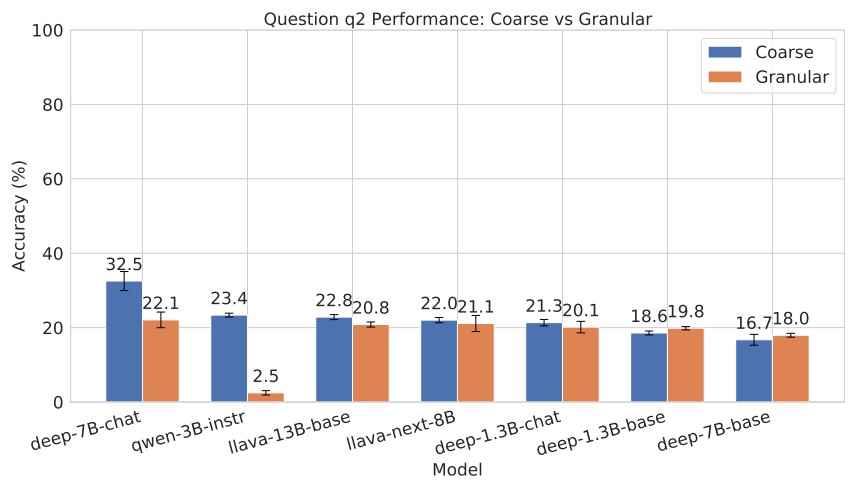

Figure 30: Mean and Standard Deviation of various models on Directional Facing task

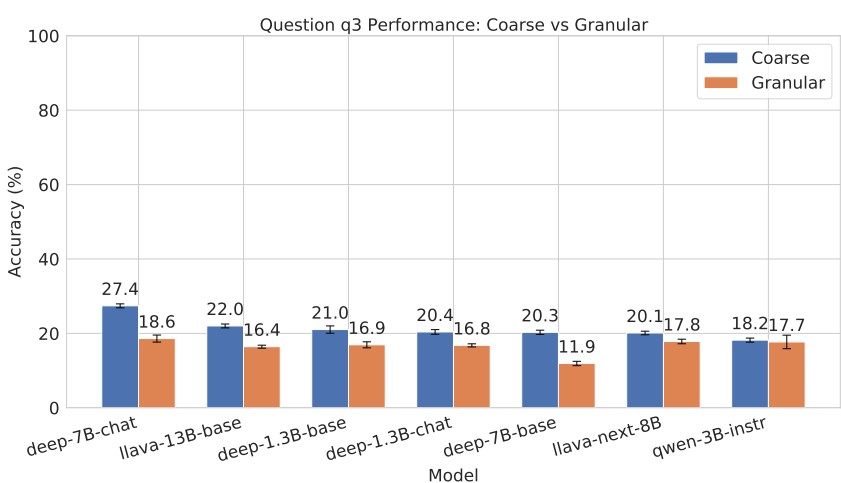

Figure 31: Mean and Standard Deviation of various models on Single-axis Rotation task

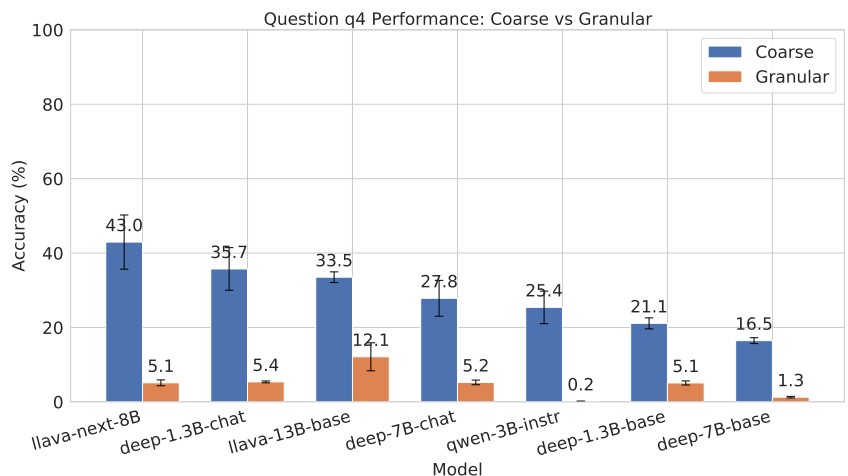

Figure 32: Mean and Standard Deviation of various models on Compound Rotation task

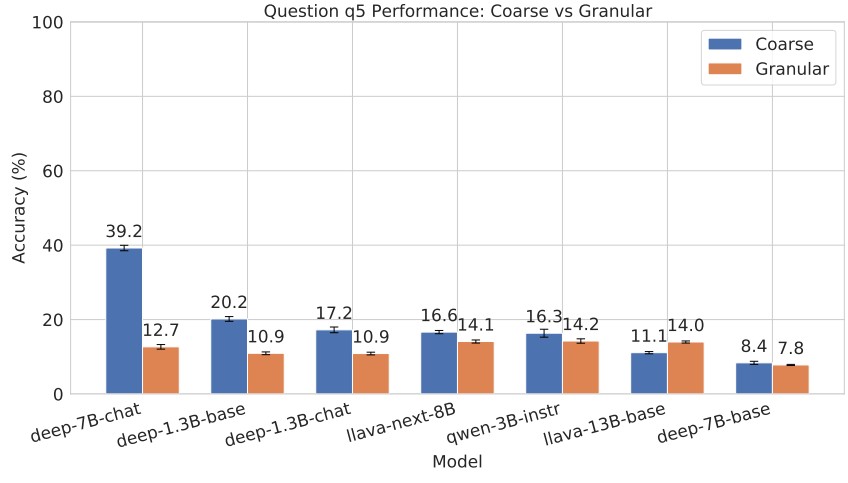

Figure 33: Mean and Standard Deviation of various models on Inter-object direction.

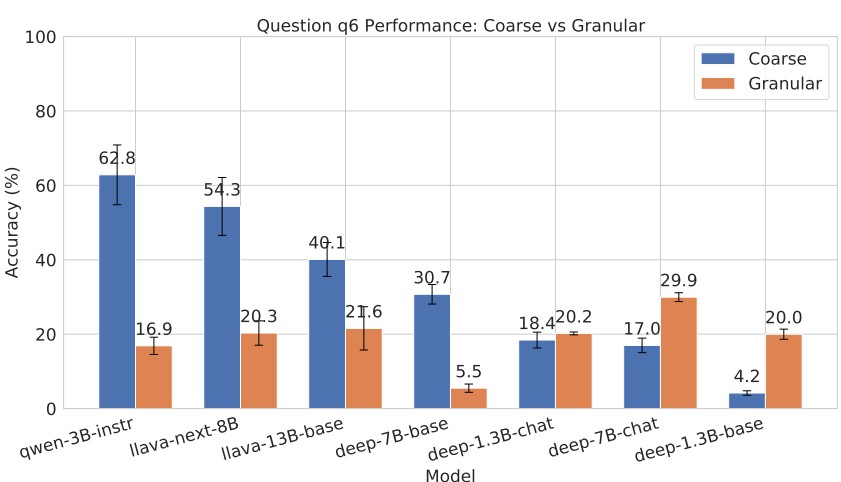

Figure 34: Mean and Standard Deviation of various models on Viewer-Scene direction task.

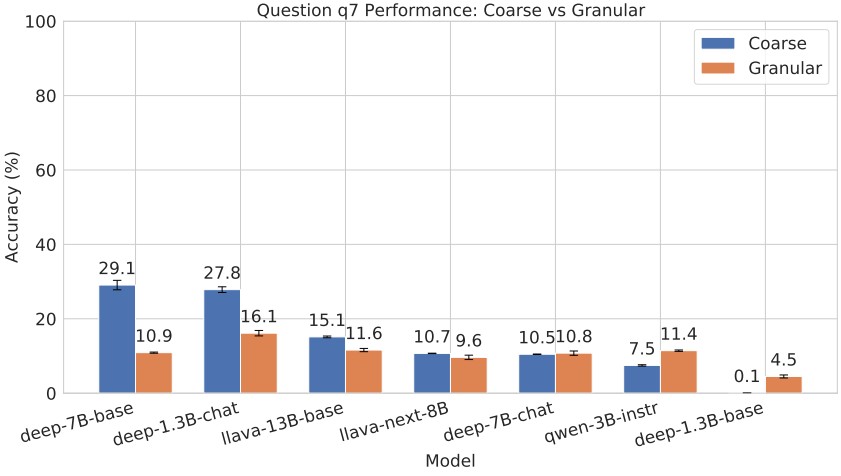

Figure 35: Mean and Standard Deviation of various models on the Canonical Orientation task.

## L  DETAILED ERROR ANALYSIS

We employ a systematic approach to identify consistent failure patterns rather than random prediction mistakes by analyzing the geometric relationship between predicted and ground truth orientations. Our methodology focuses on identifying repeatable, model-agnostic spatial reasoning deficits that indicate fundamental architectural limitations.

### SYSTEMATIC ERROR PATTERN IDENTIFICATION

We define systematic failure patterns as specific "Ground Truth → Predicted Answer" confusion pairs that satisfy following criteria: (1) occur in more than 5% of a model's incorrect predictions, and (2) are observed across at least 3 different model architectures. This threshold-based approach filters out infrequent, stochastic mistakes to focus on consistent failure modes that indicate underlying spatial reasoning deficits rather than random prediction errors. We categorize confusion patterns by their geometric and cognitive properties:

- **Perpendicular Confusion**: Systematic misclassification between parallel (0°–15°) and perpendicular (65°–95°) orientations, indicating categorical spatial representation
- **Directional Reversal**: Confusion between opposite directions (e.g., left↔right, toward↔away), suggesting directional processing failures
- **Angle Compression**: Tendency to predict intermediate angles when extreme angles are correct, indicating quantization of continuous spatial information
- **Frontal Bias**: Over-prediction of frontal/canonical orientations regardless of true orientation, likely reflecting training data distribution bias
- **Rotational Symmetry Confusion**: Systematic confusion between rotational equivalents (e.g., 90°↔270°, quarter vs. three-quarter turns)
- **Uncertainty Cascade**: Inappropriate uncertainty responses when spatial ambiguity should trigger systematic reasoning attempts

Below, Table 10 presents the complete systematic error analysis across all spatial reasoning dimensions evaluated in DORI. The patterns reveal fundamental limitations in how current MLLMs process and represent spatial information.

Table 10: Systematic Error Patterns Across All DORI Questions

| Question Type | True Answer | Predicted Answer | Error (%) | Pattern Type |
|---|---|---|---|---|
| View Parallelism | 0°–15° | 65°–95° | 20.5 | Perpendicular Confusion |
| View Parallelism | 65°–95° | 0°–15° | 10.3 | Perpendicular Confusion |
| Directional Facing | 30° left | 30° right | 8.3 | Directional Reversal |
| Directional Facing | 180° (away) | 0° (facing camera) | 14.7 | Directional Reversal |
| Inter-object Direction | 46–90° CCW | 46–90° CW | 5.7 | Directional Reversal |
| View Parallelism | 135°–180° | 65°–95° | 15.8 | Angle Compression |
| Single-axis Rotation | 90° | 45° | 7.6 | Angle Compression |
| Directional Facing | 30° left | 0° (facing camera) | 15.6 | Frontal Bias |
| Directional Facing | 30° right | 0° (facing camera) | 13.7 | Frontal Bias |
| Canonical Orientation | Cannot determine | No change needed | 19.5 | Frontal Bias |
| Viewer-scene Rotation | 270° | 90° | 13.1 | Rot. Symmetry Confusion |
| Viewer-scene Rotation | 180° | 90° | 11.6 | Rot. Symmetry Confusion |
| Single-axis Rotation | 0° | 90° | 9.2 | Rot. Symmetry Confusion |
| View Parallelism | 0°–15° | Cannot determine | 12.3 | Uncertainty Cascade |
| Canonical Orientation | Cannot determine | UNKNOWN | 23.5 | Uncertainty Cascade |
| Viewer-scene Rotation | 270° | Cannot determine | 12.0 | Uncertainty Cascade |

**Results:**   Our analysis reveals six systematic error categories affecting all evaluated models. The most severe is perpendicular confusion, where models systematically misclassify parallel (0°–15°)

Table 11: Component-wise Error Decomposition for Compound Rotations (Q4)

| Model | Order Swap (%) | Horizontal Acc. (%) | Vertical Acc. (%) | Both Wrong (%) |
|---|---|---|---|---|
| Qwen-3B-instr | 0.2 ± 0.1 | 22.7 ± 2.1 | 21.4 ± 1.9 | 55.9 ± 3.2 |
| LLaVA-13B-base | 3.8 ± 0.7 | 21.3 ± 1.8 | 19.5 ± 1.6 | 59.2 ± 2.9 |
| DeepSeek-7B-chat | 4.6 ± 0.8 | 20.4 ± 1.7 | 20.5 ± 1.7 | 59.1 ± 2.8 |
| DeepSeek-1.3B-chat | 4.2 ± 0.7 | 20.4 ± 1.6 | 19.7 ± 1.5 | 59.9 ± 2.7 |
| LLaVA-Next-8B | 4.9 ± 0.9 | 20.7 ± 1.8 | 19.6 ± 1.6 | 59.6 ± 2.9 |
| Magma-8B | 4.6 ± 0.8 | 20.1 ± 1.6 | 20.0 ± 1.6 | 59.9 ± 2.8 |
| DeepSeek-1.3B-base | 4.7 ± 0.8 | 19.9 ± 1.5 | 20.0 ± 1.6 | 60.1 ± 2.8 |
| DeepSeek-7B-base | 1.2 ± 0.3 | 19.7 ± 1.5 | 20.1 ± 1.6 | 60.3 ± 2.8 |
| **Average** | 3.5 ± 1.7 | 20.6 ± 0.9 | 20.1 ± 0.6 | 59.3 ± 1.4 |

and perpendicular (65°–95°) orientations with 20.5% and 10.3% error rates respectively, indicating coarse categorical rather than continuous angular encoding. Models also demonstrate consistent directional processing failures including left-right confusion (8.3%), front-back reversal (14.7%), and clockwise-counterclockwise errors (5.7%), suggesting fundamental directional encoding limitations. Additional patterns include angle compression where extreme positions are systematically predicted as intermediate values (7.6–15.8% error rates), frontal bias reflecting training data distribution effects (13.7–19.5% over-prediction of frontal orientations), rotational symmetry confusions particularly between quarter and three-quarter turns (9.2–13.1% error rates), and uncertainty cascade failures where models inappropriately handle spatial ambiguity (12.0–23.5% error rates). These systematic patterns across diverse architectures indicate fundamental limitations in current MLLM spatial processing mechanisms rather than model-specific deficits.

COMPONENT-WISE ERROR DECOMPOSITION

Compound rotation tasks (Q4 in DORI) present the most cognitively demanding spatial reasoning challenge, requiring models to track sequential 3D transformations around multiple axes. To understand the specific failure modes, we decompose compound rotation errors into orthogonal components that isolate different aspects of spatial transformation understanding. We analyze three distinct error categories that provide insight into different failure modes:

1. **Order Swap Errors**: Models correctly identify both rotation components but reverse their sequence (e.g., ground truth "90° Horizontal then 180° Vertical" predicted as "180° Horizontal then 90° Vertical"). This isolates sequence understanding from content understanding.

2. **Component Accuracy**: Percentage of predictions where individual rotation components (horizontal or vertical) are correctly identified regardless of the accuracy of the other component. This measures partial understanding capabilities.

3. **Complete Joint Failure**: Percentage of predictions where both rotation components are incorrectly predicted, indicating total breakdown of 3D spatial reasoning.

Table 11 presents the complete component-wise error decomposition across all evaluated models, revealing distinct failure patterns and architectural effects.

**Results:** The component-wise decomposition reveals a fundamental dissociation between sequence processing and spatial reasoning in MLLMs. While order swap errors remain low across all models (mean: 3.5%), indicating competent instruction following, approximately 60% of predictions fail on both rotation components, representing near-complete breakdown of 3D spatial processing. . Models demonstrate axis-agnostic processing (horizontal vs. vertical accuracy differential: 0.7%), unlike human embodied cognition, suggesting identical mechanisms for all rotational transformations rather than specialized processing pathways. Notably, instruction-tuned models significantly outperform larger base models—Qwen-3B-instr achieves superior performance across all metrics despite smaller parameter count, and DeepSeek-1.3B-chat outperforms DeepSeek-7B-base with 5.4x fewer parameters.

These findings indicate three critical limitations: (1) models can manipulate rotation symbols but lack geometric transformation mechanisms, (2) the consistent 60% joint failure rate across diverse architectures suggests fundamental rather than model-specific deficits, and (3) training methodology appears more crucial than parameter scaling for spatial reasoning capabilities, suggesting targeted training approaches may be more effective than architectural scaling.

SOFT ACCURACY CALCULATION

Soft accuracy metrics were introduced to provide a more nuanced evaluation of models' orientation understanding capabilities and help distinguish between models that are completely wrong versus those that have an approximate understanding of orientation concepts. Unlike standard binary accuracy that only awards points for exact matches, soft accuracy awards half points (0.5) for answers that are partially correct or adjacent to the ground truth. To calculate such accuracies, we implement carefully designed spatial tolerance thresholds and logical equivalences. Soft accuracy is only calculated for the fine-grained questions that typically demand a precise metric response.

- For **View Parallelism**, Predictions within $\pm 45°$ of the ground truth angle receive partial credit. This threshold captures predictions in adjacent sectors while excluding opposed orientations.
- For **Directional Facing**, half points are awarded exclusively for mirror-image confusions between "30 degrees left" and "30 degrees right". We intentionally do not extend partial credit to other angular errors, preserving the specificity of directional understanding assessment.
- For **Single-axis Rotatoin**, a $45°$ tolerance window applies to predicted rotations $(0°, 45°, 90°, 135°, 180°)$. This allows credit for adjacent discrete positions while maintaining distinction between major orientation categories. For instance, predicting $45°$ when the correct answer is $90°$ would not qualify, but a $135°$ prediction for $180°$ ground truth would receive partial credit.
- In **Compound Rotation**, partial credit is awarded if either the horizontal or vertical rotation component is correct in multi-axis transformations. In this "X then Y" rotation sequence response, we parse both components separately. For example, prediction of "$90°$ horizontal then $0°$ vertical" would receive 0.5 points for either correct component when compared to the ground truth "$90°$ horizontal then $180°$ vertical"
- For **Inter-object Direction**, half points are given for adjacent magnitude ranges, but only if the direction (clockwise vs. counterclockwise) matches. For instance, if the ground truth was "0 to 45 degrees clockwise" and the prediction was "46 to 90 degrees clockwise", we award 0.5 points. However, if the prediction was "46 to 90 degrees counterclockwise," it would earn 0 points despite similar magnitude
- For **Viewer-scene direction**, the soft accuracy specifically addresses confusion between opposite rotational directions. For example, if an object has rotated $90°$ clockwise between images, but the model reports $270°$ clockwise (which is equivalent to $90°$ counterclockwise), it receives 0.5 points. No partial credit is given for other angle confusions.
- For **Canonical Orientation**, the soft accuracy addresses confusion in the order of operations. For example, if an image requires rotation followed by flipping to restore its canonical orientation, but the model suggests flipping followed by rotation, it receives 0.5 points.

**Results.** Figs. 36, 37, 38, 39, 40, 41, 42, compares standard (hard) vs. soft accuracy on DORI questions.

Figures 36 and 37 reveal that for View Parallelism and Directional Facing tasks, soft accuracy provides no benefit over standard accuracy, with identical performance metrics across all models. This indicates that when models err on these fundamental orientation tasks, they tend to make categorical mistakes rather than near-miss approximations. The lack of improvement suggests that errors in these tasks stem from fundamental misunderstandings rather than subtle misjudgments.

In contrast, the **Single-axis Rotation** task (Figure 38) shows the most substantial gains under soft accuracy metrics, with improvements ranging from 8.6% to 17.7% across models. DeepSeek-7B-Chat achieves the most dramatic improvement, a remarkable 17.7% gain. LLaVA-Next-8B improves of +11.5%, while Qwen-3B-Instruct shows improvement from + 16.4%. These substantial gains suggest that while models often fail to identify the exact rotational angle, they frequently select adjacent angular categories, demonstrating partial understanding of rotational relationships.

For Compound Rotation (Figure 39), all models except Qwe2.5-3B-Instruct (0.5% improvement) show notable improvements under soft accuracy metrics. Both LLaVA-7B-Chat and LLava-Next-8B improves from by +14.6%, indicating that models often correctly identify one of the two rotation components (horizontal or vertical) while missing the other. This partial success highlights both the inherent complexity of multi-axis rotations and the models' fragmentary grasp of compound transformations.

The Inter-object Direction task (Figure 40) shows similar amount of soft accuracy gains across all models, with improvements ranging from 7.7% to 17.1%. DeepSeek-7B-Chat improves from 12.7% to 29.8%, more than doubling its effective performance. This suggests that models often select directionally appropriate answers that fall into adjacent angular ranges, indicating a coarse understanding of relative orientations despite lacking precise angular discrimination.

For Viewer-scene Direction (Figure 41), soft accuracy provides moderate improvements, with DeepSeek-7B-Chat showing the largest gain. The comparatively smaller improvements here suggest that models correctly identify rotational changes, with fewer "near miss" responses than in other tasks.

Canonical Orientation (Figure 42) shows minimal improvement under soft accuracy metrics. The small gains observed suggest that models are rarely confused in the order of operations. When they fail on canonical orientation tasks, they typically misidentify the necessary operations entirely rather than simply reversing their order. This indicates a more fundamental gap in understanding canonical object positioning rather than mere sequencing errors.

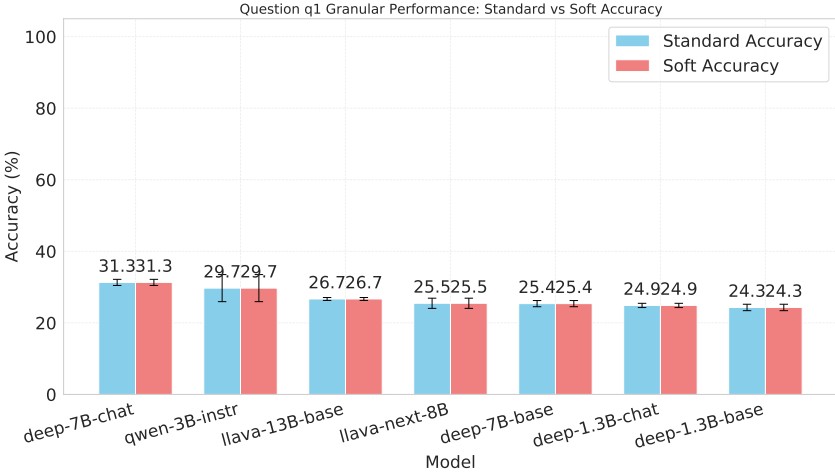

Figure 36: Comparing the mean and standard deviation of soft vs. standard (hard) accuracy on View Parallelism task. See discussion Sec. L.

Fig. 43 summarizes the relative gains of soft accuracy across all tasks and models. Notably, this more lenient metric only helps some questions (Fig. 38 to 42). The heatmap reveals that Single-axis Rotation, Compound Rotation, and Inter-object Direction tasks benefit most from soft accuracy metrics, with improvements frequently exceeding 10%. This pattern suggests that rotational and relational orientation understanding in current MLLMs exists on a spectrum rather than in binary states of correctness. In addition, the highest gain reported was for Single-axis Rotation, but is significantly below human performance using standard accuracy (18% vs. avg of about 30%). This shows that even when given an advantage, these models still fall significantly below human ability.

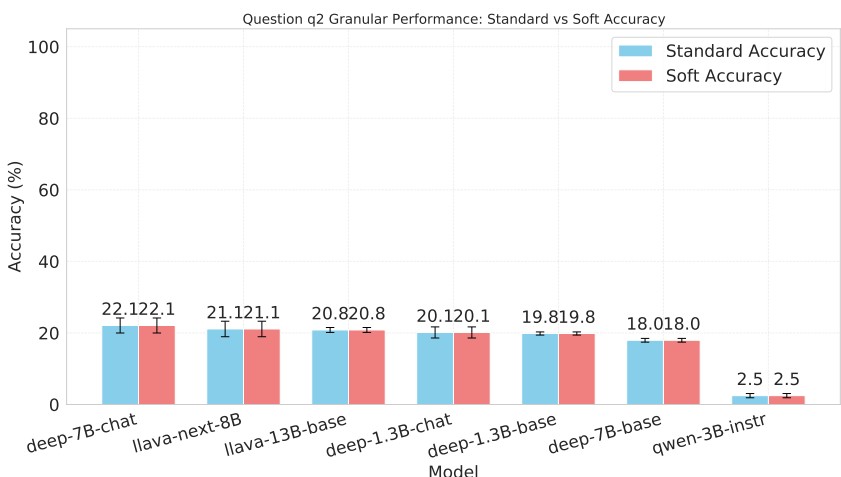

Figure 37: Comparing the mean and standard deviation of soft vs. standard (hard) accuracy on Directional Facing task. See discussion Sec. L.

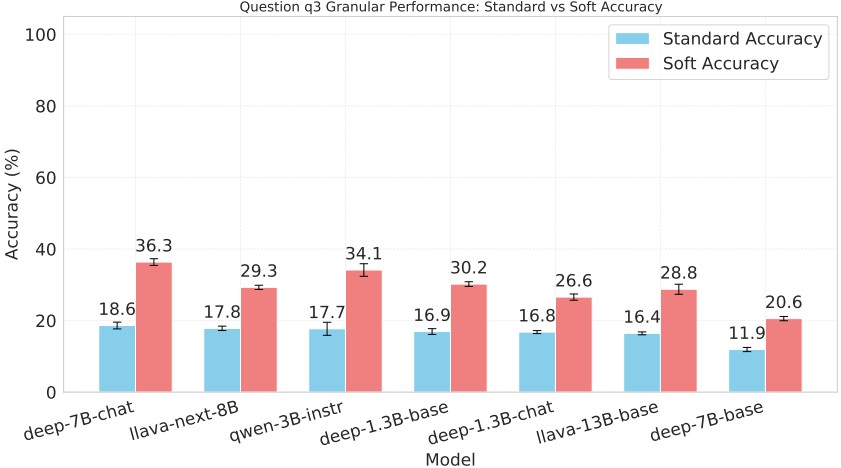

Figure 38: Comparing the mean and standard deviation of soft vs. standard (hard) accuracy on Single-axis Rotation task. See discussion Sec. L.

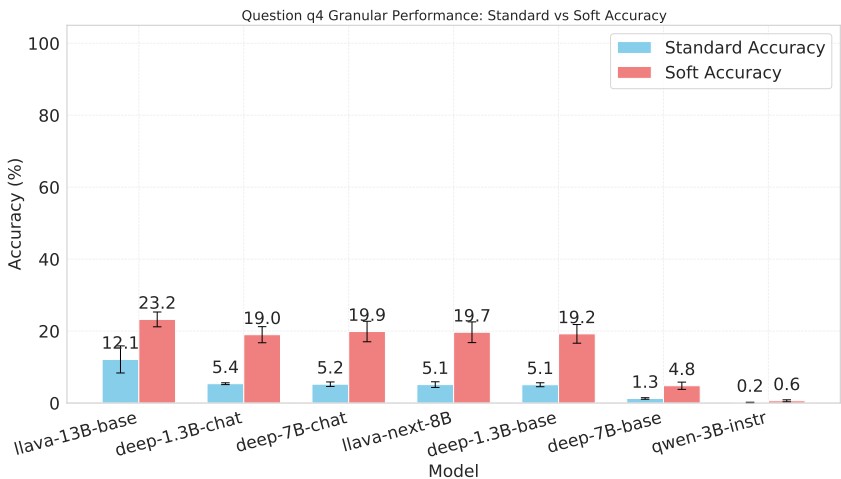

Figure 39: Comparing the mean and standard deviation of soft vs. standard (hard) accuracy on Compound Rotation task. See discussion Sec. L.

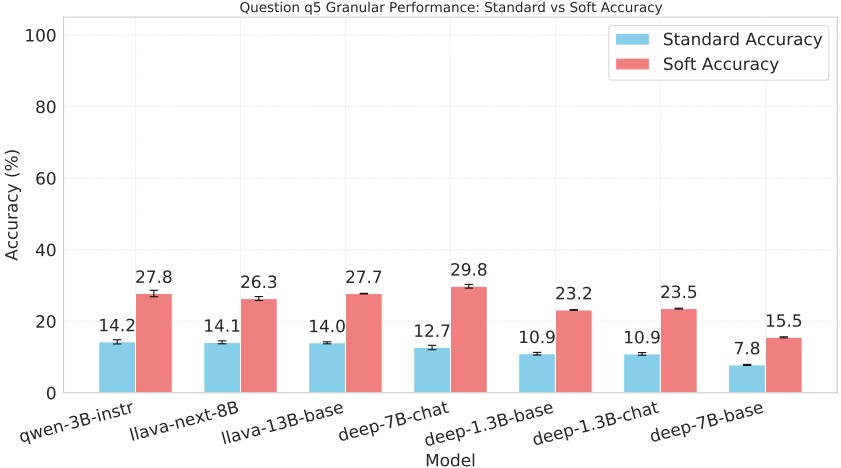

Figure 40: Comparing the mean and standard deviation of soft vs. standard (hard) accuracy on Inter-object Direction task. See discussion Sec. L.

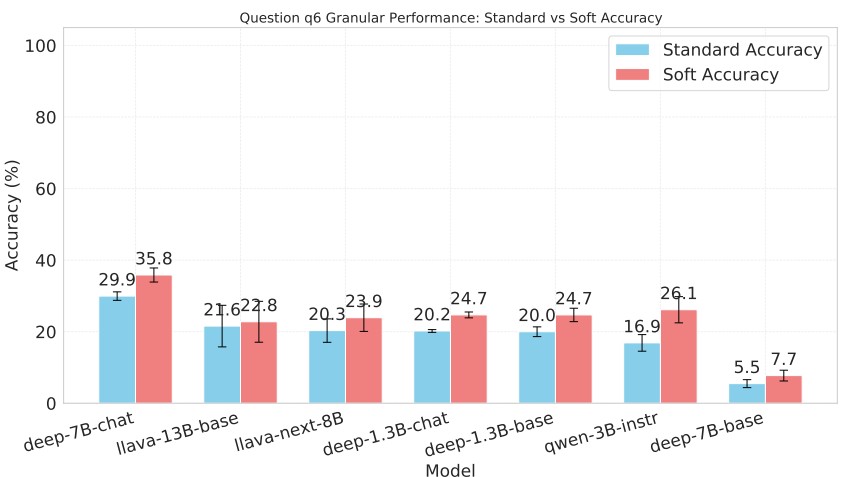

Figure 41: Comparing the mean and standard deviation of soft vs. standard (hard) accuracy on Viewer-scene Direction task. See discussion Sec. L.

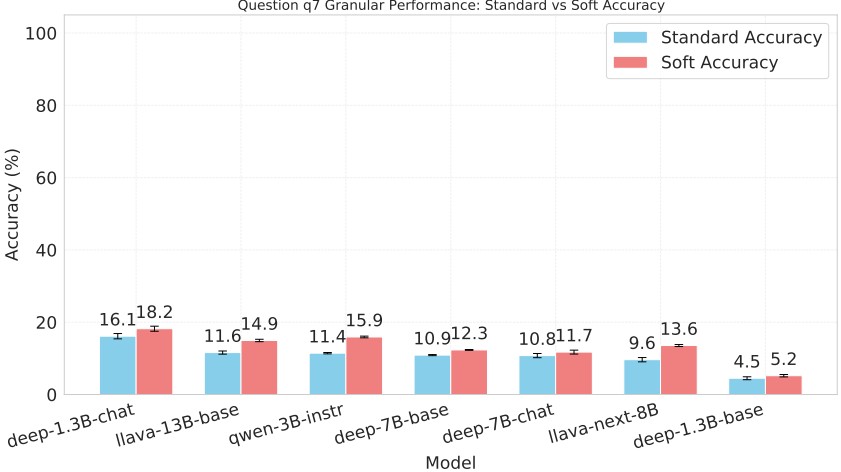

Figure 42: Comparing the mean and standard deviation of soft vs. standard (hard) accuracy on Inter-object Direction task. See discussion Sec. L.

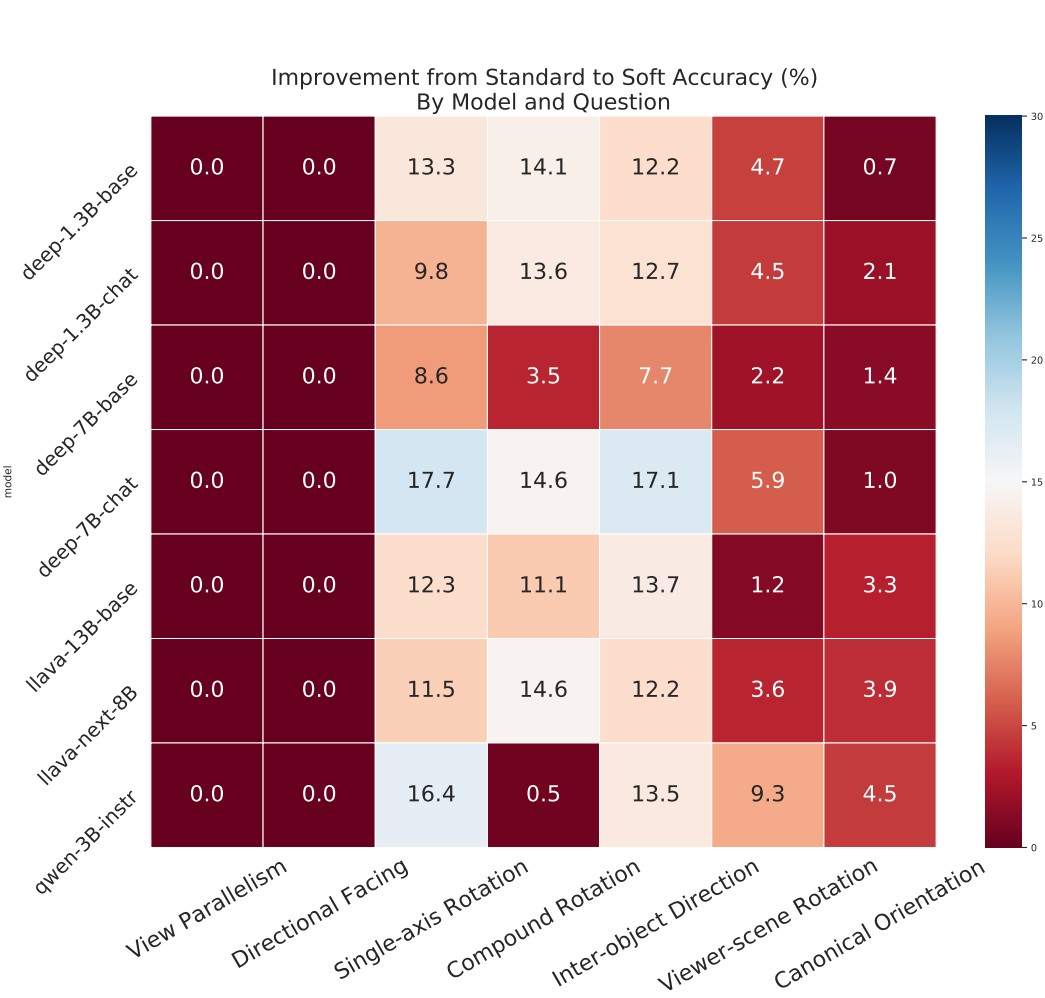

Figure 43: Relative gain from using Soft accuracy rather than Standard (hard) accuracy per question. See L for discussion

Table 12: Performance of Base and Fine-tuned Models across DORI Question Types with C being Coarse and G being Granular

| | Frontal Alignment | | | | Rotational Transformation | | | | Relative Orient. | | | | Canonical Orient. | | Avg. | |
| | View Parallel. | | Dir. Facing | | Single-axis Rot. | | Compound Rot. | | Inter-Obj. Dir. | | Viewer-scene Dir. | | | | | |
| | C | G | C | G | C | G | C | G | C | G | C | G | C | G | C | G |
|---|---|---|---|---|---|---|---|---|---|---|---|---|---|---|---|---|
| Base Model | 57.7 | 34.8 | 27.2 | 0.0 | 22.9 | 20.4 | 37.7 | 5.6 | 7.3 | 13.7 | 77.7 | 22.0 | 0.0 | 0.0 | 32.9 | 13.7 |
| + Finetuned w/DORI | 87.8 | 72.8 | 81.9 | 2.6 | 76.8 | 76.1 | 64.5 | 17.3 | 79.2 | 60.6 | 99.0 | 84.9 | 42.9 | 37.1 | 76.0 | 50.2 |

## M    LoRA Finetuning

We create a finetuning dataset with 27K real + synthetic random samples from our dataset and the remaining 7K for evaluation. We utilize the Qwen2.5-VL-3B with LoRA (Hu et al., 2022) to highlight the performance improvement when using our dataset, this can be seen in Table 12. As shown, finetuning via LoRA results in a 37-46% gain, demonstrating that better alignment to the capabilities measured in our task can result in greatly improved performance.

## N    Out-Of-Pretraining Analysis

Most MLLMs have closed pretraining datasets, i.e., it is not possible to be completely sure what they were trained on. However, to provide insight into what the MLLM model might have learned (i.e., if it saw similar images in the pretraining dataset) we use the approach described in (Teterwak et al., 2025) to filter images in COCO and Cityscapes into those that were likely seen during pretraining. Specifically, we measured the cosine similarity between text features representing the object our question referred to (e.g., "a photo of a person" for a question asking about the orientation of a person) and the image using LLaVa-13B. Those with high similarity (using a 0.19 threshold for COCO and 0.18 for Cityscapes) were removed, leaving only images that were not learned well by the model during pretraining. On COCO we evaluated performance on person images, which accounted for 547 images in the Directional Facing questions, of which 253 were removing (leaving 294 images). On Cityscapes Directional Facing, we filtered based on car questions, resulting in removing 122 of 342 car images, leaving 220 images. Cars and person categories were selected as they were common objects in their respective datasets. We refer to these splits as ALL, which includes questions that have every car or person image, and Out-Of-Pretraining (OOP) for the images that remained after our filtering process. We then compare the performance on a number of models armed with these splits, as seen in Table 13.

Armed with these splits, we compared performance on LLava-13B, LLava-34B, Yi-VL-6B and Deepseek-7B, shown below. Interestingly, the OOP results are generally better than ALL, which seems counterintuitive at first glance. However, we suspect that this is an effect of being pretrained with a different objective than one focused on orientation. For example, if a model was pretrained to align images to alt-text, which give a high level description of the image that often does not include orientation information, then when the model saw similar images, it naturally assumes that the goal is to match to a high-level description and extracts features accordingly, essentially overfitting to that task. That said, there are many confounding factors that could provide alternative explanations, and is an interesting avenue for exploration in future work.

Table 13: Out-Of-Pretraining (OOP) performance when evaluated on the COCO and Cityscapes dataset.

| COCO | All | OOP | All | OOP |
|---|---|---|---|---|
| Category (Person) | Coarse | Coarse | Granular | Granular |
| LLava-v1.6-13B | 23.0 | 25.1 | 20.8 | 21.0 |
| LLava-v1.6-34B | 42.6 | 41.3 | 39.6 | 39.6 |
| Yi-VL-6B | 30.3 | 28.6 | 33.5 | 33.1 |
| Deepseek-Base-7B | 17.7 | 17.5 | 17.7 | 17.9 |
| Cityscapes | All | OOP | All | OOP |
| Category (Car) | Coarse | Coarse | Granular | Granular |
| LLava-v1.6-13B | 22.6 | 25.7 | 16.7 | 18.1 |
| LLava-v1.6-34B | 36.5 | 41.1 | 21.6 | 21.7 |
| Yi-VL-6B | 25.6 | 29.1 | 21.6 | 22.1 |
| Deepseek-Base-7B | 17.5 | 18.5 | 16.9 | 17.1 |

Instructions

Thank you for participating in our pilot survey on understanding Object Orientation!

We are conducting this survey to determine how people perform on questions related to orientation for objects and scenes.

For the image(s) shown below, please answer the question to the best of your ability.

There may be one or two images depending on the question.

Please **take your time** and do the HIT's diligently. We will be monitoring for rapid random clicks and will reject your HIT if we find evidence for rapid clicking.

"• Examine the two images shown side by side
• Each image contains one object
• Compare how the object is positioned in both images
• **Horizontal rotation:** (like a cartwheel coming out of the screen)
• **Vertical rotation:** (like a ballerina spinning turning clockwise)
• **Rotation sequence:** A series of rotations applied in order (first rotation, then second rotation)""

**Examples:**

• If an object appears to have moved left or right between images, select ""Only Vertical""
• If an object appears to have moved up or down (like nodding), select ""Only Horizontal""
• If an object has rotated both horizontally and vertically, select ""Both Horizontal and Vertical""
• If the object appears identical in both images with no rotation, select ""No Change""
• If you cannot clearly determine the type of rotation, select ""Cannot be determined"" "

"Between the two images, which if the following **kind of rotation** has the object undergone?"

Question 1)

Both Horizontal and Vertical ○

Cannot be determined○

Only Horizontal○

Figure 44: An example of the high level instructions shown for the Human Evaluation

"The natural front of the object based on its inherent structural features. For example:

• For a person: the body
• For a building: the front door or main entrance
• For a vehicle: the headlights or front grille
• For an animal: the face/head ""

**Examples:**

• If you see a car with its headlights directly pointing toward the camera, select ""Toward the camera""
• If you see a person with their back to the camera, select ""Away from the camera"""

" **From the camera's viewpoint** , indicate which direction **Object A's** front-facing surface is oriented?"

Question 1)

Toward the camera ○

Cannot be determined○

Away from the camera○

Figure 45: An example of a sample for the coarse-level VQA for the Directional Facing task in DORI shown for the Human Evaluation

## O  HUMAN EVALUATION INTERFACE

An example of the high level instructions shown for this task can be seen in Fig. 44, examples of questions are shown in Fig. 45 and Fig. 46

"• **Rotation:** Movement of an object along a vertical axis (like hands of a clock)
• **Stationary camera:** The camera position and angle remain fixed between both images
• **Vertical axis:** An imaginary line running up and down through the object .""

**Examples:**

• **Clockwise rotation:** Movement in the direction of a clock's hands (from your perspective looking at the object)
• Objects that are symmetrical like a ball or a box would be considered ""Cannot be determined"" "

"Between the two images, by **how many degrees CLOCKWISE has the object rotated?** "

### Question 1)

90 degrees ○
180 degrees ○
Cannot be determined ○

### Question 2)
90 degrees ○

Figure 46: An example of a sample for the coarse-level VQA for the Viewer-Scenen Direction task in DORI shown for the Human Evaluation

