# OpenReview forum: "Right Side Up? Disentangling Orientation Understanding in MLLMs with Fine-grained Multi-axis Perception Tasks"
_ICLR.cc/2026/Conference — Submitted to ICLR 2026_

### Official Review · Reviewer_X2sf · 2025-10-27

**Soundness:** 3
**Presentation:** 3
**Contribution:** 3
**Rating:** 8
**Confidence:** 4

**Summary:**

The paper introduces a hierarchical benchmark that evaluates MLLM’s ability to understand and reason about orientation. The paper evaluates multiple MLLMs on the benchmarking and shows poor performance.

**Strengths:**

* The motivation behind the curated dataset is well justified. The curated dataset is diverse and contains common objects that shall be viewed by MLLM during pre-training.
* The paper has a clear presentation of the dataset and complete evaluation of the popular open-source and closed-source MLLMs.
* This dataset has a potential be improved to 3D. If so, this will be very beneficial for active learning and robotic pre-training, etc..

**Weaknesses:**

* I have a concern about whether the "counter-clockwise" and "clockwise" are consistently defined. In a 3D setting, when talking about rotation, we always need to specify the direction of the z-axis. But such information is not provided in the dataset.
* I also have a concern about whether "face toward" is well defined. This clearly requires the described object to have a "face" that is visually decidable. If the object is a human, it is simple. But for other objects like tables or sofa, this language may not apply.
* This is not a weakness. Since the tasks proposed in the paper mostly require 3D reasoning, it may make the dataset stronger if 3D point cloud or depth are also provided for those simulated images.

**Questions:**

No question.

---

> ### Author Response · Authors · 2025-11-21
>
> We thank the reviewer for their comments which we have used to revise our paper.  We respond to individual comments below.
>
> > 1. I have a concern about whether the "counter-clockwise" and "clockwise" are consistently defined. In a 3D setting, when talking about rotation, we always need to specify the direction of the z-axis. But such information is not provided in the dataset.
>
> We explicitly define the z-axis in our prompts and consistently define the clockwise and counter-clockwise directions. This can be seen in Figure 19 for the Single-Axis Rotation task in the appendix where we define the z-axis and consistently define clockwise and counter-clockwise. We provide the example below
>
>
> *EXAMPLES:- If the object is already facing the camera the answer is "0 degrees"- If the object's front is facing to your right a "90 degrees" clockwise rotation would make it face the camera- If the object's front is at a 45-degree angle to your right a "45 degrees" clockwise rotation would make it face the camera- If the object's front is at a 135-degree angle to your right a "135 degrees" clockwise rotation would make it face the camera- If the object's front is facing away from the camera a "180 degrees" clockwise rotation would make it face the camera- If the object has no clear front or its orientation cannot be determined select "Cannot be determined"*
>
> For further clarification, we have included a glossary in Appendix J.2 and we have added examples of clockwise and counter-clockwise samples in Figure 28.
>
> > 2. I also have a concern about whether "face toward" is well defined. This clearly requires the described object to have a "face" that is visually decidable. If the object is a human, it is simple. But for other objects like tables or sofa, this language may not apply.
>
> DORI accounts for ambiguities by including the option  “Cannot be determined.” Figure 1 and 27 both provide examples, but other objects with unclear frontality like pizza, bowl, etc all have the same behavior as mentioned in our detailed construction process Appendix A. We have modified our discussion in Section 3.1 under the Frontal Alignment discussion and Section 3.2 of our main paper to note this as well to avoid any confusion. We also refer the reviewer to Figure 8 which shows an example of the Canonical Orientation question which contains some images that have the label “Cannot be determined” for instance for the images of rocks.
>
>
>
> > 3. This is not a weakness. Since the tasks proposed in the paper mostly require 3D reasoning, it may make the dataset stronger if 3D point cloud or depth are also provided for those simulated images.
>
> While for some datasets like COCO this is not possible as they do not provide 3D information, we are happy to include the object files for questions that include datasets like OmniObject3D to support obtaining the 3D point cloud.  Additionally, we will release the 3D depth information for samples from OmniObject3D and Shapenet.

---

### Official Review · Reviewer_9Ynm · 2025-10-29

**Soundness:** 3
**Presentation:** 3
**Contribution:** 3
**Rating:** 6
**Confidence:** 3

**Summary:**

This paper presents the DORI benchmark, developed to specifically evaluate how well Multimodal Large Language Models (MLLMs) understand object orientation. DORI uses a cognitive science-informed approach, assessing orientation perception across multiple facets including how objects face the viewer, how they change with rotation, their orientation relative to other objects or viewpoints, and their typical 'right-side-up' state. The evaluation includes both basic categorical questions and more demanding fine-grained angular questions. It is applied to a substantial amount of real and synthetic images (over 13k images from 14 sources) with structured prompts. Experiments involving 18 MLLMs indicated difficulties in this dataset, particularly in making precise orientation judgments versus simpler classifications. Notable performance declines happen when tasks required understanding rotations or shifts in perspective. The findings suggest current models may lack robust internal mechanisms for representing and reasoning about object orientation.

**Strengths:**

1. The most noticeable contribution is the dataset collected. The benchmark's hierarchical structure, decomposes orientation related questions into four dimensions, which care frontal alignment, rotational transformations, relative orientation and canonical orientation, is quite inspiring. The inclusion of both coarse and fine-grained questions allows a more comprehensive assessment of model proficiency

2. The paper effectively identifies and addresses a limitation in existing MLLM, which is the lack of ability to assess object orientation understanding, separate from general spatial reasoning. This dataset helped to evaluate more detailed object orientation understanding ability in MLLM.

3. DORI is constructed from a quite diverse set of images (13,652 images from 14 sources), which include both real-world and synthetic data. The evaluation is conducted across 18 different MLLMs, providing a quite holistic benchmark evaluation.

**Weaknesses:**

1. The presentation of the paper can be improved. Limited examples are provided for the VQA questions involving canonical orientation. I have remaining concerns on these types of questions since canonical orientation or frontal alignment itself might remain inherently ambiguous for certain object types, like symmetric ones. This might introduce noise into the ground truth and evaluation, and I am interested in seeing how they are addressed more detailedly.

2. Another limitation is the absence of empirical validation showing that performance on DORI actually correlates with MLLM capabilities in real-world applications like robotic manipulation or autonomous navigation. The paper claims relevance but doesn't demonstrate a predictive link between benchmark scores and success on applied tasks requiring orientation understanding.

3. The paper's writing can be improved. Some tables exceed text width. In main experiments tables, using vertical axis to separate different task is recommended. Overall, there are some minor presentation issues in the paper.

**Questions:**

Please refer to the weakness.

Will this dataset be released?

---

> ### Author Response · Authors · 2025-11-21
>
> We thank the reviewer for their comments which we have used to revise our paper.  We respond to individual comments below.
> > 1. The presentation of the paper can be improved. Limited examples are provided for the VQA questions involving canonical orientation. I have remaining concerns on these types of questions since canonical orientation or frontal alignment itself might remain inherently ambiguous for certain object types, like symmetric ones. This might introduce noise into the ground truth and evaluation, and I am interested in seeing how they are addressed more detailedly.
>
> DORI accounts for ambiguities by including the option  “Cannot be determined.” Figure 1 and 27 both provide examples, but other objects with unclear frontality like pizza, bowl, etc all have the same behavior as mentioned in our detailed construction process Appendix A. We have modified our discussion in Section 3.1 under the Frontal Alignment discussion and Section 3.2 of our main paper to note this as well to avoid any confusion. We also refer the reviewer to Figure 8 which shows an example of the Canonical Orientation question which contains some images that have the label “Cannot be determined” for instance for the images of rocks.
>
> > 2. Another limitation is the absence of empirical validation showing that performance on DORI actually correlates with MLLM capabilities in real-world applications like robotic manipulation or autonomous navigation. The paper claims relevance but doesn't demonstrate a predictive link between benchmark scores and success on applied tasks requiring orientation understanding.
>
> We would like to point the reviewer to the other datasets we have evaluated on, namely BLINK, 3DSRBench and SAT, these results can be found in Table 4 of the main paper and we have copied it below for easy reference. These different dataset demonstrate a connection between spatial reasoning and real-world task performance. Datasets like BLINK showcase how MLLMs misunderstand object rotation which leads to incorrect action relevant predictions which are required for application tasks like grasping. For 3DSRBench it highlights how MLLMs with poor 3D spatial understanding can underperform on manipulation and navigation style queries which are essential for autonomous navigation. Finally for SAT, which evaluates on tasks related to frame alignment, rotation handling and relational understanding, highlight for MLLMs key tasks that are prerequisites for robotics and embodied tasks.
>
>
> | Model        |          BLINK          |                 |                 |              3DSRBench              |                 |                 |                 |  SAT  |
> |--------------|---------------------------|-----------------|-----------------|-------------------------------------|-----------------|-----------------|-----------------|-------|
> |              | Multi-View Reasoning      | Visual-Corresp. | Relative-Depth  | Orient.                             | Multi-Obj. View-To-Obj.    | Multi-Obj. Parallel        | Multi-Obj. Same Dir.       |       |
> | Base Model   | 42.9                      | 25.3            | 64.1            | 32.5                                | 11.4            | 11.4            | 46.8            | 51.3  |
> | +Finetuning  | **45.1**                  | **26.5**        | **65.3**        | **38.6**                            | **14.0**        | **38.1**        | **50.3**        | **63.3** |
>
>
>
> > 3. The paper's writing can be improved. Some tables exceed text width. In main experiments tables, using vertical axis to separate different task is recommended. Overall, there are some minor presentation issues in the paper.
>
> We have added the vertical axes to our main experiments tables and will make sure the tables are within the text width and will update the table and comment once this is done.
>
> > 4. Will this dataset be released?
>
> Yes we will release this dataset

---

### Official Review · Reviewer_Vs7f · 2025-11-01

**Soundness:** 2
**Presentation:** 3
**Contribution:** 3
**Rating:** 4
**Confidence:** 5

**Summary:**

This work introduces DORI, a benchmark designed to evaluate the orientation perception ability of current multimodal large language models (MLLMs). DORI comprises 13,652 images from 14 sources, forming a total of 33,656 samples. It assesses four key aspects of object orientation understanding: frontal alignment, rotational transformations, relative directional relationships, and canonical orientation comprehension. The results show that even the best-performing models achieve only 54.2% accuracy on coarse-level tasks and 33.0% on fine-grained orientation judgments, with performance degrading significantly on tasks involving reference frame shifts or compound rotations.

**Strengths:**

1. The proposed benchmark addresses an important problem.
2. The writing is generally clear and easy to follow.
3. The related work section is detailed and clearly explains the limitations of existing benchmarks.
4. The proposed benchmark is novel in terms of its practical usability.

**Weaknesses:**

1. Figure 1 is not very clear and does not effectively convey the definitions of the four task categories. In particular, the examples for rotational transformation and relative orientation appear very similar, making it difficult to distinguish between them.
2. Although the paper cites many works from related fields such as cognitive science to explain how humans understand rotation, the definitions of the four subproblems lack clear logic and structure. The relationships among them are not well articulated, leaving it unclear whether the proposed categorization is both complete and necessary.
3. For some objects, the front face is inherently ambiguous (e.g., a table). Although the paper mentions that specific prompt designs are used to define the front face for tested models, such strategies cannot fully resolve these ambiguities. This raises concerns about the correctness and answerability of certain questions.
4. Based on the above, I suspect that some samples may be ambiguous. However, the paper does not describe any quality control process to ensure dataset accuracy. For a benchmark, it is generally expected that every sample be manually verified to guarantee correctness.
5. While the paper evaluates several state-of-the-art models, it omits important models such as the InternVL series, and for the Qwen family, only the 3B variant is tested without including larger models.
6. Lines 418–419: The observed difference may stem from variations in training data, so this conclusion should not be drawn too hastily.
7. Table 5: Please correct the label from “GPT-4 O” to “GPT-4o.”
8. The paper claims that the proposed systematic approach isolates orientation understanding from scene perception skills and minimizes confounding factors such as object recognition difficulty, scene clutter, linguistic ambiguity, and contextual distractions that affect existing benchmarks. However, no experiments or examples are provided to substantiate these claims.

**Questions:**

1. In Section 3.1, the definition of the viewing plane is not clearly explained.

---

> ### Author Response · Authors · 2025-11-21
>
> We thank the reviewer for their comments which we have used to revise our paper.  We respond to individual comments below.
>
> > 1. Figure 1 is not very clear and does not effectively convey the definitions of the four task categories. In particular, the examples for rotational transformation and relative orientation appear very similar, making it difficult to distinguish between them.
>
> Thank you for your feedback. We have adjusted Figure 1 to better distinguish these categories. Rotational Transformation targets single-object transformations with specific axes, while Relative Orientation targets multi-object spatial relationships around anchors. We are happy to take any suggestions on ways to improve this figure.
>
> > 2. Although the paper cites many works from related fields such as cognitive science to explain how humans understand rotation, the definitions of the four subproblems lack clear logic and structure. The relationships among them are not well articulated, leaving it unclear whether the proposed categorization is both complete and necessary.
>
>
>
> Based on established frameworks in cognitive neuroscience, DORI decomposes object orientation comprehension into four fundamental dimensions that reflect distinct neural mechanisms and cognitive processes.
>
> Cognitive studies confirm V1/V2 directional perception precedes complex spatial reasoning [A]. These studies also suggest that infants’ visual cortex development enables directional perception (e.g., "Which way does the sensor point?") necessary for grasping and object manipulation, which maps to our Frontal Alignment dimension.
>
> Certain works on Mental Rotation also suggest that at ages 4-5, premotor and parietal cortex support mental rotation; imagining objects transforming across axes [B]. Our Rotational Transformation evaluates this capability in models (e.g., "Rotate 90° horizontal + 45° vertical").
>
> By ages 6-7, hippocampal-pariental networks enable allocentric orientation reasoning to understand external frame relations and not just egocentric viewpoints [C]. DORI tests this reasoning of orientation relative to each other e.g., "chairs facing each other around a table").
>
> Our Canonical Orientation evaluation, requiring abstract reasoning, is associated with the dimension involving semantic reasoning about objects’ “correct” or functional orientations, (e.g., "which way is the bowl right-side-up?") [D], supported by prefrontal cortex and inferotemporal cortex development in kids (~7-9 years).
>
> This categorization is complete and necessary as they span a full development arc (1 year to 7-9 years). The irreducibility of each dimension also necessitates the categorization as each alone cannot access the other. Our work also ensures this separation due to their hierarchy; frontal identification -> transformation (manipulate) -> relational (coordinate) ->  semantic (understand).  Below we summarize this mapping.
>
> | Dimension                 | Cognitive Stage        | Neural Substrate                     | Age       | Aspect Tested                              |
> |---------------------------|-------------------------|----------------------------------------|-----------|----------------------------------------------|
> | Frontal Alignment         | Basic Perception        | Visual cortex (V1/V2) [A]              | ~1 year   | Single object directional perception         |
> | Rotational Transformation | Mental Rotation         | Premotor/parietal cortex [B]           | ~4–5 yrs   | Single object transformation                 |
> | Relative Orientation      | Allocentric Reasoning   | Hippocampus/parietal [C]               | ~6–7 yrs  | Multi-object relationship                    |
> | Canonical Orientation     | Semantic Understanding  | Inferotemporal/prefrontal [D]          | ~7–9 yrs  | Semantic/physical object properties          |
>
>
>
> We have added this discussion to App. A of our paper and referred the reader to it from Sec. 3.1 of our main paper.
>
> [A] Otten et al. (2025). The maturation of infant and toddler visual cortex neural activity and associations with fine motor performance. Dev. Cogn. Neurosci. 71, 101501.
>
> [B] Frick et al. (2014). Development of mental transformation abilities. Trends Cogn. Sci., 18(10), 536-542.
>
> [C] Pullano & Foti (2022). The development of human navigation in middle childhood: A narrative review. Brain Sci., 12(8), 1097.
>
> [D] Dumontheil (2014). Development of abstract thinking during childhood and adolescence: the role of rostrolateral prefrontal cortex. Dev. Cogn. Neurosci., 10, 57-76.

---

> ### Author Response · Authors · 2025-11-21
>
> > 3. For some objects, the front face is inherently ambiguous (e.g., a table). Although the paper mentions that specific prompt designs are used to define the front face for tested models, such strategies cannot fully resolve these ambiguities. This raises concerns about the correctness and answerability of certain questions.
>
> DORI accounts for ambiguities by including the option  “Cannot be determined.” Figure 1 and 27 both provide examples, but other objects with unclear frontality like pizza, bowl, etc all have the same behavior as mentioned in our detailed construction process Appendix A. We have modified our discussion in Section 3.1 under the Frontal Alignment discussion and Section 3.2 of our main paper to note this as well to avoid any confusion. We also refer the reviewer to Figure 8 which shows an example of the Canonical Orientation question which contains some images that have the label “Cannot be determined” for instance for the images of rocks.
> > 4. Based on the above, I suspect that some samples may be ambiguous. However, the paper does not describe any quality control process to ensure dataset accuracy. For a benchmark, it is generally expected that every sample be manually verified to guarantee correctness.
>
> In our dataset there are two types of annotations. First, the vast majority of our annotations used data with complete 3D information about the objects, and, thus, their orientation information is completely accurate as we have full information about them.  We have manually inspected many of these samples, and in Table 5 we also conducted a human study which reported human agreement as high as 100%.  In other words, the kind of study the reviewer suggests has already been done, but, as noted, by design it is infeasible for those questions to be incorrect as we had complete 3D information.  Second, a very small number of samples that used the COCO were manually annotated, which we also verified, but as seen in Figure 2 account for a very small percentage of only two questions.
>
> > 5. While the paper evaluates several state-of-the-art models, it omits important models such as the InternVL series, and for the Qwen family, only the 3B variant is tested without including larger models.
>
> Thank you for your comment. We have added InternVL3.5 14B and 30B-A3B-Inst. as well as QWEN3 8B and 32B models to Table 2 and 3 of our paper (we did not include the 38B as its size makes it infeasible to use with our resources), which we partly reproduce below. We will highlight that the most recent InternVL model was released on August 26, 2025 which is after the period for recent work for ICLR which is July 24, 2025, which makes it an example of concurrent work.
>
> We can see that the InterVL and QWEN3 models appear to have improved average results versus other models like LLaVa and Mantis.  However, these are not universal, as some question types like Canonical Orientation and Viewer Scene Direction they do poorly. For example, on Canonical Orientation the InternVL models perform worse than LLaVa, which highlights how models can still struggle understanding what is the correct canonical state for objects in a scene. This highlights that there is still a gap in recent MLLM’s when it comes to orientation related reasoning.
>
>
> | Model | View Parallelism |               | Direction Facing |               | Single-axis Rotation |               | Compound Rotation |               | Inter-object Direction |               | Viewer-scene Direction |               | Canonical Orientation |               | Average |        |
> |-------|------------------|---------------|------------------|---------------|-----------------------|---------------|--------------------|---------------|-------------------------|---------------|-------------------------|---------------|------------------------|---------------|---------|--------|
> |       | Coarse (C)       | Granular (G)  | Coarse (C)       | Granular (G)  | Coarse (C)            | Granular (G)  | Coarse (C)         | Granular (G)  | Coarse (C)              | Granular (G)  | Coarse (C)              | Granular (G)  | Coarse (C)             | Granular (G)  | Avg-C   | Avg-G  |
> | IntVL-14B-Inst.  | 63.4 | 37.2 | 37.9 | 48.6 | 30.3 | 21.4 | 26.4 | 7.2 | 4.3 | 15.0 | 94.3 | 44.1 | 7.0  | 16.0 | 37.7 | 27.1 |
> | IntVL-30B-A3B-Inst.   | 72.1 | 33.9 | 46.6 | 50.0 | 17.4 | 17.7 | 33.3 | 6.8 | 19.3 | 10.0 | 90.1 | 35.0 | 8.3  | 22.2 | 41.0 | 25.1 |
> | Qwen3-8B-Inst.  | 73.7 | 45.6 | 50.8 | 54.8 | 32.9 | 45.9 | 49.0 | 7.5  | 18.2 | 21.9  | 79.1  | 37.2  | 6.4   | 21.1  | **44.3** | 33.4 |
> | Qwen3-32B-Inst  | 75.8 | 54.4 | 61.1 | 59.2 | 31.6 | 44.1 | 39.6 | 8.6  | 9.2  | 20.5  | 78.9  | 46.1  | 7.0   | 25.5  | 43.3 | **36.9** |
> | Mantis-Idfs-8B | 57.8 | 33.0 | 22.5 | 12.7 | 25.7 | 23.4 | 25.9 | 6.6 | 17.6 | 9.0 | 55.4 | 24.5 | 48.8 | 41.0 | 34.5 | 17.5 |
> | LLaVA-v1.6-34B | 52.8 | 35.3 | 32.6 | 26.4 | 22.1 | 26.2 | 13.0 | 4.3 | 16.2 | 14.8 | 34.8 | 25.4 | 9.9 | 11.6 | 25.9 | 20.5 |

---

> ### Author Response · Authors · 2025-11-21
>
> > 6. Lines 418–419: The observed difference may stem from variations in training data, so this conclusion should not be drawn too hastily.
>
> Below we report results on a held-out test set results of an ablation of fusion methods using a Qwen2.5-3B-Inst. backbone trained on our dataset for 2000 steps.  We find that the token-based fusion method performs the best.  However, this is also the default fusion method for the Qwen model, and, thus, this could simply just bias the model to prefer token-based fusion.  Completely controlling for all the various factors to validate architecture choices would require training these MLLMs from scratch.  Therefore, these results should simply be seen as an observation that needs verification.
>
> | Model | View Parallelism |               | Direction Facing |               | Single-axis Rotation |               | Compound Rotation |               | Inter-object Direction |               | Viewer-scene Direction |               | Canonical Orientation |               | Average |        |
> |-------|------------------|---------------|------------------|---------------|-----------------------|---------------|--------------------|---------------|-------------------------|---------------|-------------------------|---------------|------------------------|---------------|---------|--------|
> |       | Coarse (C)       | Granular (G)  | Coarse (C)       | Granular (G)  | Coarse (C)            | Granular (G)  | Coarse (C)         | Granular (G)  | Coarse (C)              | Granular (G)  | Coarse (C)              | Granular (G)  | Coarse (C)             | Granular (G)  | Avg-C   | Avg-G  |
> | Linear Proj  |55.9 |39.4 | 19.4 | 0.0 |  36.6 | 28.0 | 55.9 | 5.6 | 25.0 | 17.8 | 77.0 | 26.2 | 0.0 | 0.0 | 38.5 | 16.7
> | Linear Proj+Btlneck | 55.6 | 32.4 | 26.1 | 0.0 | 19.1 | 23.8 | 30.1 | 5.0 | 19.0 |10.8 | 41.6 | 18.8 | 12.9 | 2.9 | 29.2 | 13.4
> | Token based fusion | 87.2 | 72.4 | 75.0 | 2.5 | 64.9 | 65.6 | 63.1 | 14.9 | 67.7 | 56.5 | 94.1 | 69.5 | 42.9 | 35.2 | 70.7 | 45.2
>
> We have added a discussion about confounding factors in Section 4.1 of our paper and also included these experiments and discussion in Appendix H.3.
>
>
> > 7. Table 5: Please correct the label from “GPT-4 O” to “GPT-4o.”
>
> We have updated the label to reflect this thank you.
> > 8. The paper claims that the proposed systematic approach isolates orientation understanding from scene perception skills and minimizes confounding factors such as object recognition difficulty, scene clutter, linguistic ambiguity, and contextual distractions that affect existing benchmarks. However, no experiments or examples are provided to substantiate these claims.
>
> In our dataset we have carefully designed our questions for both the prompt and image to reduce the amount of different confounding factors that are present in past datasets. We include aspects like: defining axes, defining what canonical view is, we direct the reviewer to Appendix G and H which showcases these elements present in our questions.
>
> We can see in examples like Figure 16 we utilize a bounding box which helps the model/rater determine which object to focus on; this helps reduce object recognition difficulty. For scene clutter we can look at Figure 26 which focuses on particular objects without any objects being present in the image. Figure 18 showcases an example of reducing linguistic ambiguity, by describing what the MLLM should look at and identify its front facing surface, additionally objects that have ambiguous front facing surface like a table we have labelled those as “Cannot be determined”. For contextual distractions we can also look to Figure 21 which contains backgrounds that do not influence the exact orientation of the object.
>
> We have added this discussion to Appendix A. and provide a reference to this section of the appendix in the main paper under Section 3.2
>
> > 9. In Section 3.1, the definition of the viewing plane is not clearly explained.
>
> We define the viewing plane as the imaginary plane onto which a scene is projected onto. We have added this definition to the paper

---

### Official Review · Reviewer_dzt4 · 2025-11-01

**Soundness:** 2
**Presentation:** 2
**Contribution:** 3
**Rating:** 6
**Confidence:** 4

**Summary:**

The paper proposes DORI, a diagnostic benchmark for orientation understanding in MLLMs across four dimensions (frontal alignment, relative orientation, rotational transformation, canonical orientation).
It uses standardized MCQ prompts (with a Cannot be determined option) and reports that models handle coarse judgments better than granular angles; token-based fusion appears stronger than linear projection.
LoRA fine-tuning on DORI reportedly transfers to external spatial benchmarks.

**Strengths:**

The paper cleanly isolates orientation understanding into complementary abilities (frontal, relative, rotational, canonical) and tests them with a coarse–granular design that probes both category and precise angle reasoning.
The benchmark is well engineered, and broad model coverage exposes consistent weaknesses.
Findings are actionable, making DORI a practical diagnostic tool for geometry-sensitive applications.

**Weaknesses:**

- Since prompting is part of the measurement apparatus, please ablate the components to quantify their contribution and ensure models aren’t over-relying on the scaffold rather than vision.

- Ground-truth fidelity and metric design: While synthetic sources yield precise angles, human-annotated natural images can have ambiguous frontality (e.g., symmetric furniture) and unknown camera intrinsics, which may distort a fixed discrete angle taxonomy.

- Architectural claims need stronger controls: The observation that token-based integration > linear projection is compelling but potentially confounded by pretraining data or instruction tuning.

- Human study scale and reporting: Human evaluation covers 30 examples per type with seven experts. This is useful but small.

**Questions:**

- Have you tried free-form numeric responses (regression-style) and then quantized at evaluation time? Do model rankings persist? Please share results with permuted answer choices and removed examples section to quantify prompt-component effects.

- For the claim that token-based fusion > linear projection, can you provide experiments with the same visual backbone and identical instruction-tuning, changing only the fusion scheme? Any results with feature token counts swept to test capacity vs mechanism? (If it's not possible, that's understandable)

- Would you consider scaling the human study to 300–500 items with crowdworkers + expert adjudication, and report results to better anchor the human–model gap?

---

> ### Author Response · Authors · 2025-11-21
>
> We thank the reviewer for their comments which we have used to revise our paper.  We respond to individual comments below.
> > 1. Ground-truth fidelity and metric design: While synthetic sources yield precise angles, human-annotated natural images can have ambiguous frontality (e.g., symmetric furniture) and unknown camera intrinsics, which may distort a fixed discrete angle taxonomy.
>
> DORI accounts for ambiguities by including the option  “Cannot be determined.” Figure 1 and 27 both provide examples, but other objects with unclear frontality like pizza, bowl, etc all have the same behavior as mentioned in our detailed construction process Appendix A. We have modified our discussion in Section 3.1 under the Frontal Alignment discussion and Section 3.2 of our main paper to note this as well to avoid any confusion. We also refer the reviewer to Figure 8 which shows an example of the Canonical Orientation question which contains some images that have the label “Cannot be determined” for instance for the images of rocks.
>
>
> > 4. Architectural claims need stronger controls: The observation that token-based integration > linear projection is compelling but potentially confounded by pretraining data or instruction tuning.
> > 5. For the claim that token-based fusion > linear projection, can you provide experiments with the same visual backbone and identical instruction-tuning, changing only the fusion scheme? Any results with feature token counts swept to test capacity vs mechanism? (If it's not possible, that's understandable)
>
> Below we report results on a held-out test set results of an ablation of fusion methods using a Qwen2.5-3B-Inst. backbone trained on our dataset for 2000 steps.  We find that the token-based fusion method performs the best.  However, this is also the default fusion method for the Qwen model, and, thus, this could simply just bias the model to prefer token-based fusion.  Completely controlling for all the various factors to validate architecture choices would require training these MLLMs from scratch.  Therefore, these results should simply be seen as an observation that needs verification.
>
> | Model | View Parallelism |               | Direction Facing |               | Single-axis Rotation |               | Compound Rotation |               | Inter-object Direction |               | Viewer-scene Direction |               | Canonical Orientation |               | Average |        |
> |-------|------------------|---------------|------------------|---------------|-----------------------|---------------|--------------------|---------------|-------------------------|---------------|-------------------------|---------------|------------------------|---------------|---------|--------|
> |       | Coarse (C)       | Granular (G)  | Coarse (C)       | Granular (G)  | Coarse (C)            | Granular (G)  | Coarse (C)         | Granular (G)  | Coarse (C)              | Granular (G)  | Coarse (C)              | Granular (G)  | Coarse (C)             | Granular (G)  | Avg-C   | Avg-G  |
> | Linear Proj  |55.9 |39.4 | 19.4 | 0.0 |  36.6 | 28.0 | 55.9 | 5.6 | 25.0 | 17.8 | 77.0 | 26.2 | 0.0 | 0.0 | 38.5 | 16.7
> | Linear Proj+Btlneck | 55.6 | 32.4 | 26.1 | 0.0 | 19.1 | 23.8 | 30.1 | 5.0 | 19.0 |10.8 | 41.6 | 18.8 | 12.9 | 2.9 | 29.2 | 13.4
> | Token based fusion | 87.2 | 72.4 | 75.0 | 2.5 | 64.9 | 65.6 | 63.1 | 14.9 | 67.7 | 56.5 | 94.1 | 69.5 | 42.9 | 35.2 | 70.7 | 45.2
>
> We have added a discussion about confounding factors in Section 4.1 of our paper and also included these experiments and discussion in Appendix H.3.

---

> ### Author Response · Authors · 2025-12-01
>
> > 2. Have you tried free-form numeric responses (regression-style) and then quantized at evaluation time? Do model rankings persist? Please share results with permuted answer choices and removed examples section to quantify prompt-component effects.
>
> > 3. Since prompting is part of the measurement apparatus, please ablate the components to quantify their contribution and ensure models aren’t over-relying on the scaffold rather than vision.
>
> To examine the effect of prompt engineering on orientation understanding, we conducted ablation studies on 8 models by removing key prompt components: structural formatting, answer examples, detailed task conceptualizations, and step-by-step reasoning instructions. The results are included in Table 7 and provided below. Corresponding discussions are also updated and expanded in Appendix Section C:
> | Model | VP-C | VP-G | DF-C | DF-G | SR-C | SR-G | CR-C | CR-G | IO-C | IO-G | VS-C | VS-G | OR-C | OR-G | AVG-C | AVG-G |
> |-------|------|------|------|------|------|------|------|------|------|------|------|------|------|------|--------|--------|
> | LLaVA-v1.6-13B | 57.9 | 33.0 | 25.4 | 0.01 | 20.2 | 16.5 | 27.7 | 0.01 | 16.7 | 12.2 | 60.4 | 19.8 | 0.01 | 0.0 | 29.7 | 11.6 |
> | Yi-VL-6B | 52.4 | 37.8 | 23.9 | 20.3 | 36.8 | 15.9 | 19.7 | 21.6 | 11.0 | 16.5 | 78.6 | 18.55 | 29.1 | 12.9 | 35.9 | 20.5 |
> | Mantis-CLIP | 40.5 | 9.1 | 6.9 | 6.8 | 6.2 | 2.6 | 0.9 | 1.8 | 0.2 | 0.0 | 57.9 | 1.9 | 36.3 | 46.7 | 22.4 | 6.7 |
> | Mantis-If2-8B | 51.2 | 29.9 | 29.7 | 13.6 | 15.3 | 10.5 | 62.5 | 5.0 | 5.2 | 18.4 | 85.9 | 32.1 | 45.9 | 46.1 | 53.4 | 19.9 |
> | LLava_next-8B | 56.9 | 1.8 | 25.5 | 29.9 | 22.2 | 11.7 | 34.1 | 6.6 | 24.3 | 10.0 | 66.0 | 26.4 | 11.3 | 11.0 | 34.3 | 13.9 |
> | DS-1.3B-Chat | 40.7 | 30.2 | 22.6 | 21.8 | 12.2 | 19.8 | 35.6 | 5.6 | 16.2 | 14.2 | 31.1 | 21.6 | 35.2 | 9.8 | 27.6 | 17.5 |
> | DS-7B-Base | 45.2 | 35.0 | 21.6 | 21.2 | 25.4 | 18.0 | 2.8 | 5.7 | 15.0 | 11.7 | 33.6 | 25.7 | 3.7 | 16.5 | 21.0 | 19.1 |
> | DS-7B-Chat | 48.1 | 32.4 | 27.9 | 25.7 | 47.0 | 19.5 | 1.0 | 5.1 | 49.6 | 15.8 | 36.6 | 22.7 | 32.9 | 10.70 | 34.7 | 18.8 |
>
> Overall, we have observed that most models exhibit severe degradation with unstructured prompts, with LLaVA and LLava-Next showing near-complete collapse on several fine-grained tasks, indicating that structured prompts are critical for multi-step spatial reasoning. However, prompt sensitivity varies dramatically across model families. Mantis-Idfics-8B counterintuitively improves under unstructured prompts, suggesting that structured prompts may impose reasoning templates incompatible with its interleaved vision-language architecture. DS-7B-Chat exhibits non-monotonic behavior, improving on some tasks while collapsing on others, indicating task-dependent interference. Notably, Canonical Orientation remains challenging regardless of prompt structure across all models, suggesting that prompt engineering alone cannot compensate for fundamental gaps in geometric priors.
>
>
>
> > 6. Human study scale and reporting: Human evaluation covers 30 examples per type with seven experts. This is useful but small.
> > 7. Would you consider scaling the human study to 300–500 items with crowdworkers + expert adjudication, and report results to better anchor the human–model gap?
>
> Our human evaluation’s scale already provides a strong anchor of the human-model gap as exemplified by its scale compared to other recent benchmarks.  Specifically, it contains a total of 420 samples (14 types x 30 questions), which means our human study is similar in size to the entire datasets from recent benchmarks summarized in Table 1 like Spaital-MM [A], BLINK [B], KIVA [C], and SR-Bench [D].  A request of 500 samples per question would result in 500 x 14 = 7000 total samples- infeasible to do in the rebuttal period.  That said, to help further ground these results we expanded our human study to include an additional 20 samples per question from 7 additional annotators, resulting in a total of 14 x 50 = 700 samples from 14 annotators.  These results have been updated in Table 5 of our paper, but any changes from our previous results are negligible.  Thus, we can conclude that further scaling is unlikely to provide additional insights as the results would be largely unaffected.
>
> [A] Fatemeh Shiri, Xiao-Yu Guo, Mona Far, Xin Yu, Reza Haf, and Yuan-Fang Li. An empirical analysis on spatial reasoning capabilities of large multimodal models. EMNLP, 2024.
>
> [B] Blink: Multimodal large language models can see but not perceive. ECCV 2024.
>
> [C] Eunice Yiu, Maan Qraitem, Anisa Noor Majhi, Charlie Wong, Yutong Bai, Shiry Ginosar, Alison Gopnik, and Kate Saenko. KiVA: Kid-inspired visual analogies for testing large multimodal models. ICLR 2025
>
> [D] Ilias Stogiannidis, Steven McDonagh, and Sotirios A. Tsaftaris. Mind the gap: Benchmarking
> spatial reasoning in vision-language models. In Greeks in AI Symposium 2025

---

### Author Response · Authors · 2025-12-02
**Summary of Reviews and Rebuttal**

Dear AC,

Thank you for your efforts through this unusual ICLR cycle.  In an effort to assist you, we have provided a summary of the discussion across reviewers.

Overall the reviewers found that our dataset provides a comprehensive and well-engineered benchmark for orientation reasoning (Reviewer dzt4, Vs7f, 9Ynm), that clearly distinguishes from prior work (Reviewer Vs7f, X2sf), with experiments on a broad range of multimodal models (Reviewer dzt4, 9Ynm, X2sf) that provides actionable insights that would be useful for downstream applications (Reviewer dzt4, 9Ynm, X2sf), and is within a well-written paper (Reviewer Vs7f, X2sf).

## Summary of Questions

1. All reviewers asked about how we handle cases where orientation is ambiguous (e.g., symmetric objects like balls or tables).  In our response we acknowledged that we considered these cases in our dataset’s construction, and the correct answer for these types of questions is that the orientation “Cannot be determined” and provided some examples (e.g., in Figure 1).  Furthermore, we added a comment to note how these cases were handled to our paper around the discussion of “frontal” alignment questions (where a majority of reviewers raised their question) to avoid any further confusion.

2. Reviewers dzt4 and Vs7f both highlighted that some observations about correlations between architectures and performance on our benchmark are confounded by other factors, such as differences in pretraining data.  We did provide an experiment on Qwen models to validate these observations, but we noted that this experiment was still imperfect.  Specifically,  properly validating these observations require re-training each MLLM from scratch, which is not possible (and Reviewer dzt4 had acknowledged that it likely would not be possible in their review).   We added these observations, experiments, and a note on confounding variables to our paper.

3. Reviewers dzt4, Vs7f, and 9Ynm all requested some additional experiments.  We note that none of these experiments were requested by more than one reviewer, which raises questions as to their importance, but, regardless, we provided the desired results.  For example, Reviewer Vs7f asked for additional MLLM results, which we provided using the requested models, but we would also note that our initial submission already contained results on 18 models- a substantial amount. Reviewer dzt4 asked for more human annotations that we used to provide more insight into the human-machine gap, but after completing this task we found it made a negligible difference in our analysis.  We argue that all possible requests have been fulfilled and they have simply further validated the importance and quality of our benchmark.

4. Reviewers Vs7f, 9Ynm, and X2sf made suggestions for improving our paper’s clarity.  We made the requested adjustments to our paper based on these comments in our revised version.

As seen above, we have addressed the reviewers comments, who all acknowledged the benefits of our benchmark. We hope the AC finds this summary useful.  We are happy to answer any further questions.

Best,

Authors of Submission 13396

---

### Meta-Review · Area_Chair_UMQF · 2026-01-06

**Summary:**

After considering the reviewers comments and the authors rebuttal, I agree that many of the technical concerns raised during the review process have been adequately addressed in the revision. These include questions about ambiguity in orientation labeling, sensitivity to prompt design, requests for additional experiments, and issues related to clarity and presentation.

However, I recommend rejection based on the overall scope and impact of the contribution.

The paper focuses narrowly on evaluating object orientation understanding as an isolated capability. While orientation perception is an important component of spatial reasoning, the benchmark intentionally abstracts away other closely related factors such as object geometry, affordances, interactions, and task level consequences. As a result, the contribution is primarily a specialized diagnostic benchmark rather than a more comprehensive evaluation of spatial or embodied reasoning.

In addition, some of the broader conclusions drawn from the benchmark are limited in novelty. For example, the paper hypothesizes that poor model performance stems from common pretraining practices, noting that most evaluated MLLMs rely on CLIP style contrastive objectives that emphasize high level image text semantic alignment rather than explicit geometric understanding. This limitation of CLIP style pretraining has already been widely discussed in prior work, and the benchmark does not provide new evidence that substantially advances this line of understanding or translates it into actionable guidance for model design.

For ICLR, benchmark papers are generally expected to either cover a broader portion of the target capability or to clearly demonstrate that a narrowly scoped evaluation leads to actionable insights for model development or downstream task performance. Although the authors motivate orientation understanding as foundational for applications such as robotics and augmented reality, the paper does not provide sufficient evidence that performance on the proposed benchmark translates to improvements in real world tasks or meaningfully informs architectural or training decisions.

Overall, while the work is technically sound and well executed, the limited scope of the benchmark constrains its broader impact and general interest for the ICLR audience.

**Reviewer Concerns:**

Addressed Reviewer Concerns

The rebuttal satisfactorily addressed most of the technical concerns raised by the reviewers. In particular, concerns regarding ambiguity in object orientation were clarified through the consistent use of a “Cannot be determined” option, additional examples, and expanded discussion of dataset construction and labeling procedures. Reviewers’ requests for additional experiments were largely fulfilled, including prompt ablations, expanded human evaluation, and the inclusion of additional model results. Questions about prompt sensitivity, rotation conventions, viewing plane definitions, and presentation issues were also addressed with added experiments, clarifications, and revisions to figures and tables. Overall, the rebuttal demonstrated careful engagement with reviewer feedback and improved the clarity and completeness of the work.

Outstanding Reviewer Concerns

Despite these improvements, some higher level concerns remain. Several reviewers questioned whether conclusions about architectural design choices could be drawn without stronger controls and noted that observed differences may be confounded by pretraining or instruction tuning. More importantly, reviewers expressed reservations about the narrow scope of the benchmark and the lack of direct empirical validation linking benchmark performance to downstream real world tasks such as robotic manipulation or navigation. While the rebuttal argues for the foundational importance of orientation understanding, it does not fully resolve concerns about the broader impact and generality of the contribution. These scope and impact related concerns remain outstanding and ultimately informed the recommendation.

**Reviewer Scores:**

dzt4 -> Maintained the score since the concerns are partially addressed
Vs7f -> Maintained the score since the concerns are partially addressed
9Ynm -> Maintained the score since the concerns are partially addressed
X2sf -> Lowered the score since reviewer was initially mistaken about few points

---

### Decision · Program_Chairs · 2026-01-26

Reject